# Balancing stability and flexibility when reshaping archaeal membranes

**Miguel Amaral[1†], Felix Frey[1†], Xiuyun Jiang[2], Buzz Baum[3], Anđela Šarić[1*]**

[1]Institute of Science and Technology Austria, Klosterneuburg, Austria; [2]Department of Physics and Astronomy, University College London, London, United States; [3]MRC Laboratory of Molecular Biology, Cambridge, United Kingdom

## eLife Assessment

This **fundamental** study characterizes the mechanics and stability of bolalipids from archaeal membranes using a minimalist, physics-based computational model. The authors present a robust mesoscale model of bolalipids-containing membranes, systematically evaluating it across diverse membrane configurations. The results are **compelling**, demonstrating that the incorporation of bolalipids and regular bilayer lipids in archaeal membranes significantly enhances membrane fluidity and structural stability.

**\*For correspondence:**
andela.saric@ist.ac.at

†These authors contributed equally to this work

**Competing interest:** The authors declare that no competing interests exist.

**Abstract** Cellular membranes differ across the tree of life. In most bacteria and eukaryotes, single-headed lipids self-assemble into flexible bilayer membranes. By contrast, thermophilic archaea tend to possess bilayer lipids together with double-headed, monolayer spanning bolalipids, which are thought to enable cells to survive in harsh environments. Here, using a minimal computational model for bolalipid membranes, we explore the trade-offs at play when forming membranes. We find that flexible bolalipids form membranes that resemble bilayer membranes because they are able to assume a U-shaped conformation. Conversely, rigid bolalipids, which resemble the bolalipids with cyclic groups found in thermophilic archaea, take on a straight conformation and form membranes that are stiff and prone to pore formation when they undergo changes in shape. Strikingly, however, the inclusion of small amounts of bilayer lipids in a bolalipid membrane is enough to achieve fluid bolalipid membranes that are both stable and flexible, resolving this trade-off. Our study suggests a mechanism by which archaea can tune the material properties of their membranes as and when required to enable them to survive in harsh environments and to undergo essential membrane remodelling events like cell division.

## Introduction

A bounding membrane separates the interior of every biological cell from its extracellular environment and gives the cell its shape. To perform its function, this bounding membrane must be sturdy and impermeable to the leakage of small charged molecules like ions, to allow the generation of an electrochemical gradient (*Phillips et al., 2012*), but also flexible enough to be remodelled during essential cellular processes like cell division and vesicle formation (*McMahon and Gallop, 2005*). These requirements are partially conflicting and difficult to combine.

Interestingly, two different generic membrane designs have evolved across the tree of life to solve this problem (*Gould, 2018*). Bacterial and eukaryotic membranes possess fatty acid lipids that have a hydrophilic head group linked to hydrophobic tails via ester-linkages, which self-assemble into bilayer membranes (*Albers and Meyer, 2011*). By contrast, archaeal membranes are constructed from branched isoprenoid lipids. These can include cyclopentane rings and are attached via an ether

linkage to one (e.g. archaeol, *Figure 1A* left) or two hydrophilic heads, leading them to be called bipolar lipids or simply bolalipids (e.g. caldarchaeol, *Figure 1A* right) (*Albers and Meyer, 2011*). The hydrophilic heads can be composed of different functional groups with phosphatidyl and sugar being the most relevant moieties. For bolalipids, the two head groups at either end of the molecule are typically distinct (*Figure 1A* right) (*Oger and Cario, 2013*). As an ensemble, bilayer and bolalipids can self-assemble into fluid lipid membranes of different architectures – as bilayer lipid membranes, bolalipid membranes, or mixture membranes (*Figure 1B* right).

Like for bacteria and eukaryotes, archaea must keep their lipid membranes in a fluid state (homeo-viscous adaptation). This is important even under extreme environmental conditions, such as hot and cold temperatures, or high and low pH values (*Chong, 2010*). Because of this, many archaea adapt to changes in their environment by tuning the lipid composition of their membranes: altering the ratio between bola- and bilayer lipids in their membranes (*Tourte et al., 2020*; *Kim et al., 2019*) and/or by changing the number of cyclopentane rings in their lipid tails, which are believed to make lipid molecules more rigid (*Oger and Cario, 2013*). For example, *Thermococcus kodakarensis* increases its tetraether bolalipid ratio from around 50% to over 80% when the temperature of the environment increases from 60 to 85 °C (*Siliakus et al., 2017*). Along the same lines, the cell membrane of *Sulfolobus acidocaldarius* can contain over 90% of bolalipids with up to eight cyclopentane rings at 70 °C and pH 2.5 (*Oger and Cario, 2013*; *Grogan, 1989*). It is worth mentioning that in exceptional cases bacteria also synthesise bolalipids in response to high temperatures (*Sahonero-Canavesi et al., 2022*), highlighting that the study of bolalipid membranes is relevant not only for archaeal biology but also from a general membrane biophysics perspective. Besides temperature adaptation, archaeal lipids also exhibit dense packing, high viscosity, and low porosity to small molecules like protons (*Siliakus et al., 2017*).

All this makes the study of reshaping archaeal membranes extremely interesting. Unfortunately, it is fairly challenging to investigate archaeal cells experimentally due to their unusual chemistry and extreme living conditions (*van Wolferen et al., 2022*). Therefore, most of what we know about archaeal tetraether lipid membranes thus far has been collected from studying in vitro reconstituted membranes. For instance, the conformation of individual lipids in bolalipid membranes was studied at the water-air interface (*Bakowsky et al., 2000*; *Jeworrek et al., 2011*) or using NMR experiments on lipid vesicles (*Brownholland et al., 2009*). This suggested the existence of U-shaped lipid conformations in archaeal type membranes (*Figure 1B*). Moreover, in vitro reconstituted vesicles primarily composed of bolalipids can fuse with influenza virus particles at similar kinetic rates compared to bilayer vesicles, further suggesting that bolalipids exist in U-shape allowing for membrane remodelling and fusion (*Bhattacharya et al., 2024*). Membrane properties like bending stiffness (*Vitkova et al., 2020*) or lipid phase (*Chang, 1994*) have also been measured in vesicles prepared from archaeal tetraether lipids to demonstrate that archaeal lipids derived membranes are exceptional in being stable up to temperatures of 80 °C (*Chang, 1994*). Moreover, experiments with lipid vesicles made from synthetic bilayer lipids that include cyclopentane rings, which naturally appear in lipids of extremophilic archaea, showed that increasing the number of these rings increases membrane rigidity (*Batishchev et al., 2020*).

From the viewpoint of membrane physics, the remodelling of bilayer membranes has been studied for decades using continuum models and computer simulations (*Seifert, 1997*; *Noguchi, 2009*; *Frey and Idema, 2021*). By contrast, only a few studies have investigated bolalipid membranes applying computational or theoretical tools (*Huguet et al., 2017*; *Galimzyanov et al., 2016*). Specifically, the pore closure time in bolalipid membranes, and the role of cyclopentane rings for membrane properties have been investigated using all-atom simulations, showing decreased lateral mobility, reduced permeability to water, and increased lipid packing (*Galimzyanov et al., 2025*; *Chugunov et al., 2025*; *Pineda De Castro et al., 2016*). Moreover, using coarse-grained simulations, it was suggested that bolalipid membranes are thicker (*Bulacu et al., 2012*), exhibit a gel-to-liquid phase transition at higher temperature (*Davis et al., 2007*), and exhibit a reduced diffusivity (*Dey and Saha, 2020*). However, little research has been devoted to investigating mechanics and the reshaping of bolalipid membranes at the mesoscale despite the obvious importance of this question from evolutionary, biophysics, and biotechnological perspectives and although different membrane physics is expected to manifest.

Here, we have designed a minimal model of archaeal membranes that can be used to explore the impact of both bolalipids and bilayer lipids on membrane biophysics and membrane reshaping. Our

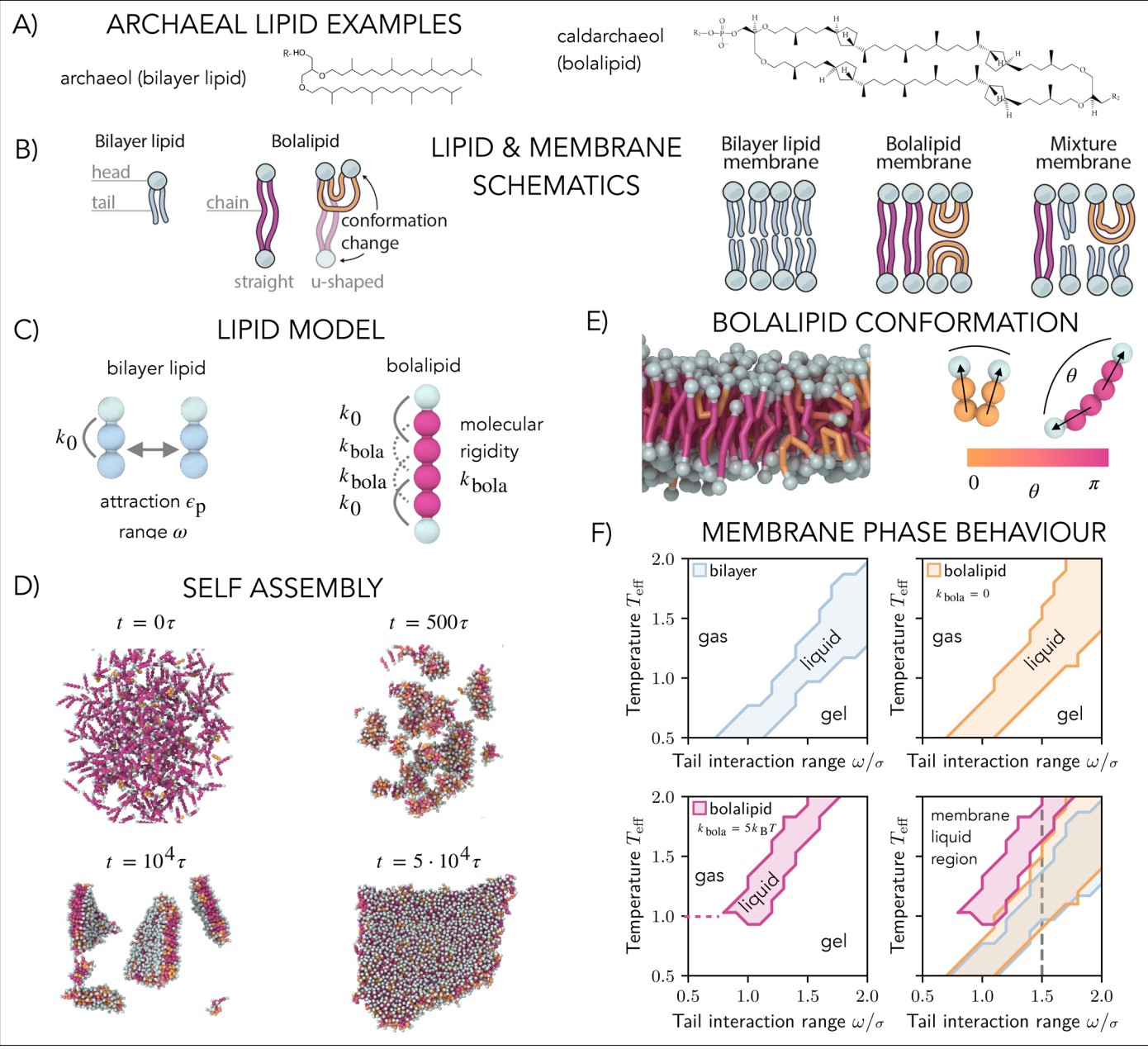

**Figure 1.** Computational model and phase space of bilayer and bolalipid membranes. (**A**) Structure of the diether bilayer lipid archaeol (left) and the tetraether bolalipid caldarchaeol, including four cyclopentane rings (right), both present in the membrane of *Sulfolobus acidocaldarius*, a common archaeal model system that lives at high temperatures and low pH (*Rastädter et al., 2020*). The hydrophilic head of a bolalipid can be composed of different functional groups represented by R1 and R2 (right). (**B**) Schematics for a bilayer lipid, a bolalipid and its two in-membrane conformations, membranes made of bilayer molecules only, bolalipid molecules only, and a mixture of the two (left to right). (**C**) Bilayer lipid (left) is described with one head bead and two tail beads straightened by an angular potential of strength $k_0$. Tail beads of different lipids attract with the strength $\epsilon_p$ and the range $\omega$. Bolalipids (right) consist of two bilayer lipids connected by a bond and straightened by an angular potential of strength $k_{\text{bola}}$. (**D**) Snapshots of bolalipids self-assembling into a flat membrane ($k_{\text{bola}} = 0.3 \, k_B T$). (**E**) Cross-section of self-assembled membrane (right), with bolalipids coloured according to their conformation: straight lipids in crimson and U-shaped lipids in orange. (**F**) Membrane phase behaviour: liquid, gel and gas regions as a function of the effective temperature $T_{\text{eff}}$ and tail interaction range $\omega$ for bilayer (top left) and membranes made of flexible $k_{\text{bola}} = 0$ (top right) and stiff $k_{\text{bola}} = 5 \, k_B T$ (bottom left) bolalipid molecules. Overlays of all liquid regions (bottom right) show that stiffer lipids exhibit fluid membrane region at higher temperatures. The dashed line marks $\omega = 1.5\sigma$, the value we used in the rest of the work.

The online version of this article includes the following video(s) for figure 1:

**Figure 1—video 1.** Self-assembly of flexible bolalipid molecules into a flat membrane ($k_{\text{bola}} = 0$).

https://elifesciences.org/articles/105432/figures#fig1video1

*Figure 1 continued on next page*

*Figure 1 continued*

**Figure 1—video 2.** Close-up view of a flat membrane, with tail beads shown like sticks.

https://elifesciences.org/articles/105432/figures#fig1video2

**Figure 1—video 3.** Close-up view of a flat membrane, with tail beads shown like sticks.

https://elifesciences.org/articles/105432/figures#fig1video3

**Figure 1—video 4.** Close-up view of a flat membrane, with tail beads shown like sticks.

https://elifesciences.org/articles/105432/figures#fig1video4

coarse-grained molecular dynamics simulations show that the geometry of bolalipids is, through the effects of entropy alone, sufficient to shift the fluid phase of archaeal-type membranes so that they are stable at high temperatures. We show that membranes assembled from bolalipids can have a much higher bending rigidity than bilayer-derived membranes, and are more likely to resist membrane shape changes as in fission. During membrane deformation, stress in these bolalipid membranes is relieved by a small fraction of bolalipids taking up a U-shaped conformation, which renders them a mechanically switchable material. Without these U-shaped bolalipids, stress leads to the formation of large pores. However, when combined, a mixture of bilayer and bolalipids generates membranes that are stable at high temperature, which become softer as the fraction of bilayer lipids increases. Taken together, our results demonstrate how doping a bolalipid membrane with a small fraction of bilayer lipids can relieve the trade-off, enabling membranes to be both stable under extreme conditions and to be remodelled without leaking.

## Results

### Computational model

To study bolalipid membranes and compare them to bilayers, we extended the Cooke and Deserno model for bilayer membranes (*Cooke and Deserno, 2005*). In the original model for the bilayer a single bilayer lipid is represented by a chain of three nearly equally sized beads of diameter of $\sim 1\sigma$, where $\sigma$ is our distance unit and roughly maps to $1\,\text{nm}$ (*Figure 1C* left); one bead stands for the head group (cyan), while the others represent the hydrophobic tail (blue). Each adjacent pair of beads in a lipid is linked by a finite extensible nonlinear elastic (FENE) bond. The angle formed by the chain of three beads is kept near 180° via an angular potential with strength $k_0$, instead of the approximation by a bond between end beads of the original model (*Cooke and Deserno, 2005*).

While lipid heads interact exclusively through volume exclusion, the beads of lipid tails interact via a soft attractive potential of the strength $\epsilon_p$ and range $\omega$ (*Figure 1C* left), effectively modelling hydrophobic interaction in an implicit solvent. This interaction strength governs the membrane phase behaviour and can be interpreted as the effective temperature or reduced temperature $T_{\text{eff}} = k_B T / \epsilon_p$. As the distinction between scaling interactions ($T_{\text{eff}}$) or temperature ($T$) is not important for our analysis (see Appendix 1–Section 14), for simplicity, we refer to $T_{\text{eff}}$ as temperature in the following.

To model a bolalipid molecule, we joined two bilayer lipids so that a lipid molecule is formed with a head bead (cyan) that is linked to four tail beads (crimson) which are again linked to another head bead (*Figure 1C* right). In this way, both bilayer lipids and bolalipids share the same molecular structure and the same interactions between lipid beads. To decouple the effect of the connected geometry of the bolalipids from that of lipid asymmetry, we assume both head beads of a bolalipid to share the same properties. Bolalipids in archaeal membranes can differ in the number of cyclopentane rings or the branching of the tail and thus in the molecular stiffness (*Chong, 2010*; *Chong et al., 2012*). To represent this effect, we added two angular potentials between the second and the fourth and the third and the fifth tail bead with variable strength $k_{\text{bola}}$. By varying $k_{\text{bola}}$, we can control the molecular stiffness of the bolalipid molecules and thus model different types of bolalipids. The model is simulated within molecular dynamics implemented in the LAMMPS open-source package (*Thompson et al., 2022*), and simulations are visualized with OVITO *Stukowski, 2010*. To include the implicit effect of the surrounding water and to simulate membranes at vanishing tension, we used a Langevin thermostat combined with a barostat that kept the membrane at zero pressure in the $x - y$ plane. Details on the computational model are given in Appendix 1–Section 1.

## Self-assembly and phase behaviour

We first tested the ability of both the bilayer and bolalipid molecules to self-assemble into membranes. To do so, we placed dispersed lipids in a periodic 3D box. For both bilayer and bolalipids, we found that membranes self-assembled over a wide range of lipid interaction parameters (*Figure 1D*, *Figure 1—video 1*, Appendix 1–Section 2). We then explored the influence of flexibility in the bolalipid molecule. At small values of the molecular rigidity $k_{bola}$, single lipids are flexible and can thus adopt a range of possible conformations. The different conformations can be classified by the angle $\theta$ between the two lipid heads (*Figure 1E*). Two lipid conformations dominated the conformation distribution in the context of a membrane: the U-shaped conformation with both head beads on the same membrane leaflet ($\theta \approx 0$) and the straight conformation with one head bead in each opposing membrane leaflet ($\theta \approx \pi$) (*Figure 1—video 2*, *Figure 1*, Appendix 1–Section 3; for comparison, bilayer membranes in *Figure 1—video 3* and rigid bolalipid membranes in *Figure 1—video 4*). In the self-assembled bolalipid membrane, we marked bolalipids as being in the U-shape conformation if $\theta < \pi/2$ and in the straight conformation otherwise.

To study the phase behaviour of bilayer and bolalipid membranes, we analysed the diffusion of single lipids as a function of the lipid interaction parameters $\epsilon_p$ and $\omega$ (see Appendix 1–Section 4 and Appendix 1–Section 5). The diffusion constant $D$ exhibited a discontinuity as a function of the temperature, which marks the transition as the membrane moves from the gel phase to the liquid phase. The discontinuity occurred at different values of interaction strength $\epsilon_p$ and interaction range $\omega$ for bilayer and bolalipid lipids, and it also depended on the values of the molecular stiffness $k_{bola}$ (*Appendix 1—figure 4*). In these simulations, the disintegration of the membrane defined an upper limit to the liquid phase and the transition to the gas phase. Based on this classification, we plotted the phase diagram for bilayer membranes (*Figure 1F* top left), fully flexible bolalipid membranes ($k_{bola} = 0$, *Figure 1F* top right), and rigid bolalipid membranes ($k_{bola} = 5\,k_BT$, *Figure 1F* bottom left) as a function of the range of the hydrophobic interaction $\omega$ and the temperature $T_{eff}$. Membranes made of bilayers (blue) and flexible bolalipid molecules ($k_{bola} = 0$, orange) behaved similarly under these conditions (*Figure 1F* bottom right). Strikingly, just as observed in extremophile archaea, as the molecular stiffness of bolalipids increased ($k_{bola} = 5\,k_BT$, magenta), the liquid region was shifted toward higher temperatures and larger values of the interaction range $\omega$. This is due to the fact that bolalipid molecules are able to engage in more extensive interactions with partners when in the extended conformation, which helps to stabilize the membrane at higher temperatures.

## Bolalipid conformations and mechanical properties

To explore mechanical properties of bolalipid membranes, we chose $\omega = 1.5\sigma$ for the remainder of the work (dashed line in *Figure 1F* bottom right). *Figure 2A* shows the phase diagram for bolalipid membranes replotted as a function of temperature and bolalipid rigidity for the chosen interaction range. To be able to compare membranes made of bolalipids of different molecular rigidities, we needed to adjust temperature for each to reach similar fluidities, shown by dashed lines in *Figure 2A*. We then characterised the conformations of individual bolalipids in flat membranes, measured by the fraction of bolalipids in U-shape conformation $u_f$. We find that for flexible bolalipids ($k_{bola} = 0$), more than 50% of all lipids are in the U-shape conformation (*Figure 2B*). This fraction decreases with increasing rigidity of bolalipid molecules and vanishes around $k_{bola} \geq 2\,k_BT$, for which almost all bolalipids take up linear conformations.

This behaviour can be easily captured by considering bolalipids as a two-state system, with straight and U-shaped conformations, as argued before (*Appendix 1—figure 1*). We assumed that the energy of the straight conformation vanishes $E_s = 0$ and the energy of the U-shaped conformation reads $E_u = c_0 + c_1 k_{bola}$, where, for simplicity, we assumed that the energy linearly depends on $k_{bola}$ which is the only relevant energy scale in the system and the two constants $c_0$ and $c_1$, which determine the conformation energy of U-shaped bolalipids. The fraction of bolalipids in U-shape conformation then follows $u_f(k_{bola}) = 1/(1 + \exp(\beta(c_0 + c_1 k_{bola})))$, with $\beta = 1/(k_BT)$, as shown by the fit in *Figure 2B* (dashed grey line, $R^2 = 0.99$, Appendix 1–Section 6). For the fit, it appears that $c_0 < 0$, which implies that bolalipids in U-shape conformation are slightly favoured over straight bolalipids at $k_{bola} = 0$ (explored in Appendix 1–Section 6).

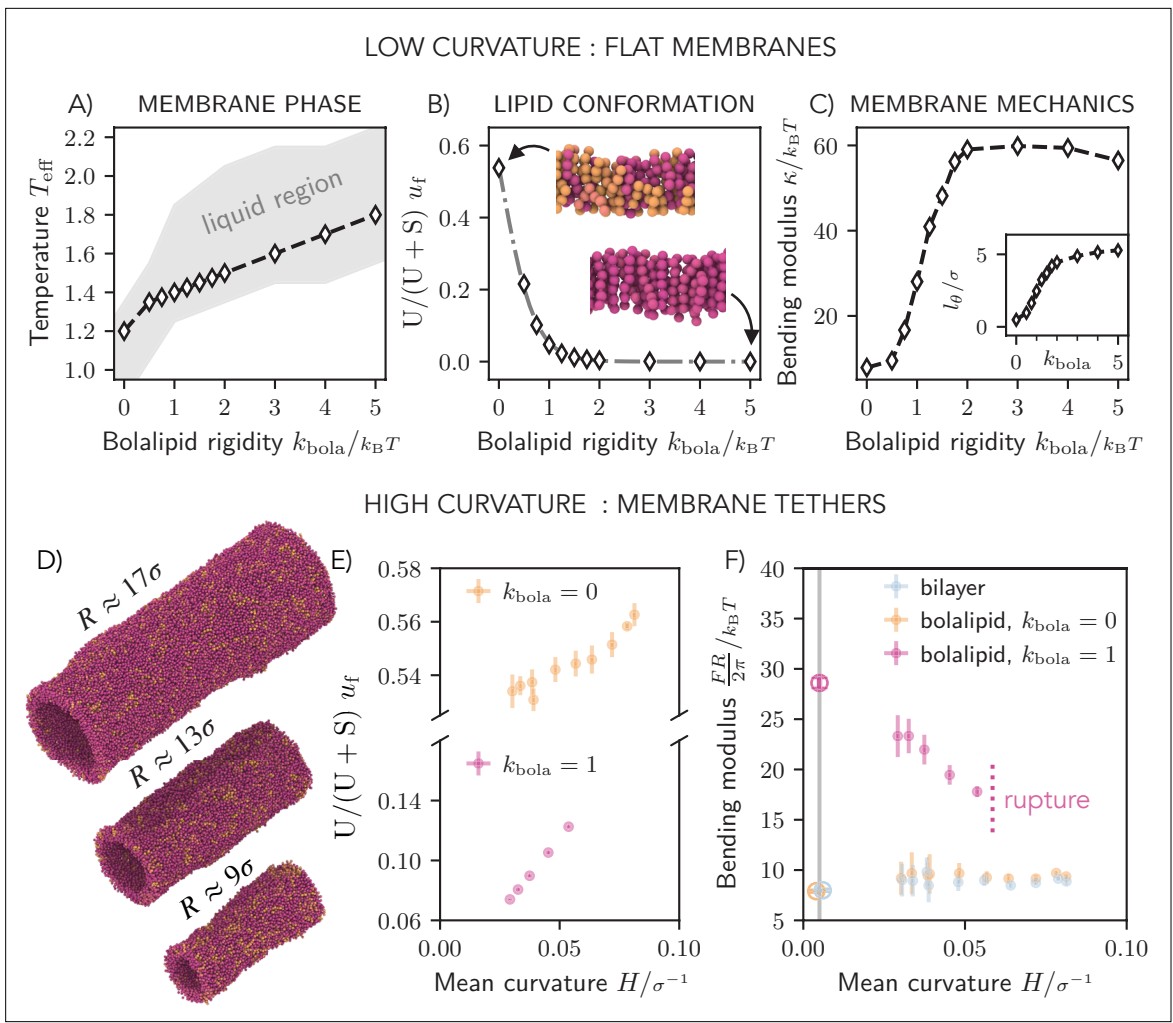

**Figure 2.** Mechanics of pure bolalipid membranes. (**A**) Liquid region as a function of temperature and bolalipid rigidity for pure bolalipid membranes (grey). The dashed line shows the bolalipid membranes of approximately same fluidities. (**B**) Fraction of bolalipids in the U-shape conformation ($\theta = 0$), fitted to $u_f(k_{bola}) = 1/(1 + \exp\left(\beta(-0.16 + 3k_{bola})\right))$ (grey dashed line) according to a two-state model. Insets: simulation snapshots with bolalipids coloured according to their conformations. (**C**) Bending modulus as a function of bolalipid molecule rigidity $k_{bola}$. Inset: Tilt persistence length $l_\theta$ as a function of bolalipid rigidity $k_{bola}$. (**D**) Snapshots of bolalipid membranes at the range of explored curvatures for $k_{bola} = 1k_BT$. (**E**) Fraction of bolalipid molecules in the U-shaped conformation as a function of the mean membrane curvature $H = 1/(2R)$ for membranes made of flexible ($k_{bola} = 0$) and semi-flexible ($k_{bola} = 1k_BT$) bolalipid molecules. (**F**) Bending modulus as a function of curvature. For the flat membrane ($H \approx 0$), the corresponding bending rigidity from (**C**) is marked by the vertical line and empty circles.

The online version of this article includes the following video(s) for figure 2:

**Figure 2—video 1.** Fluctuating flat membrane patch of bolalipid molecules at $k_{bola} = 0$ used for height fluctuation spectrum measurements.
https://elifesciences.org/articles/105432/figures#fig2video1

**Figure 2—video 2.** Cylindrical membranes after equilibration, made of bolalipid molecules of intermediate stiffness ($k_{bola} = 1\,k_BT$) at the largest simulated radius $R \approx 17\sigma$.
https://elifesciences.org/articles/105432/figures#fig2video2

## Bolalipid conformations and membrane rigidity

It was previously hypothesized that an increasing fraction of bolalipids in straight configuration would increase the membrane rigidity (*Vitkova et al., 2020*). To determine the membrane rigidity using our model, we assessed the height fluctuation spectrum $\langle h^2 \rangle$ of flat membranes in a periodic box (*Cooke and Deserno, 2005*; *May et al., 2007*). Interestingly, we found that the original theory of *Helfrich, 1973* failed to describe the resulting height fluctuation spectrum. However, the extended theory by *Hamm and Kozlov, 2000*, which also includes the energetic cost of lipid tilt, successfully

captured bolalipid fluctuations (Appendix 1–Section 16). In this case, the resulting height spectrum of the membrane at vanishing membrane tension is given by *May et al., 2007*

$$\langle |h_n|^2 \rangle = \frac{k_B T}{L^2} \left( \frac{1}{\kappa q^4} + \frac{1}{k_\theta q^2} \right)$$
$$= \frac{k_B T}{L^2} \frac{1}{\kappa q^4} \left( 1 + q^2 l_\theta^2 \right) ,$$

(1)

where $q = 2\pi n/L$ is the wave number ($n \in \mathbb{Z}$), $L$ is the box size, $\kappa$ is the bending rigidity of the membrane, $\kappa_\theta$ is the tilt modulus and $l_\theta = \sqrt{\kappa/\kappa_\theta}$ is a characteristic length scale related to tilt, which we call the tilt persistence length. Considering *Equation 1*, the tilt term is expected to matter if the analysed inverse wave numbers become of the same order as the tilt persistence length $l_\theta$. The wave numbers that we analysed correspond to wavelengths that are at least twice the thickness of the membrane ($q < 2\pi/(12\sigma) \approx 0.5\sigma$). Thus, the tilt term is expected to noticeably contribute if $l_\theta > 2\sigma$. For typical bilayer membranes, one finds $\kappa_\theta = 12\, k_B T\, \text{nm}^{-2}$ (*May et al., 2007*) and $\kappa = 20\, k_B T$ (*Deserno, 2015*), so $l_\theta \sim 1nm \approx 1\sigma$ and, therefore, the tilt term can be neglected as practised before (*Cooke and Deserno, 2005*). However, when the membrane rigidity increases as we expect for bolalipid membranes, the tilt persistence length $l_\theta$ increases and the tilt term in *Equation 1* becomes relevant.

By fitting the height spectrum for bolalipid membranes (and bilayer membranes for comparison) (*Appendix 1—figure 6*), we measured the bending rigidity (*Figure 2C*) and the tilt persistence length $l_\theta$ (*Figure 2C* inset) as a function of $k_{\text{bola}}$ (see Appendix 1–Section 7, *Figure 2—video 1*). With increasing bolalipid molecular rigidity $k_{\text{bola}}$, the bending rigidity $\kappa$ rose from $8k_B T$ and plateaued at $60k_B T$, showing bolalipid membranes can be very rigid while liquid. Strikingly, the increase in membrane rigidity coincided with U-shaped bolalipids vanishing from the membrane (*Figure 2B*), which confirmed the hypothesis that straight bolalipids render lipid membranes rigid. At the same time, the tilt persistence length $l_\theta$ increases with bolalipid rigidity, starting near zero, crossing the $2\sigma$ threshold at $k_{\text{bola}} = 1\, k_B T$ and plateauing at $5\sigma$. Since membrane bending and lipid tilting are two modes of membrane deformations that compete, we conclude that bilayer and flexible bolalipids molecules form flexible membranes that prefer to bend rather than to tilt, while bolalipids in straight configuration form rigid membranes that prefer to tilt rather than to bend. Taken together, bolalipid membranes made of flexible lipid molecules are as flexible as lipid bilayers, adopting U-shaped conformations, where those made of bolalipids in straight configurations are rigid.

## Interplay between membrane curvature and rigidity

Changing membrane curvature alters the area differently in the two membrane leaflets. To adapt to the area difference, we thus expect the fraction of U-shaped bolalipids to change as the membrane curvature changes. Moreover, the results of *Figure 2B and C* showed that the U-shaped bolalipid fraction and the membrane bending rigidity are correlated. As a result, we predict that the fraction of straight versus U-shaped bolalipids in a membrane will change in response to membrane bending, in a way that makes the bending rigidity of a bolalipid membrane curvature dependent.

To investigate the effect, we measured the bending rigidity of bolalipid membranes in cylindrical shapes as a function of their radii (*Harmandaris and Deserno, 2006*) (see *Figure 2D*, *Figure 2—video 2*, Appendix 1–Section 8). We first noticed that while membrane tubes made of bilayer and flexible bolalipids were stable up to small cylinder radii $R$, almost as small as the membrane thickness itself, we found that membrane made from stiffer bolalipids ($k_{\text{bola}} = 1k_B T$) ruptured well before. Strikingly, while for stiffer bolalipids ($k_{\text{bola}} = 1k_B T$) the U-shaped bolalipid fraction increased strongly over a short range of the mean membrane curvature ($H = 1/(2R)$), we only found a small change in the U-shaped bolalipid fraction of flexible bolalipid membranes ($k_{\text{bola}} = 0$) (*Figure 2E*). Consequently, since the fraction of U-shaped bolalipid molecules controls the bending rigidity of bolalipid membranes (*Figure 2B and C*), we found that there is a strong dependency of the bending rigidity $\kappa$ on the membrane mean curvature of stiffer bolalipids ($k_{\text{bola}} = 1k_B T$) (*Figure 2F*).

In an elastic material, the strain modulus holds constant and deformation is reversible. For bolalipid membranes at $k_{\text{bola}} = 1k_B T$, however, the bending modulus decreases when deformation increases, rendering bolalipid membranes hypoelastic. Fortunately, this dependency on curvature does not invalidate our fluctuation results, where the curvature is small enough that its effect on the bending modulus is negligible (Section 15). In contrast, we did not find that $\kappa$ was curvature-dependent for

bilayer or flexible bolalipid membranes ($k_{bola} = 0$). Taken together, the rigidity of bolalipid membranes is not only controlled by the molecular stiffness of their lipid constituents but also by the emerging geometry of the ensemble of lipids. Since membrane geometry and thus membrane rigidity will change upon membrane deformations, this gives rise to hypoelastic material properties.

## Gaussian rigidity of bolalipid membranes

Another important material parameter is the Gaussian bending modulus $\bar{\kappa}$, which characterizes the reshaping behaviour of fluid lipid membranes under topological changes (*Deserno, 2015*). $\bar{\kappa}$ is notoriously difficult to measure since it only becomes detectable when the membrane changes its topological state. Continuum membrane theory, combining shape stability arguments and elasticity, predicts $-\bar{\kappa}/\kappa \in [-0.5, -1]$ (*Deserno, 2015*), where the former value is expected for incompressible membranes. Indeed, most of the numbers we know for the ratio of the two bending rigidities, many of which were deduced from simulations, lie within this range (*Hu et al., 2012*). Using the same method as developed by *Hu et al., 2012*, we determined $\bar{\kappa}$ by measuring the closing efficiency of membrane patches into a sphere (Appendix 1–Section 9). At $T_{eff} = 1.2$, we obtained $\bar{\kappa} = -4.30(22)\,k_B T$ and thus a ratio of $-\bar{\kappa}/\kappa = 0.89 \pm 0.04$ for bilayer membranes, similar to what has been reported previously (*Hu et al., 2012*). For flexible bolalipid membranes, we got a slightly smaller value for $\bar{\kappa} = -5.04(37)\,k_B T$. Due to the larger bending modulus, however, flexible bolalipid membranes show a significantly smaller ratio $-\bar{\kappa}/\kappa = 0.64 \pm 0.04$ ($k_{bola} = 0$). At larger temperature ($T_{eff} = 1.3$), the ratio can be even smaller $-\bar{\kappa}/\kappa = 0.45 \pm 0.07$ (Appendix 1–Section 9). The result shows that in addition to the differences in bending rigidities, the ratio of the two bending moduli differs strongly in bilayer and bolalipid membranes.

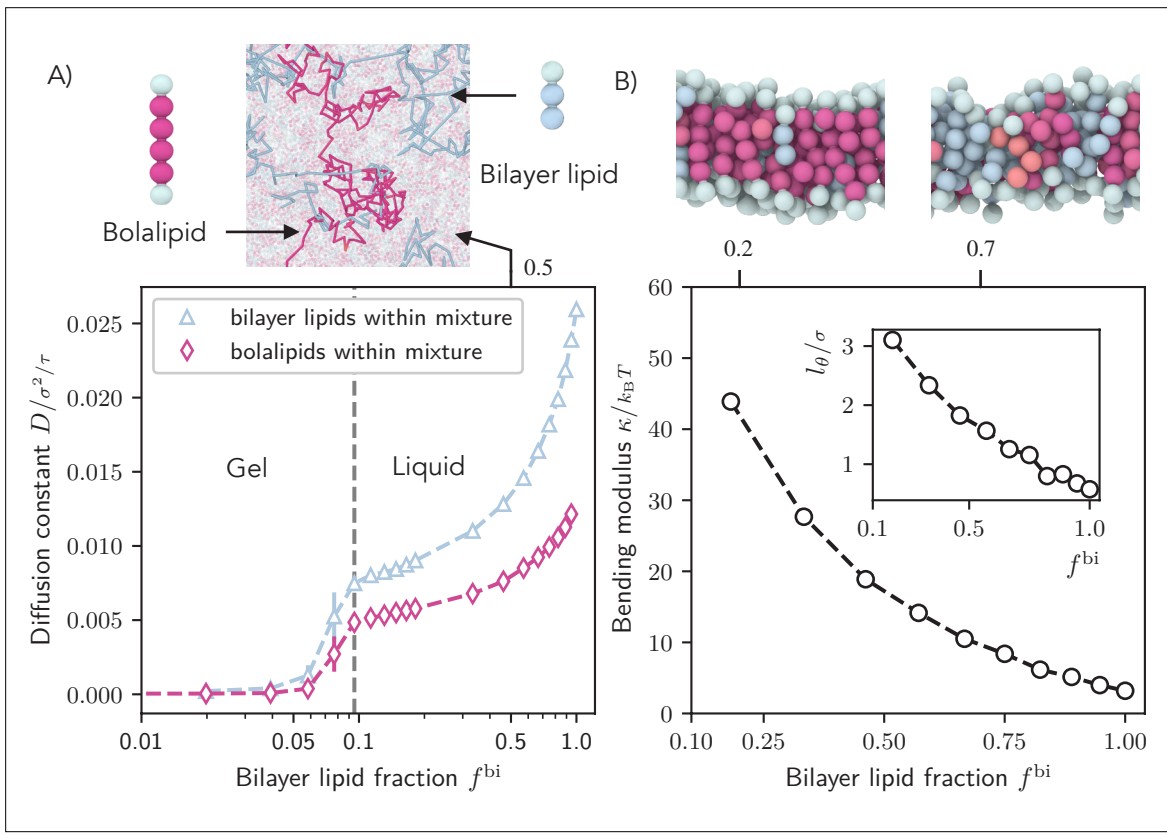

**Figure 3.** Fluidity and rigidity of mixed bilayer/bolalipid membranes. (**A**) Single lipid diffusion constant for each species as a function of bilayer fraction $f^{bi}$ ($k_{bola} = 2k_B T$, $T_{eff} = 1.3$). For $f^{bi} \geq 0.1$, the resulting mixture becomes liquid. Top: Diffusion trajectories of a bolalipid (crimson) and a bilayer lipid (blue) in a mixture membrane at $f^{bi} = 0.5$. (**B**) Bending rigidity $\kappa$ and (Inset) tilt persistence length $l_\theta$ as a function of the fraction of bilayer molecules $f^{bi}$. Top: Snapshots show bilayer lipids (blue) in mixed membranes at two different values of $f^{bi}$.

## Archaeal membranes made of mixtures of bolalipids and bilayer-forming lipids

Archaeal membranes contain varying amounts of bilayer lipids (*Oger and Cario, 2013*; *Tourte et al., 2020*). The exact bolalipid/bilayer fraction depends on the growth temperature, with higher levels of bolalipids with increasing temperature (*Kim et al., 2019*), and higher fraction of cyclopentane rings in the tails (*Chong et al., 2012*). In order to investigate the effect of different lipid contents on membrane mechanical properties, we wanted to model the archaeal membrane by mixing bilayer lipids into bolalipid membranes. Since in our model, the liquid regions of rigid bolalipid membranes and bilayer membranes do not overlap (*Figure 1F* bottom right), we picked the temperature $T_{\text{eff}} = 1.3$ to minimize fluidity mismatch and we set the molecular rigidity $k_{\text{bola}} = 2\,k_B T$ to limit U-shaped bolalipids (*Figure 2B*). We then measured the diffusion constant $D$ as a function of the fraction of bilayer lipids $f^{\text{bi}}$ (Appendix 1–Section 5). Interestingly, we found that mixing only 10% bilayer lipids into the bolalipid membrane in gel state is enough to fluidize the membrane (*Figure 3A*). We then measured the bending rigidity and the tilt persistence length $l_\theta$ of flat mixture membranes by analysing the fluctuation spectrum. Both the bending rigidity $\kappa$ (*Figure 3B*) and the tilt persistence length $l_\theta$ decreased non-linearly with the bilayer lipid fraction $f^{\text{bi}}$ (*Figure 3B* inset). Taken together, the bolalipid membrane can be substantially softened either through bolalipids acquiring the U-shaped conformation or through addition of bilayer-forming lipids.

## Curving bolalipid membranes

To investigate the response of bolalipid membranes to large membrane curvature and topology changes like those induced upon vesicle budding, which regularly occurs in archaea, we simulated membrane wrapping of an adhesive cargo bead (*Figure 4A*, *Figure 4—videos 1–3*, Appendix 1–Section 10). Importantly, this provided us with a method to study how lipid organisation is affected by externally imposed membrane curvature and mechanics. We first simulated membrane wrapping at different adhesion energies $\epsilon_{\text{mc}}$ between lipid head beads and the cargo until we observed that the membrane wrapped the cargo completely (including membrane fission). Then the minimum adhesion energy, for which a membrane bud completely enveloped the cargo bead, is the onset adhesion energy $\epsilon_{\text{mc}}^*$ (*Figure 4A*), which we measure as a function of the bolalipid stiffness $k_{\text{bola}}$ (*Figure 4B*). For small molecular stiffness $k_{\text{bola}}$, $\epsilon_{\text{mc}}^*$ first increases linearly with $k_{\text{bola}}$ before it saturates around $k_{\text{bola}} = 3\,k_B T$. We expect that the onset energy is proportional to the membrane bending rigidity $\epsilon_{\text{mc}}^* \propto \kappa$, because the bending energy to wrap a spherical particle is size-invariant (*Deserno, 2004*; *Deserno, 2015*). When we increased the bending rigidity, through increasing stiffness of bolalipid molecule $k_{\text{bola}}$, $\epsilon_{\text{mc}}^*$ increased by a factor of 3 (*Figure 4B*), suggesting that also $\kappa$ increased by a factor of 3. However, from directly measuring the membrane rigidity from the fluctuation spectrum (*Figure 2C*), we saw that $\kappa$ increased by a factor of 10. To reconcile these seemingly conflicting observations, we reason that the bending rigidity $\kappa$, similar to *Figure 2F*, is not constant but softens in the range $k_{\text{bola}} \in [0, 2]k_B T$, upon increasing membrane curvature. This is due to the dynamic change in the ratio between bolalipids in straight and U-shaped conformation. Hence, bolalipid membranes show marked hypoelastic behaviour as they soften during reshaping.

Through analysing the bolalipid conformations, we found that the membrane was able to curve by increasing the fraction of U-shaped bolalipids in the outer layer of the deformation (*Figure 4C*, Appendix 1–Section 12). To a lesser degree, we also observed this effect on the inner membrane neck (*Appendix 1—figure 14*). Remarkably, though, even when lipids are so stiff that there are no more U-shaped bolalipids in the flat mother membrane ($k_{\text{bola}} = 2\,k_B T$), the outer layer of the curved membrane retained a non-negligible fraction of $\approx 10\%$ U-shaped bolalipids, which in turn decreases $\kappa$ and softens the membrane. However, the softening effect on the membrane, indicated through a constant onset energy for $k_{\text{bola}} \geq 3\,k_B T$ (*Figure 4B*), persists even for those very stiff bolalipids. Since for stiff membranes, practically all U-shaped bolalipids are gone (*Figure 4C*), this suggested that an additional membrane-curving mechanism must be involved.

Looking more closely, at high molecular rigidity ($k_{\text{bola}} \geq 2k_B T$), we observed the formation of multiple pores on the membrane bud, which we quantified by measuring the time-averaged maximum pore diameter (*Figure 4D*, *Figure 4—video 4*, Appendix 1–Section 12). While large pores were not observed in the flat membrane, the diameter of membrane pores around the cargo was found to grow with the increase in bolalipid stiffness. We reasoned that pores form when the energetic cost

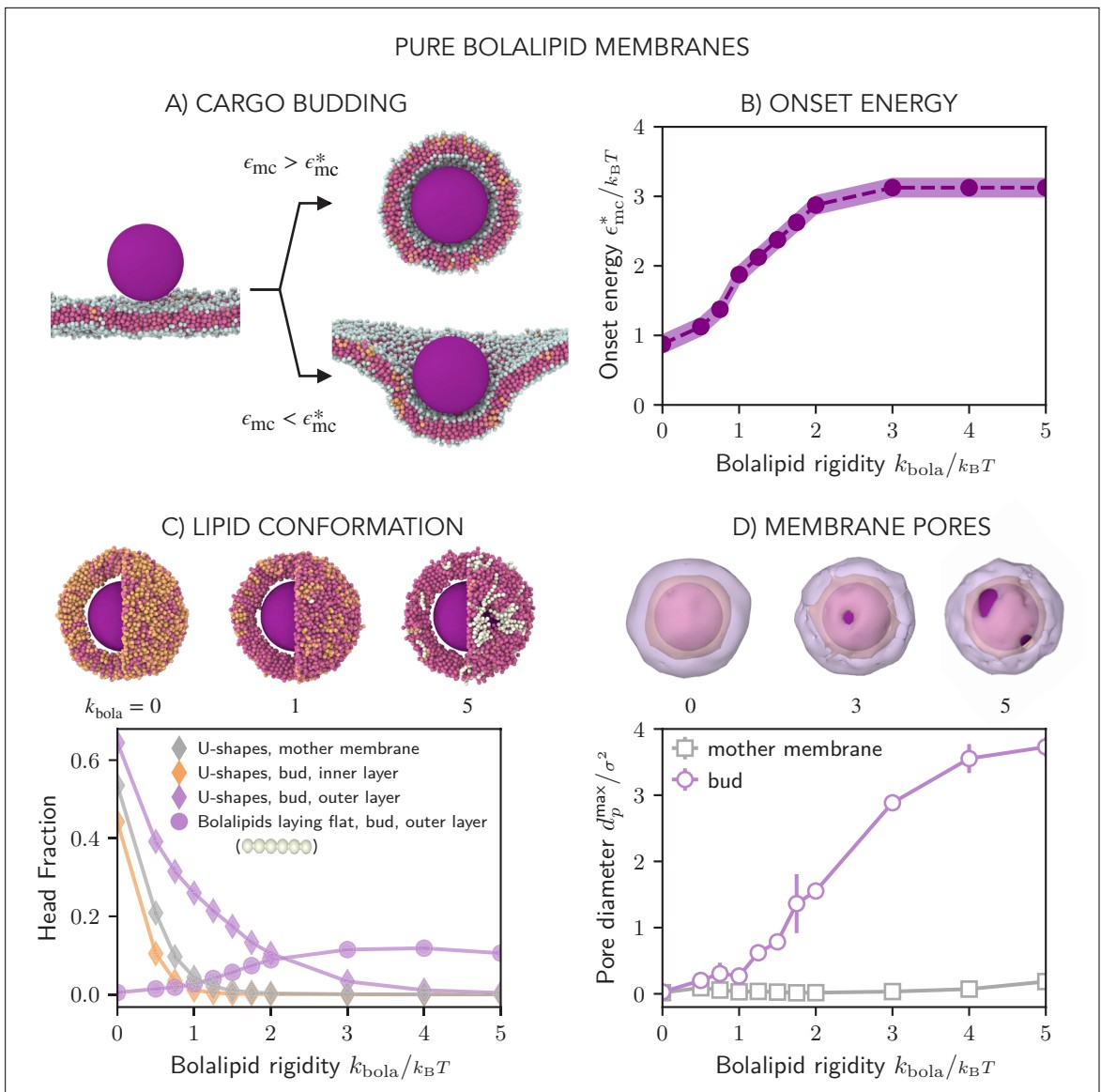

**Figure 4.** Reshaping of pure bolalipid membranes. (**A**) Simulation snapshots of the membrane wrapping a cargo bead adhering to it. Above the onset adhesion energy $\epsilon_{mc}^*$, the cargo is fully wrapped by the membrane and buds off the mother membrane. (**B**) Onset energy $\epsilon_{mc}^*$ as function of the bolalipid molecule rigidity $k_{bola}$ (for the parameters defined by the line in *Figure 2A*). (**C**) Bottom: Fraction of bolalipids in the U-shape conformation $u_f$ in the outer and inner layers of the membrane bud, and in the flat mother membrane, as a function of the bolalipid molecule rigidity $k_{bola}$. Top: Snapshots and cross-sections of the membrane around the cargo bud. At high bolalipid rigidity, the pores form around the cargo, and are lined with bolalipid molecules lying flat around the pore in a straight conformation, with both heads in the outer layer (coloured in white). The rest of bolalipids was coloured according to their head-to-head angle as before. (**D**) Bottom: Average diameter of transient pores in the membrane bud and the mother membrane as a function of the bolalipid molecule rigidity $k_{bola}$. Pores are defined as membrane openings through which a sphere of diameter $1\sigma$ can cross. Top: Snapshots of the membrane surface with outer and inner leaflet surface coloured in purple and orange, respectively, intersecting at the rim of the pore (grey).

The online version of this article includes the following video(s) for figure 4:

**Figure 4—video 1.** Successful budding, showing only a cargo-centred cross-section.
https://elifesciences.org/articles/105432/figures#fig4video1

**Figure 4—video 2.** Successful budding, showing only a cargo-centred cross-section.
https://elifesciences.org/articles/105432/figures#fig4video2

**Figure 4—video 3.** Successful budding, showing only a cargo-centred cross-section.
https://elifesciences.org/articles/105432/figures#fig4video3

*Figure 4 continued on next page*

*Figure 4 continued*

**Figure 4—video 4.** Rotating snapshot of a bud formed from a membrane made out of stiff rigid bolalipid molecules ($k_{bola} = 5\,k_BT$), showing several pores ($\epsilon_{mc} = 3.25\,k_BT$).

https://elifesciences.org/articles/105432/figures#fig4video4

required to change the bolalipid conformation to release bending stress is larger than the energetic cost of opening a lipid edge surrounding the pore. Hence, for relatively flexible bolalipids, U-shaped bolalipids provide the necessary area difference between the outer and inner layer of the membrane bud and thereby soften the membrane. For stiff bolalipid molecules, however, membrane pores start to form to enable membrane curvature as U-shaped bolalipids become prohibited. Both mechanisms help to explain the discrepancy between $\epsilon_{mc}^*$ and the bending modulus $\kappa$ obtained by studying membrane fluctuations (*Figure 2C*).

### Curving archaeal membranes

Having shown that bolalipid membranes can effectively soften also by including some amount of bilayer-forming lipid molecules, we next measured the onset energy $\epsilon_{mc}^*$ for cargo budding in the membranes formed by mixtures of bolalipids and bilayer-forming lipids, as a function of bilayer lipid head fraction $f_h^{bi}$ (*Figure 5A*, *Figure 5—video 1*). We found that the onset energy sharply decreases with increasing amount of bilayer forming lipids, and plateaus for 50% bilayer head fraction, where it acquires similar values as in the case of fully-flexible bolalipids (*Figure 4B*). For small bilayer fractions, U-shaped bolalipids localize almost exclusively on the outer layer of the bud (*Figure 5B*). As the bilayer fraction increases, there is a steady reduction in the percentage of U-shaped bolalipids in the outer layer in favour of bilayer lipids that take their role in supporting membrane curvature, with U-shaped bolalipids completely vanishing at high fractions of bilayer-forming lipids. The fraction of bilayer lipid head beads initially shows an asymmetry between the preferred outer layer and the penalized inner layer around the cargo (*Figure 5C*), but eventually approaches $f_h^{bi}$ in both layers. Taken together, as the bilayer lipid fraction increases, the role of U-shaped bolalipids in making up the asymmetry between the outer and inner layer is taken over by bilayer lipids. Curiously, the addition of bilayer lipids promotes the formation of U-shaped lipids, both in the flat membrane and even in the inner layer around the bud. This is likely to be explained by the fact that when more bilayer lipids are incorporated, the membrane is less densely packed (*Appendix 1—figure 12*) and thus U-shaped bolalipids are promoted.

Importantly, we observed nearly no pores in the membrane bud in mixed membranes, even when we only had a very small fraction of bilayer-forming lipids (*Figure 5D*). Only as the bilayer fraction increased, did we observe the formation of very small pores in the bud. For the flat mother membrane, however, membrane pores started to form with increasing values of $f^{bi}$. They acquired sizes similar to those obtained around the bud in pure bolalipid membranes (*Figure 4D*). The pore formation in the flat mother membrane is likely promoted because the membrane becomes destabilized by the increasing proportion of bilayer lipids which are close to the gas phase. Taken together, bolalipids can bud porelessly when bilayer-forming lipids, which cause membrane softening, are included.

## Discussion

All biological cells are enclosed in fluid lipid membranes. These must be both sturdy and flexible to contain the cellular interior and allow membrane reshaping during division and vesicle formation. Across nature, mono- and bilayer membranes, composed of bolalipids and bilayer lipids, have evolved to fulfil these partially conflicting requirements. In this work, we have developed the first minimal model that allows us to transparently compare the distinct behaviour of bilayer and bolalipid membranes. In our model, bolalipids are formed by joining together two bilayer lipids, with an adjustable molecular stiffness at the hinge point. Using our model, we find striking differences between bilayer and bolalipid membranes in terms of thermal and mechanical stability, bending rigidity and presence of hypoelasticity. Our results highlight how bolalipid membrane behave effectively as two-component systems, exchanging bolalipids between different conformations and thereby allowing the material properties to be adapted to support membrane reshaping.

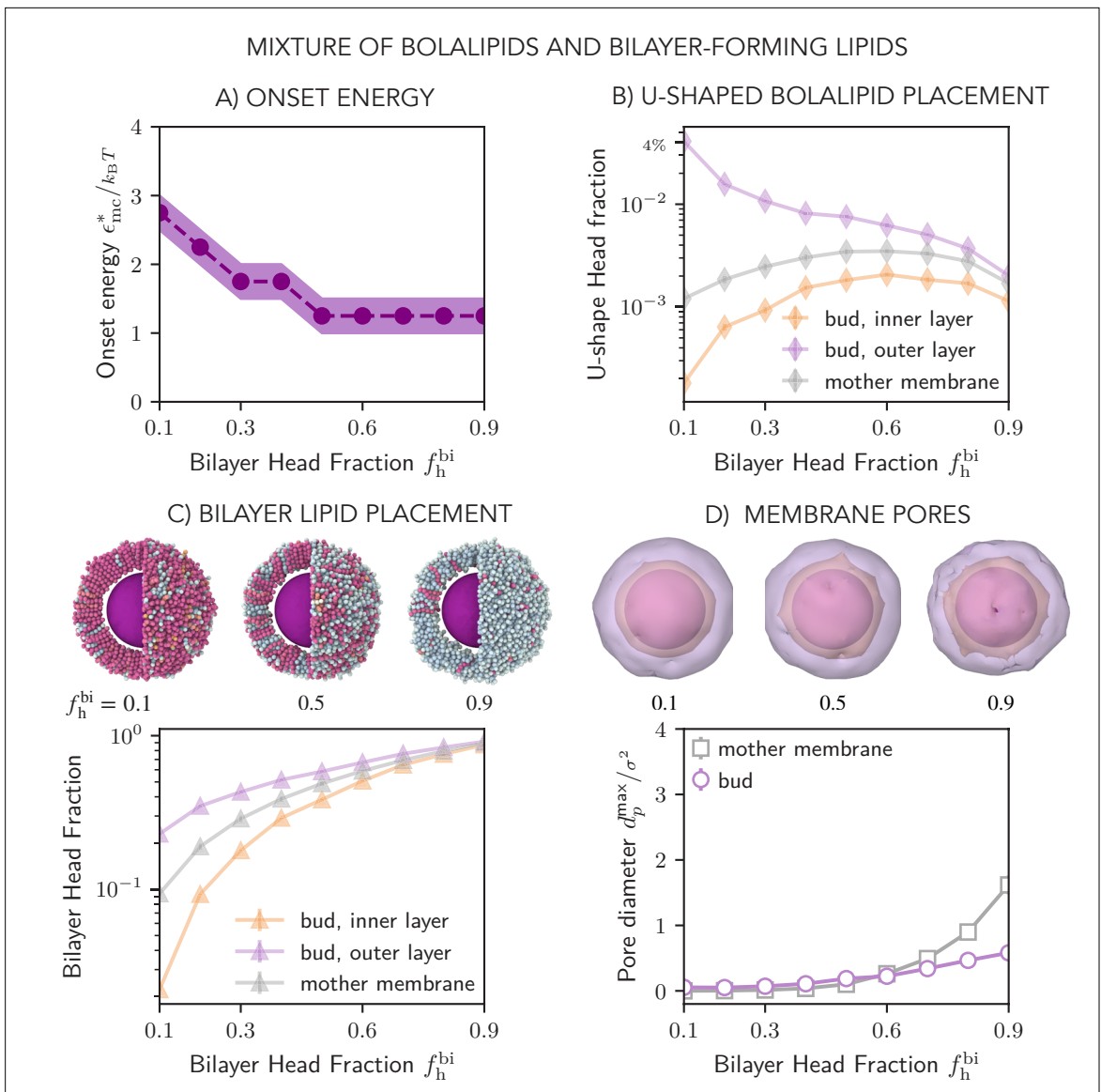

**Figure 5.** Curving of the mixed membranes, made of bilayer and bolalipid molecules. (**A**) Onset energy required to form the membrane bud, $\epsilon_{mc}^*$, as a function of bilayer head fraction $f_h^{bi}$ (for the parameters defined in **Figure 3**) (**B and C**) Fraction of U-shaped bolalipid molecules $u_f$ (**B**) and bilayer molecules (**C**) in the outer and inner layers of the membrane bud and in the flat mother membrane as a function of the bilayer head fraction $f_h^{bi}$. Top panels show the respective snapshots of membrane surface around cargo, where bilayer lipids are shown in light blue as in **Figure 4**. (**D**) Average diameter of transient pores in the membrane bud and the mother membrane as a function of bilayer head fraction $f_h^{bi}$ and respective snapshots of membrane leaflet surfaces surrounding the bud (top panel).

The online version of this article includes the following video for figure 5:

**Figure 5—video 1.** Successful budding, showing only a cargo-centred cross-section.

https://elifesciences.org/articles/105432/figures#fig5video1

While flexible bolalipid membranes are liquid under the same conditions as bilayer membranes, we found that stiff bolalipids form membranes that operate in the liquid regime at higher temperatures. These results agree well with previous molecular dynamics simulations that suggested that bolalipid membranes are more ordered and have a reduced diffusivity compared to bilayer membranes (**Bulacu et al., 2012**; **Huguet et al., 2017**). In our simulations, this is due to the fact that completely flexible bolalipids molecules adopt both straight (transmembrane) as well as the U-shaped (loop) conformation with approximately the same frequency. In contrast, stiff bolalipids typically only take on the

straight conformation when assembled in a membrane. These results agree with the previous coarse-grained molecular dynamics simulations using the MARTINI force field which showed that the ratio of straight to U-shaped bolalipids increased upon stiffening the linker between the lipid tails (*Bulacu et al., 2012*).

While previous coarse-grained simulations predicted that bolalipids spontaneously transition between the straight and U-shaped conformations (*Bulacu et al., 2012*), how this happens in archaeal membranes and whether membrane proteins are involved in this conformational transition needs to be clarified in the future. Experimental studies suggest that archaeal membranes contain flippases and scramblases for the transitioning of bilayer lipids between membrane leaflets (*Makarova et al., 2015*; *Verchère et al., 2017*), raising the possibility that similar proteins could also facilitate conformational transitions in bolalipids. In addition, it has been suggested that the viral fusion protein hemagglutinin could cause a transition from straight to U-shaped bolalipid conformation during the fusion of bolalipid vesicles with influenza viruses (*Bhattacharya et al., 2024*). However, future investigation is required.

When we determined the bending rigidity of bolalipid membranes by measuring their response to thermal fluctuations, we found that membranes made from flexible bolalipids are only slightly more rigid than bilayer membranes. This result is consistent with previous atomistic simulations, which showed that the membrane rigidity was similar for membranes composed of bilayer lipids and flexible synthetic bolalipids (*Schroeder et al., 2016*). Moreover, the result is consistent with a continuum theory which predicted that the rigidity of membranes formed of triblock copolymers is 20% larger than that of diblock copolymers (*Xu et al., 2019*). However, bolalipids in extremophilic archaea are not predicted to be fully flexible as they are expected to pack tighter due to a large number of cyclopentane rings in the lipid tails (*Chong, 2010*; *Chong et al., 2012*). Indeed, we found that membranes made of stiff bolalipid molecules can exhibit stiffness that is more than an order of magnitude larger than that of bilayer lipids at the same membrane fluidity.

It is striking that membranes made from stiffer bolalipids showed a curvature-dependent bending modulus, which is a clear signature that bolalipid membranes exhibit hypoelastic behaviour during membrane reshaping. Another marked difference between bilayer and flexible bolalipid membranes is the ratio of the Gaussian rigidity to the bending modulus. Instead of being around $-1$ as for bilayer membranes (*Hu et al., 2012*), it is around $-1/2$ and, therefore, only half of that of bilayer lipids. It is not obvious how the Gaussian bending modulus would behave upon increasing bolalipid stiffness ($k_{\mathrm{bola}} > 0$), or how to measure it due to the coupling between curvature and rigidity in bolalipid membranes. Membrane remodelling, such as the fission of one spherical vesicle into two, increases the bending energy by $8\pi\kappa$ but decreases the energy related to the Gaussian modulus by $-4\pi|\bar{\kappa}|$ (*Deserno, 2015*), giving rise to a fission energy barrier of $\Delta E_{\mathrm{fission}} = 4\pi\kappa(2 - |\bar{\kappa}|/\kappa)$. Our results indicated that while in bolalipid membranes $\kappa$ is larger, $|\bar{\kappa}|/\kappa$ is smaller compared to bilayer membranes. Our results thus predict a larger energy barrier for membrane fission $\Delta E_{\mathrm{fission}}$ in bolalipid membranes compared to bilayer membranes. It is tempting to speculate that this is one of the reasons why eukaryotes use bilayer membranes, enabling dynamic membrane remodelling and trafficking.

It is interesting to draw a parallel between monolayer membranes made of stiff bolalipid molecules and macroscopic membranes composed of rigid straight colloidal particles, which are geometrically similar, but living at different scales (*Dogic and Fraden, 2006*). It has been found that colloidal membranes at these macroscopic scales follow the standard Helfrich theory for bilayer membranes (*Barry and Dogic, 2010*), with rigidity that is three orders of magnitude higher than those of lipid bilayers (*Balchunas et al., 2019*). In this case, the tilt modulus was not pertinent, likely due to macroscopic system sizes. Similarly, we expect that the bending rigidity can be determined from membrane fluctuations independently of the tilt modulus for bolalipid membranes if they are prepared at similar relative sizes. Beyond quantitative differences, the comparison shows that monolayer membranes follow the same physics across many orders of magnitudes. However, when considering subcellular scales and the formation of high curvature, as in vesicle budding, the tilt modulus, and thus the substructure of the membrane is expected to matter.

Our model makes a number of predictions that could be tested by experiment either in cells or in vitro. First, it predicts that a small increase in the fraction of archaeal bilayer lipids should be sufficient to soften a bolalipid-rich membrane. While this could be tested in the future, so far only very few studies have yet reported experimental analysis of archaeal membrane mixtures (*Vitkova et al.,*

*2020*; *Saracco et al., 2025*). Second, we observed that membranes with moderate bolalipid molecular rigidity $k_{\text{bola}}$ exhibit curvature-dependent bending rigidity. To experimentally verify this, one could extrude membrane tethers from cells while controlling for membrane tension. Finally, to get to the core mechanism underlying our findings, it will be important to develop experimental methods that will allow the fraction of U-shaped bolalipid conformers per leaflet to be imaged and measured.

We found that membranes formed of a mixture of bilayer and bolalipids, similar to archaeal membranes, function as a composite liquid membrane that softens when adding bilayer lipids. However, while in our simulations the bending rigidity monotonically decreases with bilayer fraction, previous experiments of mixture membranes of bilayer and bolalipids with cyclopentane rings suggested that the bending rigidity non-monotonically depends on the fraction of the membrane made up of bilayer lipids (*Vitkova et al., 2020*). It remains to be determined whether the result is due to the specific lipids used, the resulting mismatch in lengths between two stacked bilayer lipids and a straight bolalipid, the experimental conditions or to non-linear effects such as the formation of lipid domains that soften the membrane with increasing bolalipid content. The same experiments reported that membranes consisting solely of bolalipids are more rigid than bilayer membranes and non-fluctuating, which is in agreement with our high bending modulus simulation results for near-pure bolalipid mixture membranes.

To investigate how bolalipid membranes respond to changing membrane curvature, we performed simulations in which small cargo particles budded from flat membranes. We found that by enforcing curvature on bolalipid membranes, the fraction of U-shaped bolalipids increased around the cargo bud, especially in the outer membrane layer and hence softened the membrane. As another mechanism to release curvature stress, we observed the formation of membrane pores, which could be mended by adding small amounts of bilayer lipids, similar to the mixture membranes that are found in archaea (*Tourte et al., 2020*). Our results suggest that enforcing membrane bending can soften bolalipid membranes locally by increasing the number of U-shapes, rendering the membrane a mechanically switchable material where large curvature decreases stiffness.

Taken together, our results show how membranes which are mixtures of bilayer and bolalipids maintain cell integrity at high temperatures, while also undergoing leak-free membrane bending. This suggests that archaeal membranes can balance opposing needs when adapting to extreme environmental conditions. Beyond understanding membrane properties and reshaping across the tree of life, these results pave the way for synthetic bolalipid membranes and bolalipid membrane containers with new and exciting material properties.

## Acknowledgements

MA, BB, and AŠ acknowledge funding by the Volkswagen Foundation Grant Az 96727. FF acknowledges financial support by the NOMIS foundation. AŠ acknowledges funding by ERC Starting Grant 'NEPA' 802960. We thank Claudia Flandoli for her help with illustrations.

## Additional information

### Funding

| Funder | Grant reference number | Author |
|---|---|---|
| Volkswagen Foundation | Grant Az 96727 | Miguel Amaral<br>Buzz Baum<br>Anđela Šarić |
| NOMIS Stiftung | | Felix Frey |
| European Research Council | Starting Grant "NEPA" 802960 | Anđela Šarić |

The funders had no role in study design, data collection and interpretation, or the decision to submit the work for publication.

## Author contributions
Miguel Amaral, Data curation, Software, Formal analysis, Investigation, Visualization, Writing – original draft, Writing – review and editing, Performed computer simulations and analysis; Felix Frey, Data curation, Software, Formal analysis, Investigation, Writing – original draft, Writing – review and editing, Performed patch closing simulations, contributed to computer simulations, and performed theoretical analysis; Xiuyun Jiang, Software, Investigation; Buzz Baum, Conceptualization, Supervision, Funding acquisition, Writing – review and editing; Anđela Šarić, Conceptualization, Resources, Data curation, Supervision, Funding acquisition, Writing – original draft, Project administration, Writing – review and editing

## Author ORCIDs
Miguel Amaral ⓘ https://orcid.org/0000-0001-5755-0001
Felix Frey ⓘ https://orcid.org/0000-0001-8501-6017
Buzz Baum ⓘ https://orcid.org/0000-0002-9201-6186
Anđela Šarić ⓘ https://orcid.org/0000-0002-7854-2139

Reviewer #2 (Public review): https://doi.org/10.7554/eLife.105432.3.sa1
Reviewer #3 (Public review): https://doi.org/10.7554/eLife.105432.3.sa2
Author response https://doi.org/10.7554/eLife.105432.3.sa3

---

# Additional files

## Supplementary files
MDAR checklist

## Data availability
The simulation input files and codes are freely available at https://doi.org/10.5281/zenodo.13934991 (*Amaral, 2024*).

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

## Appendix 1

## 1 Membrane model

To study the topological difference between bilayer lipids and bolalipids, we started from a coarse-grained model that was previously developed for lipid bilayer membranes by *Cooke and Deserno, 2005*. Each lipid is composed of a chain of beads. Bilayer lipids are composed of one head bead connected to a chain of two tail beads. In contrast, bolalipids are composed of two head beads linked to a chain of four tail beads (see *Figure 1C*). Importantly, bilayer and bolalipids have identical head groups.

Each adjacent pair of beads in a lipid is connected by a finite extensible non-linear elastic bond (FENE). For a given bond length $r$, its potential is the sum of an attractive term and a purely repulsive Lennard-Jones potential that enforces volume exclusion

$$U_{\text{bond}}(r) = -\frac{1}{2}KR_0^2 \ln\left(1 - \left(\frac{r}{R_0}\right)^2\right) + U_{\text{lj}}(r), \quad r \in [0, R_0], \tag{A1}$$

with $K = 30k_{\text{B}}T/\sigma^2$ and maximum length $R_0 = 1.5\sigma$ in the first term. We note that in the simulations, our time, distance, and energy units are, respectively, $\tau$, $\sigma$, and $k_{\text{B}}T$, and the Boltzmann constant $k_B = 1$. Consequently, our unit of mass is given by $1m = 1k_{\text{B}}T\tau^2/\sigma^2$. The second term of *Equation A1* is given by

$$U_{\text{lj}}(r) = U_{\text{m}} \cdot \left(x^{-12} - 2x^{-6} + 1\right), \tag{A2}$$

where $x = \min(r, r_c)/r_m$, where $r_m$ is where the potential reaches its minimum value $U_m$ and $r_c$ is its cutoff. For bonded beads, we set $U_m = 1k_{\text{B}}T$, $r_{\text{m}} = r_c = 2^{1/6}\sigma$, so that the repulsive part is zero for $r > r_c$.

For non-bonded beads, as a first term, we pick also a purely repulsive form with $r_{\text{m}} = r_c = 2^{1/6}\sigma \approx 1.12\sigma$, and parametrise on its strength by scanning the interval $U_{\text{m}} = \epsilon_{\text{p}} \in [0.5, 2]k_{\text{B}}T$. The inverse of this potential depth is the effective temperature $T_{\text{eff}} = k_{\text{B}}T/\epsilon_{\text{p}}$ in our system. Following *Cooke and Deserno, 2005*, we scaled down $r_{\text{m}}, r_c$ by 0.95 for interactions with head beads, to ensure no spontaneous curvature in a membrane leaflet (*Cooke and Deserno, 2005*; *Hu et al., 2012*). Accordingly, we note that in our snapshots, each head bead is represented as a sphere of diameter $0.95\sigma$ and each tail bead as a bead of diameter $1\sigma$.

To model lipid rigidity between each consecutive three beads $b_1, b_2, b_3$ in a lipid, we set a harmonic angle potential

$$U_{\text{angle}}(\theta) = K \cdot (\theta - \pi)^2, \quad \theta \in [0, \pi], \tag{A3}$$

with $\theta = \widehat{b_1, b_2, b_3}$. If one of the beads is a head bead (i.e. in a bilayer lipid), we set $K = k_0 = 5\,k_BT$. In contrast, for the two inner angles in a bolalipid, we set the interval $K = k_{\text{bola}}$, where $k_{\text{bola}}$ is a global constant in the simulation, equal for all bolalipids. In our work, for different simulations, we vary it in the interval $[0, 5]k_{\text{B}}T$. Therefore, the difference between a bolalipid and two bilayer-forming lipid molecules is the extra bond and two equal angle potentials keeping each three beads connected and aligned, respectively.

To model the (implicit) hydrophobic interaction between lipid tail beads, we add a longer range attractive cosine squared potential

$$U_{\text{cs}}(r) = -\cos^2\left(\frac{\pi}{2}\text{clip}\left(\frac{r - r_c}{\omega}, 0, 1\right)\right)$$

$$= \begin{cases} - & r \leq r_c \\ -\cos^2\left(\frac{\pi}{2}\frac{r - r_c}{\omega}\right) & r_c < r < r_c + \omega \\ 0 & r \geq r_c + \omega \end{cases} \tag{A4}$$

where $\text{clip}(x, a, b) = \max(\min(x, b), a)$ and $r_c$ is set to the cutoff $2^{1/6}\sigma$ of the volume exclusion interaction. The repulsive Lennard-Jones and the attractive potential are combined to make the tail-tail interaction repulsive in the range $[0, r_c]$ and attractive in the range $[r_c, r_c + \omega]$. We take $\omega$, the

attractive range width, as a parameter. As we joined two bilayer lipids to form a bolalipid, in our model bilayer lipids and bolalipids share the same hydrophobic interaction.

## 2 Simulation protocol for membrane self-assembly

We initially verified self-assembly of our lipids into a flat membrane: we placed lipids in a dispersed gas configuration in a periodic 3D cube. We then evolved the system with timestep $\delta t = 0.01$ under a Langevin thermostat with relaxation time of $1\tau$ and checked a flat membrane patch eventually formed (*Figure 1—video 1*).

For the rest of our simulations, we pre-assembled the membrane. This was done by placing lipids far from each other in a hexagonal grid, locking the lipids vertical position and conformation, compressing with volume exclusion until a target area per lipid is reached, followed by energy minimisation and a final check that the lipids formed a horizontal single cluster.

## 3 Bolalipid conformation in flat membranes

For simulating flat membrane patches of bolalipids, we combined the previously used Langevin thermostat with relaxation time of $1\tau$ with a Nosé–Hoover barostat with relaxation time of $10\tau$. In LAMMPS, this amounts to combining the commands 'fix langevin' with 'fix nph.' We configured the barostat to set lateral pressure $P_{xy}$ to zero by re-scaling the simulation box in the $x$-$y$ plane. We compare this setup to a fixed box length setup, and a NPT ensemble setup, in Appendix 1–Section 17.

For a bolalipid in a flat membrane, we found thresholding on the angle between its first and second half $\theta = \angle \overrightarrow{b_3, b_1}, \overrightarrow{b_4, b_6}$ is sufficient to classify its conformation (*Figure 1E*, right).

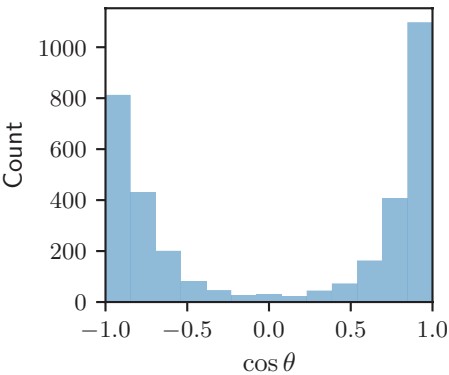

**Appendix 1—figure 1.** Histogram of the cosine of the angle $\theta$ between halves of a bolalipid ($k_{\text{bola}} = 0$) for the last frame of the simulation of a flat membrane of flexible bolalipids.

We checked for a few parameters that this angle follows a bimodal distribution (*Appendix 1—figure 1*) and that indeed bolalipids assume either a straight conformation, one head bead in opposing membrane leaflets, with $\theta \approx \pi$ or a U-shaped conformation, both head beads on the same leaflet, with $\theta \approx 0$. Accordingly, we marked a bolalipid as being in the U-shape if $\theta < \pi/2$ and in the straight conformation otherwise (see *Figure 1E* for a snapshot).

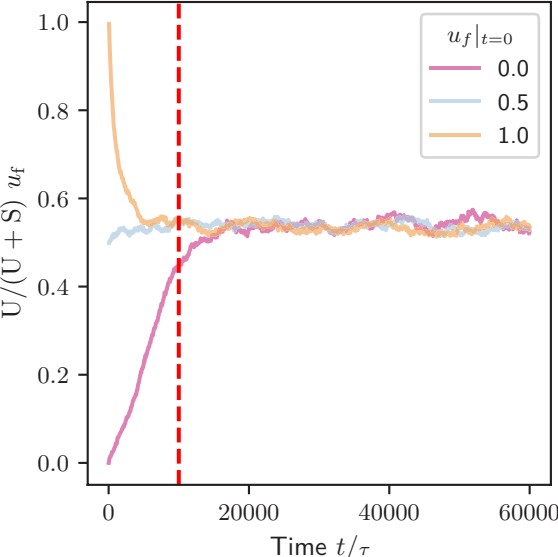

**Appendix 1—figure 2.** Time series of $u_f$, the fraction of bolalipids in the U-shaped conformation, for the simulations done for flexible bolalipids ($k_{\text{bola}} = 0$), where the system was pre-assembled with three different initial values $u_f|_{t=0}$. The red dashed line marks the typical equilibration time of $10^4\tau$.

For pre-assembly of flat membranes containing bolalipids, this implied a choice of which conformation to pick for each lipid, straight or U-shaped. We verified that for the limiting cases of flexible and rigid bolalipids, starting the system with all bolalipids in the straight and U conformation, respectively, resulted in the U-shaped bolalipid fraction $u_f$ equilibrating after simulating at most for $10^4\tau$ (*Appendix 1—figure 2*).

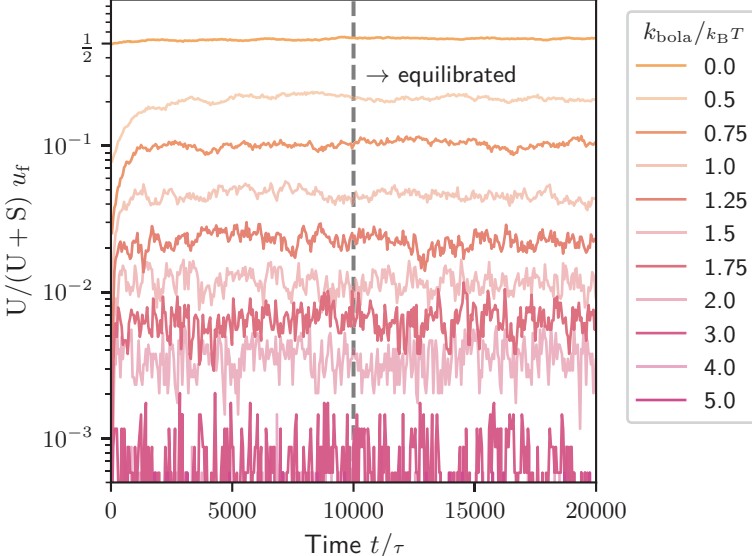

**Appendix 1—figure 3.** Time series of fraction of bolalipids in the U-shaped conformation $u_f$, for simulations of pure bolalipid membranes with different values of the bolalipid rigidity $k_{\text{bola}}$. The dashed line marks the timestamp used for defining the system as equilibrated. All equilibrium measures are taken from frames after this point.

After verifying this, we started our simulations with a fraction of bolalipids derived from assuming a two-state system, with $\Delta E$ given solely by the central angle potentials (e.g. for U-shaped bolalipids, both angles assume values of $\pi/2$). We checked our system had equilibrated quantitatively by observing that the time series of the fraction of bolalipids in the U-shaped conformation approached a steady state (*Appendix 1—figure 3*). We still equilibrated for $10^4\tau$ before taking measures.

## 4 Simulation protocol for membrane phase determination

For phase determination (*Figure 1F*), the membrane was assembled in a horizontal grid of lipids with approximately $25^2$ head beads in each leaflet. For bolalipid membranes, we set the initial fraction of U-shaped bolalipids as we did for simulations depicted in *Appendix 1—figure 3*. We then ran the simulation, halting it automatically if the box size diverged, at which point we could consider the membrane as being in the gas phase. Otherwise, the simulation ran to completion for $\Delta_{\text{total}} = 10 \times 10^3 \tau$. This was determined to be sufficient for both the box size $L$ and the relative amounts of lipid species and conformations to equilibrate.

## 5 Diffusion constant

For computing the diffusion constant $D$, we first found the corresponding mean square displacement for each $\Delta t$ and then fitted $D = \left\langle |\Delta x|^2 \right\rangle / (4\Delta t)$. To compute the M.S.D. of a specific lipid in a temporary conformation, we averaged $|\Delta x|$ over all possible intervals of duration $\Delta t$ where the conformation was held. We took care to exclude displacements of lipids floating in the gas phase of the simulation. *Appendix 1—figure 4* shows $D$ as a function of temperature $T_{\text{eff}} = k_{\text{B}}T/\epsilon_{\text{p}}$ for the different membrane types. *Appendix 1—figure 4* shows a discontinuity of the diffusion constant $D$ as a function of $T_{\text{eff}}$. We took this discontinuity as marking the transition of the membrane from the gel phase to the liquid phase, and determined it for bilayer and bolalipid membranes for different values of $k_{\text{bola}}$ (*Figure 4*). For all membranes and parameters tested, this is equivalent to setting the minimum diffusion constant for a liquid membrane to $D = 5 \times 10^{-4} \sigma^2/\tau$.

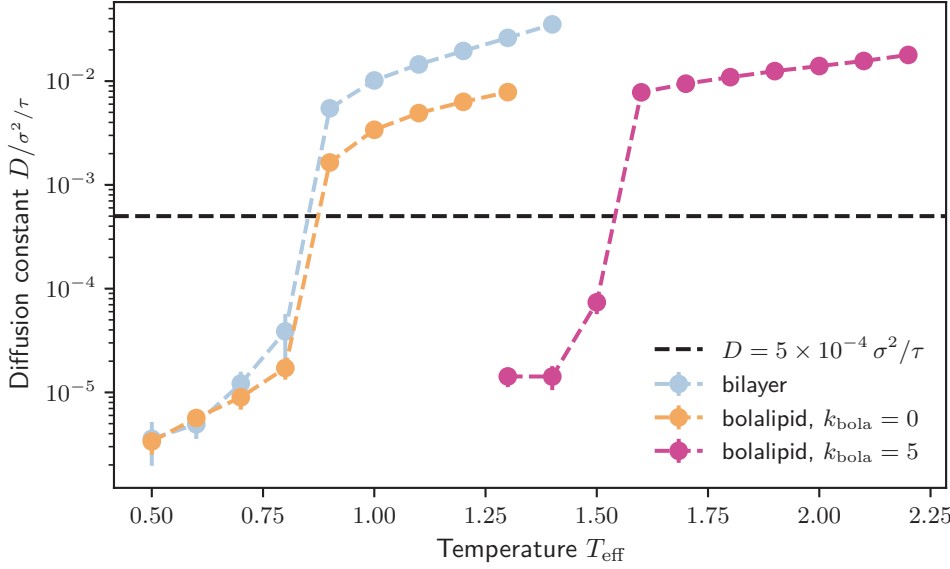

**Appendix 1—figure 4.** Diffusion constant for different pure membranes at $w = 1.5\sigma$ versus temperature $T_{\text{eff}}$. The discontinuity (jump) in $D$ marks the transition from the gel phase to the liquid phase. The dashed black line marks the minimum diffusion constant that is required to classify as a liquid membrane at $D = 5 \times 10^{-4} \sigma^2/\tau$.

## 6 Two-state model for bolalipid conformation

We judged our two-state model fitness in *Figure 2B* by first making the model linear, expressing it as $\log\left(1/u_{\text{f}} - 1\right) = x_1 k_{\text{bola}} + x_0$; then we restricted it to points where $k_{\text{bola}} \leq 2 k_B T$, or, equivalently, when there were on average $\approx 2$ or more U-shaped bolalipids; with this choices, we obtained $R^2 = 0.99$.

By splitting the average potential energy between an internal contribution (bonds, angles, and pair interactions between particles in the same molecule) and an external contribution (pair interactions between a molecule and its neighbours), we determined the transition energy from straight to U-shaped bolalipids in detail. We found that this transition lowers the internal potential energy of the bolalipid while increasing its interaction energy. In total, we obtained an energy barrier for the transition of $\Delta E_{\text{s}\rightarrow\text{u}} = 0.79(1) k_B T$. Since the fit indicates, however, that the U-shaped bolalipid conformation is preferred over the straight conformation, we conclude that there must be either an entropic contribution to the free energy or an intermolecular interaction energy favouring U-shaped bolalipids.

## 7 Fluctuation spectrum

In order to measure height fluctuation spectrums (*Figures 2C and 3B*), we used membranes with $60^2$ head beads per leaflet, with minimum $\Delta_{eq}$ set to $20 \times 10^3 \tau$; total runtime was $\Delta_{total} = 60 \times 10^3 \tau$.

For measuring the bending modulus $\kappa$, we follow the analysis done by *Cooke and Deserno, 2005*. We simulate a horizontal membrane in a periodic simulation box of dimensions $(l_x, l_y, l_z)$, that is horizontally square with $l_x = l_y = L$. Importantly, we keep membrane tension to a minimum by setting the lateral pressure $P_{xx} = P_{yy} = 0$ via a barostat. As a first equilibration check, we consider the lateral box size $L$ time series. Starting by considering the full series, we measure how much the first and last half differ. For each half, we compute the maximum, minimum and the diameter (max - min). If the relative difference is less than 30%, we consider it equilibrated. Otherwise, we exclude the first frame of the time series and repeat the check.

For each of the $N_f$ remaining frames, we intend to obtain the height field of the membrane $h(x, y)$ within the $x$-$y$ plane. However, our simulations are particle-based, and hence our system is discrete. Therefore, we divide the horizontal plane in a regular $m \times m$ grid, where $m$ is the length of a grid cell. We set $m = 40$ by trial and error so that no bins will be empty. We compute $h_b$, the average height of bin $b$. We then apply the 2D discrete Fourier transform in the $x$-$y$ plane, obtaining complex components $h_n$ for $n \in [-m/2, ..., 0, .., m/2] \times [-m/2, ..., 0, .., m/2]$, with $h_n = h_{-n}$.

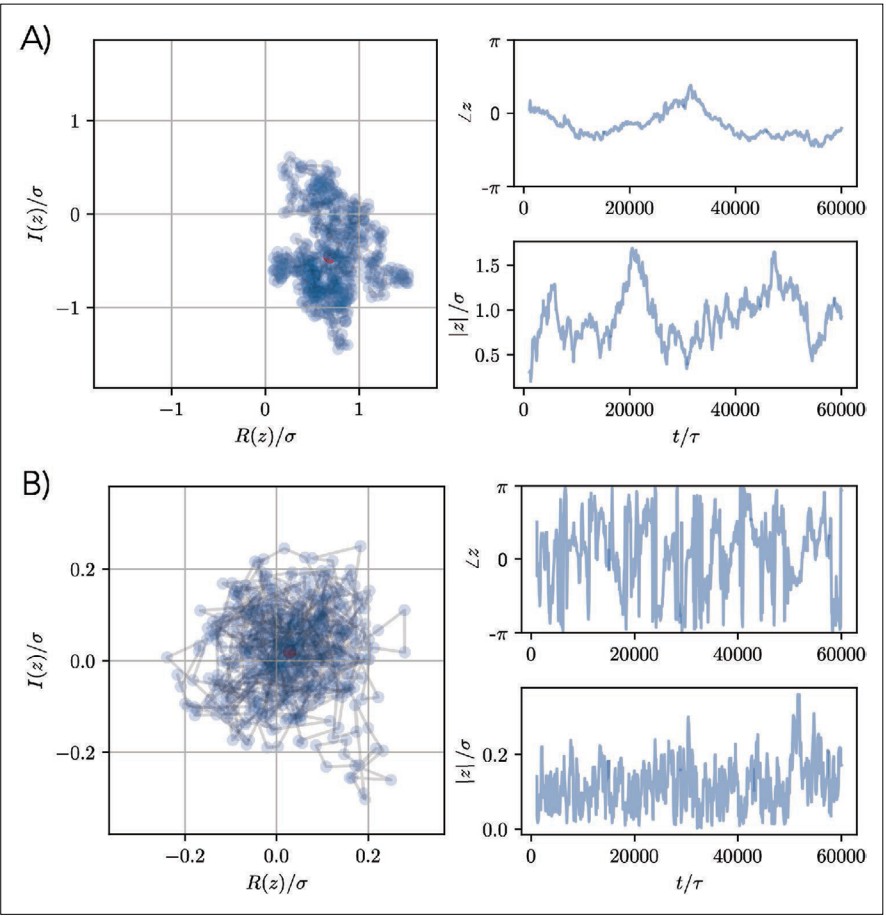

**Appendix 1—figure 5.** Path of the complex component of the Fourier transform of membrane height at $n = (1, 0)$ (**A**) and $n = (2, 0)$ (**B**) for bolalipid pure membrane at $k_{bola} = 0.5 k_B T$. Left plots show the trajectory in the complex plane, while on the right, we plot their phase and norm versus time. The mode in (**A**), with an autocorrelation time of roughly $10^4 \tau$, has only 10 uncorrelated points. On the other hand, the mode in (**B**), with an autocorrelation time of approximately $10^3 \tau$, has $\approx 60$ uncorrelated samples and thus crosses the chosen threshold of 20 samples for being considered equilibrated.

We correct for the binning by multiplying by $\text{sinc}(n_x/m)\,\text{sinc}(n_y/n)$, which mostly affects the smallest wavelengths (*Cooke and Deserno, 2005*); we also scale by $\langle L \rangle$ obtaining $u = h\langle L \rangle$. For each mode number vector $n = (n_x, n_y)$ of $u$, we compute its amplitude squared $|u_n|^2$ and phase $\angle u_n$ (examples in *Appendix 1—figure 5*). For both the amplitude and phase, we then compute the autocorrelation time $\tau_c$ in order to get the statistical inefficiency $g = 2\tau_c + 1$. The phase should regularly jump through the endpoints $[0, \pi]$, so checking it too for stationarity allows to exclude modes whose amplitude is static but which have their phase stuck. By doing this, we find that we can improve the analysis compared to the current state of the art (*Ergüder and Deserno, 2021*), which only checks the amplitude squared. Between both the amplitude squared and phase components, we take the largest $g$, which then gives us the number of uncorrelated data points as $N_f/g$. We accept a mode as equilibrated if the remaining trajectory contains at least 20 uncorrelated data points. The standard deviation of the mean of $|u_n|^2$ must then be scaled by $\sqrt{g}$.

For each spectrum measurement, we performed four simulations with different seeds for the thermal noise. We retained only modes which had equilibrated on all replicas and averaged over modes with the same mode number. We checked that the box size $L$ varied less than 1% between replicas. To not impose an artificial variable window in mode number, we computed the first equilibrated mode for all our simulations and took the maximum equilibrated mode number as a global minimum threshold. We found in our simulations $n \geq 2$. This roughly corresponds to a maximum wavelength cutoff of $32\sigma$ given that our simulation boxes have length $60\sigma$. We used twice the membrane thickness, i.e., $12\sigma$ as minimum wavelength cutoff.

According to continuum membrane theory, at zero membrane tension, the fluctuation spectrum is given by *Seifert, 1997*

$$\langle |h_n|^2 \rangle = \frac{k_B T}{L^2 \kappa q^4}, \tag{A5}$$

where $\kappa$ is the bending modulus, $L$ is the box size, $q = 2\pi n/L$ is the (angular) wave number and $n$ is the mode number (*Cooke and Deserno, 2005*). While this was adequate for bilayer membranes, we needed to add the tilt term, parametrized by the tilt modulus $\kappa_\theta$ (see main text *Equation 1*, *May et al., 2007*) to get reasonable fits for rigid bolalipid membranes. Then, we obtained

$$\langle |h_n|^2 \rangle = \frac{k_B T}{L^2} \left( \frac{1}{\kappa q^4} + \frac{1}{k_\theta q^2} \right). \tag{A6}$$

We fitted the data using $N$ measurements of mean and mean standard deviation $(y_i, \sigma_i, x_i)$, with $y = \langle |u_n|^2 \rangle = \langle L \rangle^2 \langle |h_n|^2 \rangle$ and $x = q = 2\pi|n|/\langle L \rangle$ to each model $f(x_i) = y_i$. In this case, the different models $f(x)$ are given by *Equation A5* and *Equation A6*, where *Equation A5* can be derived from *Equation A6* by formally setting $\kappa_\theta = \infty$. Typical example fits are shown in *Appendix 1—figure 6*.

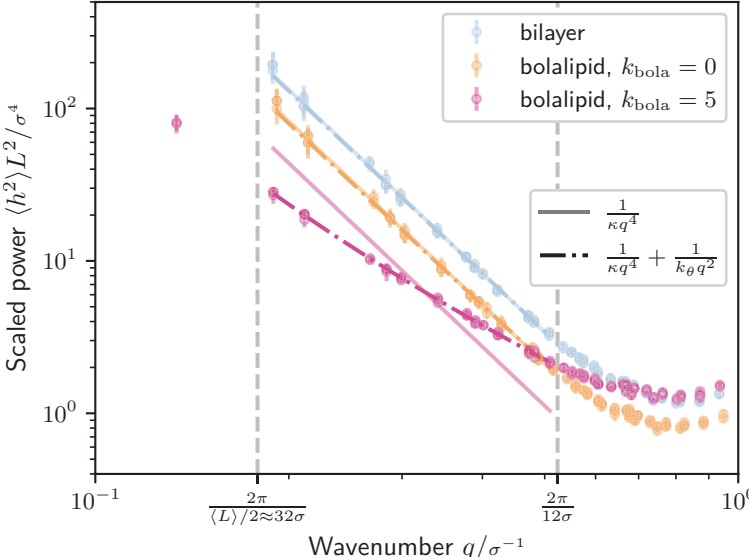

**Appendix 1—figure 6.** Fluctuation spectra and corresponding fits according to *Equation A6* for a model without and with tilt modulus, for a bilayer membrane at $T_{\text{eff}} = 1.2\,k_B T$, a flexible bolalipid membrane ($k_{\text{bola}} = 0$) and a rigid bolalipid membrane ($k_{\text{bola}} = 5\,k_B T$). In vertical dashed lines, we marked the interval of wave numbers that selects the modes used for fitting. The first rigid bolalipid equilibrated mode is to the left of this interval and thus excluded from the corresponding fit.

We used the reduced $R^2$ value as an indicator of goodness of fit, defined as $R^2 = \sum_i ((f(x_i) - y_i)/\sigma_i)^2 / N$.

Reasonable values were recognized as $R^2 \leq 1$. $R^2$ is plotted together with fit results for pure bolalipid membranes and bolalipid/bilayer mixture membranes in *Appendix 1—figure 7*. In the first row, we show the results of the fit of *Equation A5* while in the second row, we used *Equation A6*. In general, for small values of bolalipid rigidity or large bilayer fraction, the fits without the tilt term were still reasonable. However, as the bolalipid rigidity increased or the bilayer fraction decreased, the fits became worse. Then only fits with the tilt term were reasonable. In addition, we plot the same measures for temperature sweeps for the bilayer at $T_{\text{eff}} = 1.2$, the flexible bolalipid and the rigid bolalipid membranes in *Appendix 1—figure 8*. The results are summarized for all membrane types in *Appendix 1—figure 9*.

We omitted the error in the main text when less than the unit. Moreover, the low values of tilt modulus will necessarily be accompanied by smaller error bars since the smaller the value, the larger the influence the term will have on the amplitudes.

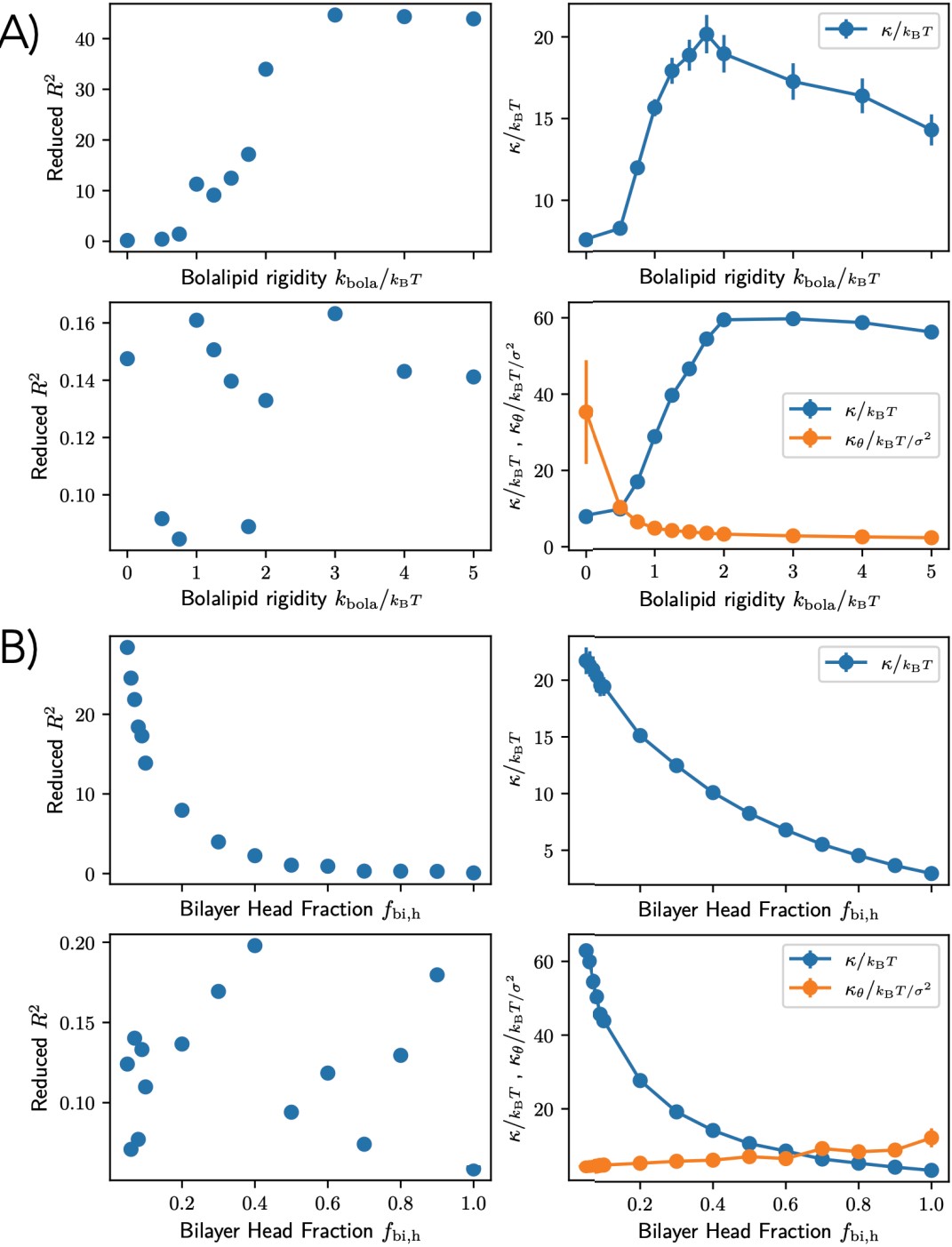

**Appendix 1—figure 7.** Fluctuation spectrum fit comparisons for pure bolalipid membranes (**A**) and bolalipid/bilayer mixture membranes (**B**). In the first row, we show the results of the fit of *Equation A5* while in the second row, we used *Equation A6*.

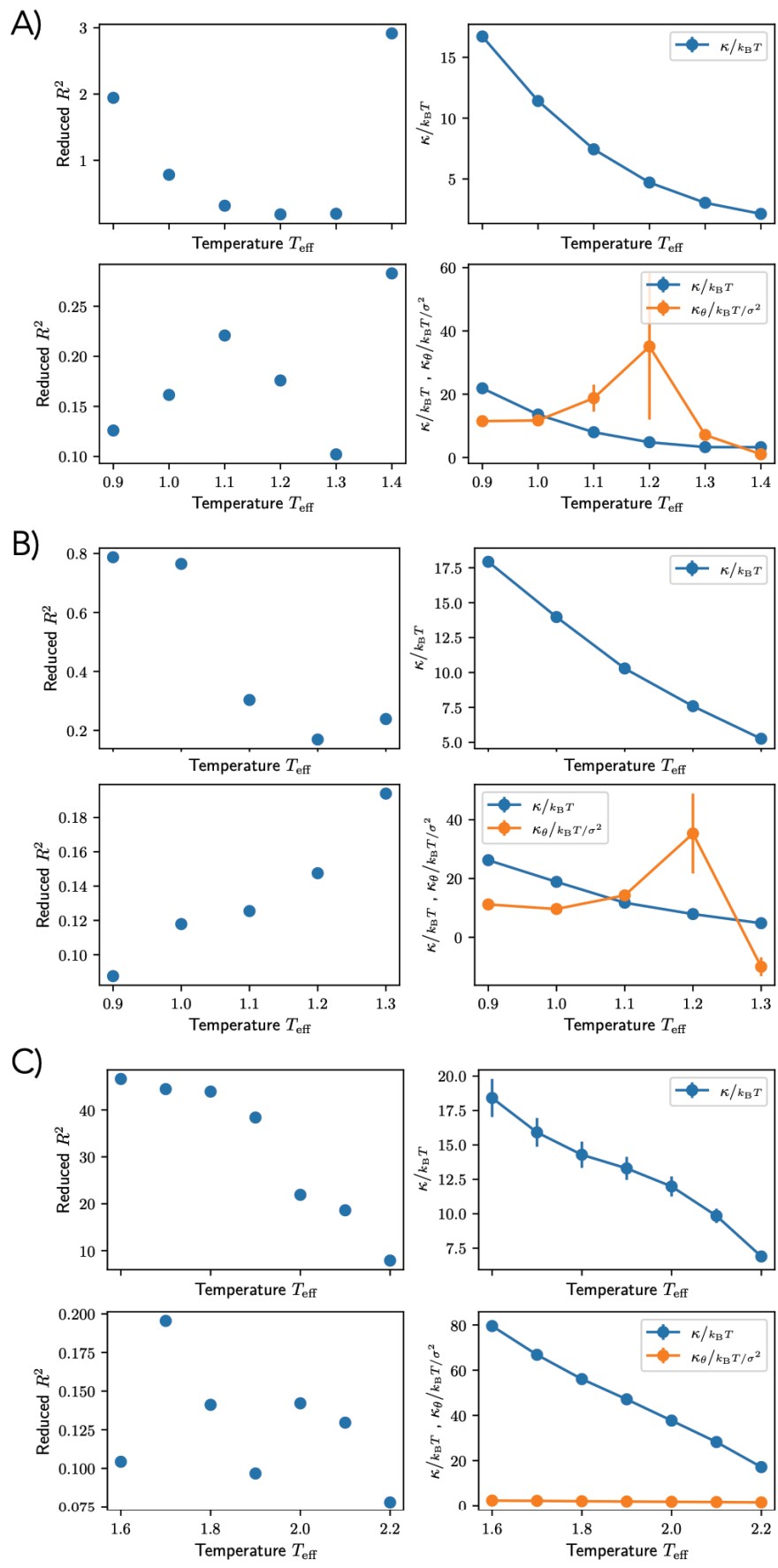

**Appendix 1—figure 8.** Fluctuation spectrum fit comparisons for pure bilayer membranes (**A**), flexible (**B**) and rigid pure bolalipid membranes (**C**). For each type of membrane, in the first row, we show the results of the fit of *Equation A5* while in the second row, we used *Equation A6*, as a function of $T_{\text{eff}}$.

We note that in our large flat membrane simulations ($L \approx 60\sigma$), flexible bolalipid membranes at $\omega = 1.5\sigma$ are stable only at temperatures smaller than $T_{\text{eff}} < 1.4$. For $T_{\text{eff}} = 1.4$ and presumably above, the membrane folds while shrinking the box until self-contact occurs. This is accompanied by massive oscillations of the box pressure, and can be avoided by halving the timestep. Since these points are near the gas phase and halving the timestep did not significantly change measurements for non-collapsing membranes (Section 17), we simply kept the timestep at 0.01. On the other hand, both the bilayer membranes and rigid bolalipid membranes of same size disassemble by pore formation, followed by simulation box expansion in response to the increased pressure, respectively at $T_{\text{eff}} = 1.5$ and 2.3. This explains why in *Appendix 1—figure 9* we have more data for bilayer at higher $T_{\text{eff}}$ than for the flexible bolalipids.

We also note that for both the bilayer and the flexible bolalipid membrane, in $T_{\text{eff}} \in [1.2, 1.3]$, the tilt modulus varies non-monotonically. It first increases, aligning with the expectation that higher pair potential temperature would reduce order and thus increase the cost of local coordinated tilting, i.e., increasing the tilt modulus $\kappa_\theta$. However, at the higher $T_{\text{eff}}$ it abnormally decreases.

Comparing the bending moduli of the flexible bolalipid membrane ($\kappa \approx 8\,k_B T$, $\kappa_\theta \approx 30(10)\,k_B T/\sigma^2$) to the bilayer membrane ($\kappa \approx 5\,k_B T$, $\kappa_\theta \approx 33(20)\,k_B T/\sigma^2$, *Appendix 1—figure 7B*), we find similar results. By systematically investigating the membrane rigidity as a function of temperature, we found that in general flexible bolalipid membranes have a slightly increased rigidity compared to bilayer membranes (*Appendix 1—figure 9*). We also verified that by increasing $T_{\text{eff}}$, a rigid bolalipid membrane softens in the same manner as bilayer and flexible bolalipid membranes (*Appendix 1—figure 9*).

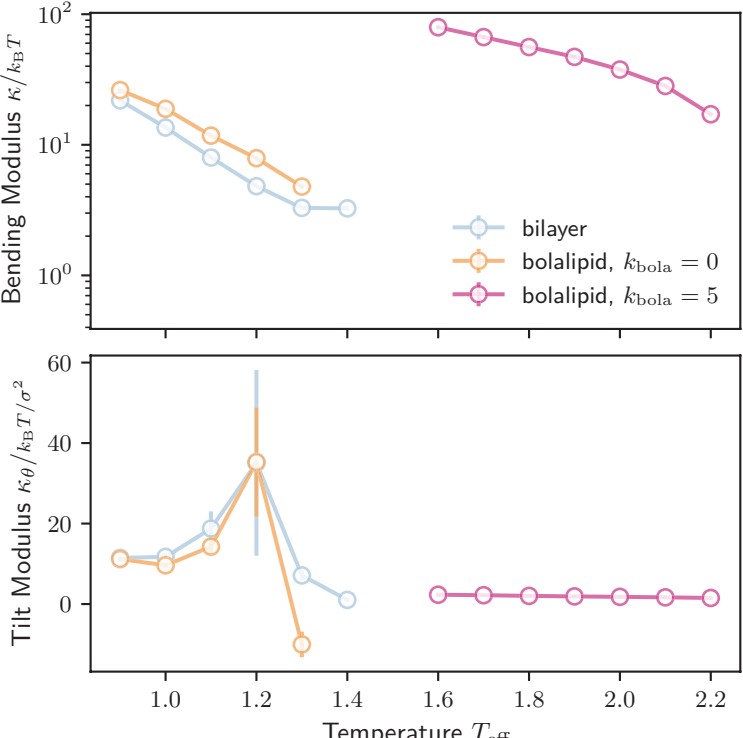

**Appendix 1—figure 9.** Fluctuation spectrum fit results, with (top) bending and (bottom) tilt modulus, for bilayer, flexible bolalipid and rigid bolalipid membranes at $w = 1.5$ as a function of $T_{\text{eff}}$.

# 8 Measuring bending modulus via membrane tether/cylinder simulations

Since the method is described elsewhere (*Harmandaris and Deserno, 2006*), here, we detail specifics. For all measurements, we ran four seeds, each over $20 \times 10^3 \tau$, integrating in the NVT ensemble using a Langevin thermostat with damping coefficient $1\tau$ and unit temperature. The membrane was assembled into a cylindrical shape, with the number of heads in each leaflet pre-balanced. The stress tensor was measured at $1\tau$ intervals. The radius was measured from trajectory frames saved every $20\tau$ in the following way. First, we excluded lipids in gas phase by clustering. Then, we computed the centre of mass of the membrane and set it as our origin for the cylinder cross-section $x$-$y$ plane. Then we computed the measured radius as $R = 1/\langle 1/r \rangle$, where the average is over beads and where $r = |x^2 + y^2|$ is the distance to the cylinder axial radius. Using $1/r$ instead of $r$ directly compensates to first order for the shell volume $2\pi r dr l_z$ dependency on $r$.

After qualitatively checking for quick equilibration of the radius, fraction of U-shaped conformers and the stress tensor components, we dispensed with lengthy equilibration, discarding only the first $1000\tau$ of the trajectory. These observables were then averaged over the rest of the trajectory, with errors determined in a blocking-equivalent manner.

Specifically, in these simulations, we focused on three pairs of parameters, which we name bilayer ($T_{\text{eff}} = 1.1$), flexible bolalipids ($k_{\text{bola}} = 0$, $T_{\text{eff}} = 1.2$), and slightly rigid bolalipids at $k_{\text{bola}} = 1\,k_B T$ ($T_{eff} = 1.4$). To guarantee correct integration, we lowered the timestep until the ratio $\left| (P_x + P_y)/(2P_z) \right|$ was less than 0.1. We also balanced increasing the cylinder length, to improve statistics on the stress tensor measurement, against computational performance and the occurrence of long-term deviations from cylindrical shape at high aspect ratio $R/l_z$. We include the exact parameters and data used in the plots in the main text in *Appendix 1—tables 1–3*.

**Appendix 1—table 1.** Parameters and measurements for bilayer lipid cylinders. $N$ is number of lipids.

| $\Delta t$ | $N$ | $l_z$ | $H$ | $\left\|\frac{P_x+P_y}{2P_z}\right\|$ | $\kappa$ |
|---|---|---|---|---|---|
| | 19457 | 120 | $(3.08+/-0.04)$e-02 | $0.03+/-0.06$ | $8.9+/-2.4$ |
| | | 70 | $(3.447+/-0.004)$e-02 | $0.03+/-0.05$ | $8.4+/-1.1$ |
| | 10000 | 80 | $(3.923+/-0.008)$e-02 | $0.04+/-0.04$ | $8.1+/-0.6$ |
| | | 40 | $(3.948+/-0.006)$e-02 | $0.05+/-0.05$ | $8.8+/-1.3$ |
| | | 50 | $(4.942+/-0.009)$e-02 | $0.011+/-0.021$ | $8.6+/-0.6$ |
| | | 60 | $(5.857+/-0.005)$e-02 | $0.018+/-0.017$ | $8.22+/-0.33$ |
| | | 70 | $(6.82+/-0.02)$e-02 | $0.010+/-0.013$ | $8.27+/-0.30$ |
| 0.001 | 5000 | 80 | $(7.17+/-0.03)$e-02 | $0.009+/-0.011$ | $8.7+/-0.3$ |
| | | 90 | $(7.86+/-0.03)$e-02 | $0.010+/-0.006$ | $9.16+/-0.21$ |
| 0.005 | 5000 | 96 | $(8.13+/-0.03)$e-02 | $0.085+/-0.013$ | $8.87+/-0.27$ |

**Appendix 1—table 2.** Parameters and measurements for flexible bolalipids tethers.

| $\Delta t$ | $N$ | $l_z$ | $H$ | $\left\|\frac{P_x+P_y}{2P_z}\right\|$ | $\kappa$ | $u_f$ |
|---|---|---|---|---|---|---|

*Appendix 1—table 2 Continued*

|  |  |  | $H$ | $\left\|\frac{P_x+P_y}{2P_z}\right\|$ | $\kappa$ | $u_f$ |
|---|---|---|---|---|---|---|
|  | 19457 | 120 | (3.024+/-0.004)e-02 | 0.06+/-0.08 | 9.2+/-1.6 | 0.534+/-0.006 |
|  |  | 70 | (3.354+/-0.004)e-02 | 0.03+/-0.08 | 9.7+/-2.1 | 0.536+/-0.004 |
|  | 10000 | 80 | (3.848+/-0.004)e-02 | 0.04+/-0.05 | 9.5+/-1.4 | 0.537+/-0.005 |
|  |  | 40 | (3.907+/-0.006)e-02 | 0.04+/-0.07 | 9.6+/-2.0 | 0.531+/-0.004 |
|  |  | 50 | (4.821+/-0.004)e-02 | 0.03+/-0.04 | 9.7+/-1.0 | 0.542+/-0.005 |
|  |  | 60 | (5.672+/-0.006)e-02 | 0.021+/-0.024 | 9.3+/-0.6 | 0.544+/-0.005 |
|  |  | 70 | (6.35+/-0.02)e-02 | 0.020+/-0.018 | 9.2+/-0.5 | 0.546+/-0.005 |
| 0.001 | 5000 | 80 | (7.19+/-0.03)e-02 | 0.008+/-0.012 | 9.1+/-0.4 | 0.551+/-0.005 |
|  |  | 90 | (7.82+/-0.03)e-02 | 0.095+/-0.007 | 9.70+/-0.16 | 0.5583+/-0.0015 |
| 0.005 | 5000 | 96 | (8.13+/-0.03)e-02 | 0.088+/-0.012 | 9.35+/-0.28 | 0.563+/-0.004 |

**Appendix 1—table 3.** Parameters and measurements for bolalipid tethers at $k_{\mathrm{bola}} = 1\,k_B T$.

| $\Delta t$ | $N$ | $l_z$ | $H$ | $\left\|\frac{P_x+P_y}{2P_z}\right\|$ | $\kappa$ | $u_f$ |
|---|---|---|---|---|---|---|
|  | 19457 | 120 | (2.921+/-0.006)e-02 | 0.020+/-0.032 | 23.3+/-2.1 | 0.07401+/-0.00033 |
|  |  | 70 | (3.260+/-0.002)e-02 | 0.043+/-0.028 | 23.3+/-1.7 | 0.0806+/-0.0009 |
|  | 10000 | 80 | (3.743+/-0.002)e-02 | 0.021+/-0.022 | 22.0+/-1.5 | 0.0898+/-0.0009 |
|  |  | 50 | (4.526+/-0.003)e-02 | 0.009+/-0.022 | 19.5+/-1.0 | 0.1053+/-0.0010 |
| 0.001 | 5000 | 60 | (5.375+/-0.004)e-02 | 0.010+/-0.015 | 17.8+/-0.5 | 0.1225+/-0.0008 |

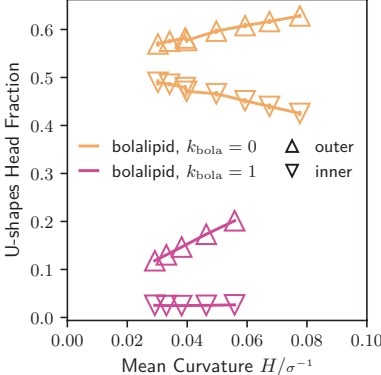

**Appendix 1—figure 10.** Fraction of lipid heads belonging to U-shaped bolalipids, for both outer (upwards triangles) and inner (downwards triangles) leaflets in cylinder membranes versus mean curvature $H$, for bolalipid membranes at $k_{\mathrm{bola}} = 0$ and $k_{\mathrm{bola}} = 1\,k_{\mathrm{B}}T$.

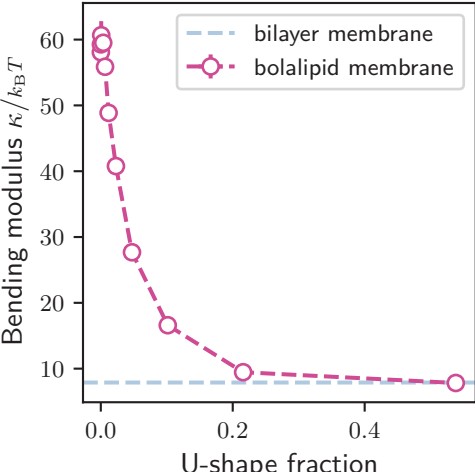

**Appendix 1—figure 11.** Bending modulus $\kappa$ versus U-shaped bolalipid fraction, using the same data as *Figure 2B and C*. For reference, the bilayer bending modulus at $T_{\mathrm{eff}} = 1.1$ is shown in blue.

We will now, in detail, explain the different behaviour of flexible versus stiffer bolalipid membranes under bending. Membrane bending rigidity in bolalipid membranes decreases dramatically once a small fraction of U-shapes is allowed to form, but then plateaus once this U-shape fraction reaches 20%. In a curved bolalipid membrane, U-shapes must accumulate in the outer leaflet to accommodate for area difference. Together, the bending rigidity non-linear dependence on U-shape fraction and the promotion of U-shapes by curvature explain why, in a membrane made of moderately stiff bolalipids ($k_{\mathrm{bola}} = 1\,k_{\mathrm{B}}T$), which contain very few U-shapes in the flat state, the bending rigidity of the membrane decreases as curvature increases. While in a membrane made of flexible bolalipid molecules ($k_{\mathrm{bola}} = 0$), where many U-shapes are present in the flat membrane, the bending rigidity does not change with curvature.

Bending rigidity $\kappa$ in flat membranes composed of bolalipids decreases dramatically once a small fraction of U-shapes is allowed to form, but plateaus once more than 20% of U-shaped bolalipids are present. In detail, our data shows that with an increasing bolalipid molecular rigidity $k_{\mathrm{bola}}$, both the number of U-shaped bolalipids decreases (*Figure 2B*) and the membrane rigidity $\kappa$ increases (*Figure 2C*). Thus, the correlation suggests that U-shaped bolalipids soften the membrane, in a non-linear way, where most of the change in membrane bending rigidity happens for U-shaped bolalipid fraction <20% (*Appendix 1—figure 11*).

Separately, membrane curvature affects the area difference between curved membrane leaflets and thus drives U-shape accumulation. To be specific, a cylindrical membrane with area $A$, mean curvature $H$ and thickness $h$ has the outer leaflet with area $A(1 + Hh)$ and the inner leaflet with smaller area $A(1 - Hh)$. This can be large, in our simulations up to an area change of $Hh = 25\%$. For pure bolalipid membranes, straight bolalipids occupy the same space in each leaflet. Area difference can then be achieved only by having a different amount of U-shaped bolalipids in each leaflet, which can result in a different U-shape fraction between leaflets and thus asymmetry between leaflets. *Appendix 1—figure 10* confirms U-shape head fraction asymmetry that increases with curvature, for both flexible ($k_{\mathrm{bola}} = 0$) and moderately stiff bolalipids ($k_{\mathrm{bola}} = 1\,k_BT$).

Together, these two effects result in membrane softening under curvature for the moderately stiff bolalipids, but constant rigidity for flexible bolalipids (*Figure 2F*). In detail: for membranes composed of moderately stiff bolalipid molecules ($k_{\mathrm{bola}} = 1\,k_BT$), the U-shape bolalipid head fraction only increases in the outer leaflet, going from 10 to 20% (*Appendix 1—figure 10*). This is in the high sensitivity region where the bending rigidity is expected to change the most (*Appendix 1—figure 11*). We hypothesize that the molecular rigidity of a U-shaped bolalipid creates compression on the outer leaflet that stabilizes the membrane curvature and thus causes membrane softening. We suspect that for membranes composed of rigid bolalipids ($k_{\mathrm{bola}} > 1k_BT$), the effect is likely not present due to the absence of U-shape formation even under strong bending.

By contrast, for membranes composed of flexible bolalipids ($k_{\mathrm{bola}} = 0$), the U-shaped bolalipid head fraction changes relatively little from its value for flat membranes (from 50% to respectively

60% and 40% for the outer and inner leaflet, *Appendix 1—figure 10*). This is in the region where the membrane bending rigidity is expected to respond weakly to U-shape fraction (*Appendix 1—figure 11*). Additionally, the change is symmetric, so presumably the outer leaflet becomes softer as the inner leaflet becomes stiffer, thus creating opposing effects and only weakly affecting the membrane bending rigidity as a whole. We note that the distinction between the U-shape head fraction that we plot (*Appendix 1—figure 10*) and U-shape fraction (*Appendix 1—figure 11*) matters little for this analysis.

## 9 Determining the Gaussian bending rigidity

In order to determine the Gaussian bending rigidity $\bar{\kappa}$ both of bilayer and flexible bolalipid membranes, we closely followed the procedure as detailed in *Hu et al., 2012*. As a consequence, we here only describe the general idea and provide the specific parameter values that we used. Details of the simulation protocol can be found in the original publication.

In short, we registered the closing frequency of membrane patches that we initialized as spherical caps. By simulating the closing of membrane patches many times and then repeating the procedure for different degrees of spherical cap opening $x$, we could sample the probability $P(x)$ for precurved caps to close. By fitting $P(x)$ with an analytical expression, the folding probability was related to the dimensionless parameter $\xi$, which itself was related to the cap radius $R$, the membrane line tension $\gamma$, the membrane rigidity $\kappa$, and the Gaussian rigidity $\bar{\kappa}$. Using all (known) parameters, we then determined $\bar{\kappa}$, following *Hu et al., 2012*

$$\bar{\kappa} = \frac{\gamma R}{\xi} - 2\kappa. \tag{A7}$$

For the folding simulations, we either used 1000 bilayer lipids or 500 flexible bolalipids ($k_{\text{bola}} = 0$) and we simulated at $\omega = 1.5\sigma$ and $T_{\text{eff}} = 1.2$ or $T_{\text{eff}} = 1.3$. Initially, we distributed the lipids along a spherical cap. We allowed the lipids to reach equilibrium while we limited the available space of the lipids between two spherical shells. The full equilibration took $10250\tau$, while the last $100\tau$ were used to introduce different seeds. After the equilibration step, we run the simulation until the precurved membrane patch either fully closed or became flat, determined by the relative shape anisotropy $\kappa_{\text{s}}^2$. We used the same cut-offs for $\kappa_{\text{s}}^2$ as in *Hu et al., 2012*. We repeated the simulations for 8 different values of $x$ and 200 seeds each. The radius of the spherical caps $R$ was determined from the cap areas. The cap areas were determined from the averaged number of lipids per cap and the area per lipid. For the line tension, we determined the tension that acts on the open edge of a flat membrane in periodic boundary conditions, following the protocol described in *Hu et al., 2012*. Using the values for $\xi$, $R$, $\gamma$, and $\kappa$, we then determined $\bar{\kappa}$. All values are summarised in *Appendix 1—table 4*.

**Appendix 1—table 4.** The membrane type, $T_{\text{eff}}$, the area per lipid, the patch radius $R$, the folding parameter $\xi$, the membrane line tension $\gamma$, the bending rigidity $\kappa$, the Gaussian bending rigidity $\bar{\kappa}$, and the ratio of $-\bar{\kappa}/\kappa$.

| Memb | $T_{\text{eff}}$ | A/ lipid[$\sigma^2$] | $R$ [$\sigma$] | $\xi$ | $\gamma$ [$k_{\text{B}}T/\sigma$] | $\kappa$ [$k_{\text{B}}T$] | $\bar{\kappa}$ [$k_{\text{B}}T$] | $-\bar{\kappa}/\kappa$ |
|---|---|---|---|---|---|---|---|---|
| Bila | 1.2 | 1.3587 | 7.238±0.046 | 1.679±0.010 | 1.237±0.035 | 4.82±0.08 | −4.30±0.22 | 0.89±0.04 |
| Bola | 1.2 | 1.2902 | 7.165±0.0000 | 1.309±0.009 | 1.957±0.043 | 7.88±0.14 | −5.04±0.37 | 0.64±0.04 |
| Bila | 1.3 | 1.4238 | 7.1999±0.122 | 1.660±0.008 | 0.917±0.031 | 3.36±0.05 | −2.74±0.18 | 0.81±0.05 |
| Bola | 1.3 | 1.3475 | 7.322±0.000 | 1.3466±0.009 | 1.381±0.038 | 4.79±0.14 | −2.17±0.35 | 0.45±0.07 |

## 10 Simulation protocol for cargo budding

To model a spherical cargo of radius $R_{\text{c}} = 8\sigma$ being adsorbed by a membrane, we set up a Lennard-Jones potential between the cargo and lipid beads using *Equation A2*. We parametrise the strength of the potential $U_m = \epsilon_{\text{mc}}$, calling $\epsilon_{\text{mc}}$ the adsorption energy. With lipid tail beads, the interaction is purely repulsive, with $r_m = r_c = 2^{1/6}(8 + 0.5)\sigma \approx 9.5\sigma$. With lipid head beads, we set up an attractive

well by setting $r_c = 2^{1/6}(8 + 0.5) \cdot 1.2\sigma \approx 11.5\sigma$, giving the well a width of $\approx 2\sigma$. This limits this attractive interaction to the head beads of the membrane leaflet closest to the cargo.

For budding simulations (*Figures 4 and 5*), we first pre-equilibrated for $10^4\tau$ a membrane with $60^2$ head beads per leaflet before placing the cargo bead on top of the membrane. To displace any lipids that might be inside the cargo's volume, we performed a short run in the constant volume ensemble, while scaling from zero to full strength the membrane-cargo interaction. We then ran the simulation at constant pressure until the system was in equilibrium for at least $\Delta_{\text{eq}} = 30 \times 10^3\tau$; the end state should then be either partial or full membrane budding. In practice, total runtime was $\Delta_{\text{total}} \in [60, 120] \times 10^3\tau$. For a significant region of membrane parameters of interest, as we increased $\epsilon_{\text{mc}}$ we observed the equilibrium state transition directly from non-budded to a collapsed state with the membrane folded around the cargo in a small simulation box. To impose the existence of a region with budding, we kept all membranes in budding simulations minimally stretched by setting the barostat target lateral pressure to $P_{\text{xx}} = P_{\text{yy}} = -0.001 k_{\text{B}}T/\sigma^3$.

Note that it was not possible to measure $\epsilon_{\text{mc}}^*$ for $f^{\text{bi}} = 1$ since budding was always followed by membrane disassembly as the bilayer membrane is close to the gas phase for the used parameters.

## 11 Area per head measurements

*Appendix 1—figure 12A* shows the area per head bead as a function of $k_{\text{bola}}$ in the inner and outer layer of the membrane bud and the flat mother membrane.

We can use the area per head as a proxy for tension, by comparing with the values for the flat mother membrane. We note that these measurements excluded membrane within $3\sigma$ of a pore. The plot shows that for bolalipid membranes, while the area per head is non-monotonic in $k_{\text{bola}}$, it varies by less than $\approx 4\%$. In contrast, for membrane mixture of bilayer and bolalipids, the area per head as a function of $f^{\text{bi}}$ monotonically increases (*Appendix 1—figure 12*), varying by $\approx 10\%$. By default, both inner and outer leaflets would be relaxed if possible, at the same value of area per head as the flat mother membrane. However, the cargo bead competes with tension and pulls head beads from the outer to the inner layer, thus increasing the area per head for the outer layer and decreasing it for the inner layer. This happens both for relatively flexible bolalipids and mixture membranes with increasing bilayer fraction, which can indeed move head beads between leaflets by, respectively, forming and/or flip-flopping U-shaped bolalipids and flip-flopping bilayer lipids. When this is not possible, the area per head values for inner and outer layer are roughly equal, such as for the mixture membranes with nearly no bilayer lipids at $f^{\text{bi}} \approx 0.2$. This also happens for bolalipids with $k_{\text{bola}} > 2 k_B T$, where we note that the area per head for the bud layers are higher than that for the relaxed membrane; this can be understood by considering that at these $k_{\text{bola}}$ values the bud membrane has pores, whose line tension then competes with tension, stretching the membrane. Lastly, the trend inversion for bolalipids at $k_{\text{bola}} = 0.5 k_B T$ can be understood by considering our choice of $T_{\text{eff}}$ for each $k_{\text{bola}}$ changes slope at the same point (see *Figure 2A*).

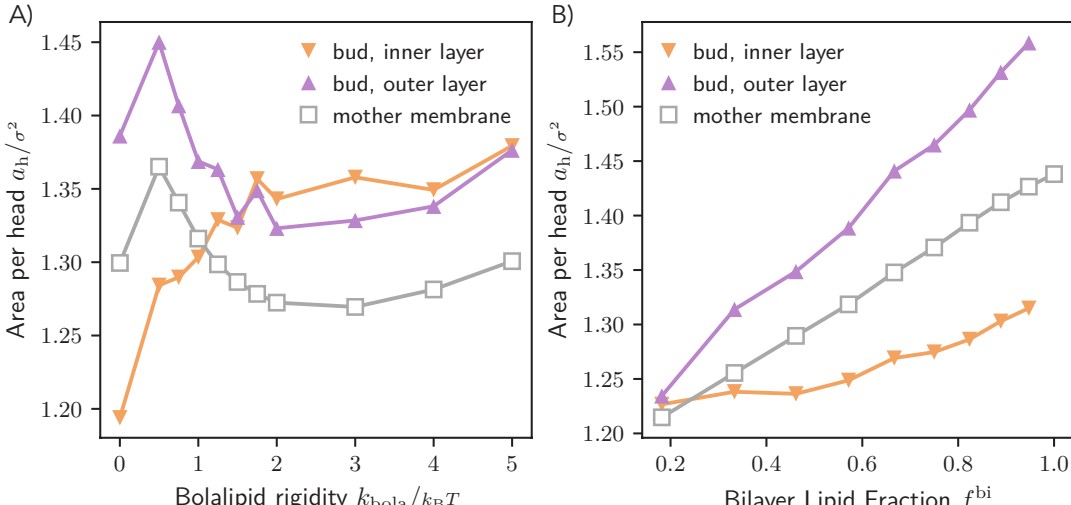

**Appendix 1—figure 12.** Area per head bead measurements, in the inner and outer layer of the membrane bud, as well as the flat mother membrane, for (**A**) bolalipid membranes as a function of $k_\mathrm{bola}$ and (**B**) membranes of a mixture of bilayer lipids and bolalipids as a function of the bilayer lipid fraction $f^\mathrm{bi}$.

## 12 Membrane structure and lipid conformation

We developed a pipeline for analysing each frame of membrane simulation trajectories, using the pipeline framework and components from OVITO *Stukowski, 2010*.

### Cluster identification and surface reconstruction

We distinguished and identified clusters by fitting mean position and orientation of lipids to the expected membranes: a horizontal plane for the flat membrane simulations and the mother membrane in budding simulations and a sphere for the budded membrane in budding simulations. We constructed the membrane surface from the set of lipid beads using the alpha-shape method with radius $1.5\sigma$ (implemented in *Stukowski, 2010*); this ensures any pore of diameter $> 1\sigma$ will not be closed over by the resulting surface.

### Midplane construction

We then constructed the midplane of the membrane by clustering the faces of the Voronoi diagram of the membrane surface vertices that are nearly coplanar and inside the membrane surface and then meshing the resulting oriented point cloud. This procedure is general enough to work for pre-budding frames of the budding simulations. The midplane orientation is adjusted to be consistent frame to frame.

### Membrane pore identification

By computing the signed distance to the midplane, we assigned a leaflet to each membrane surface element. We then intersected the membrane surface with the midplane surface, obtaining for each pore a line marking its perimeter. For our purposes, it was sufficient to project each pore perimeter into a least-squares fitted plane and compute the area and perimeter from the projected line. For leaflet area measurements, we considered the two surfaces at equal distance between the membrane surface and the midplane.

For the measurements of pore diameter in *Figures 4D and 5D*, we first took the ensemble average, i.e., time average with rescaling of std. mean deviation according to autocorrelation, of the total pore area.

We consider a point to be part of a pore if its projection in the midplane membrane is within $3\sigma$ of its surface. We can also assign to a point a leaflet corresponding to its signed distance to the midplane. We apply this location classification to the head bead of each bilayer lipid and each half of a bolalipid. Due to the bolalipid's symmetry, some combinations of conformation (given by the sign of the angle between their head beads) and location are indistinguishable, while others are transition

ephemeral states that for our ensemble measures were not relevant (e.g. U-shapes that have one head bead in each layer). When both head beads are on the same leaflet, the only relevant states are the U-shape, where the heads beads angle is $\leq \pi/2$ and the flat state, where the angle is $> \pi/2$. When the head beads are in different leaflets, we get the straight state. Thus, each lipid is assigned a state, some of which have a leaflet; additionally, we consider a lipid to be in a pore if any of it(s) head bead(s) is near a pore.

For simulations where pore formation was significant, namely the simulations with pure bolalipid membranes, we present in the main text conformation measurements that exclude lipids that are in a pore. To clarify that this does not qualitatively change our results, we redid the measurements in *Figure 4C*, considering exclusively lipids near pores and compared (see *Appendix 1—figure 13*). Not surprisingly, the only value to change significantly is the fraction of flat bolalipids, which is much larger near pores.

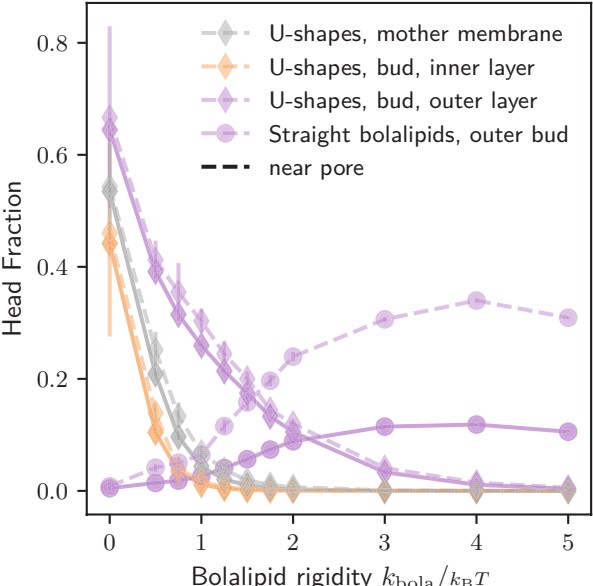

**Appendix 1—figure 13.** Lipid conformation and location in the membrane bud for pure bolalipid membranes, excluding lipids near pores (solid lines) and exclusively considering lipids near pores (dashed lines).

## Lipid species/conformation per leaflet

For analysing membrane composition in lipid species and conformation, we present head bead fractions. In this way, the denominator is a function of geometry and thus approximately conserved. For instance, the fraction of head beads that belong to bilayer lipids $f_{\mathrm{h}}^{\mathrm{bi}}$ is related in the following way to the fraction of lipids that are bilayer lipids, $f^{\mathrm{bi}}$

$$f_{\mathrm{h}}^{\mathrm{bi}} = \frac{n^{\mathrm{bi}}}{n^{\mathrm{bi}} + 2n^{\mathrm{bola}}} = \frac{1}{\frac{2}{f^{\mathrm{bi}}} - 1} = f^{\mathrm{bi}} \frac{1}{1 + (1 - f^{\mathrm{bi}})}, \tag{A8}$$

where $n^{\mathrm{bi}}$ is the number of bilayer head beads and $n^{\mathrm{bola}}$ is the number of bolalipid head beads. Therefore, near $f^{\mathrm{bi}} = 0$, $f_{\mathrm{h}}^{\mathrm{bi}}$ is nearly half $f^{\mathrm{bi}}$, while near $f^{\mathrm{bi}} = 1$, $f_{\mathrm{h}}^{\mathrm{bi}} \approx f^{\mathrm{bi}}$.

## 13 Lipid conformation during budding

To justify our qualitative observations of bilayer lipids and U-shaped bolalipids, respectively in mixtures and pure bolalipid membranes, enriching at the positive curvature leaflets during budding simulations, we plot the average composition of the axially symmetric profile around the cargo bud. To do this, we chose a trajectory section where the shape of the membrane was roughly static. Then, within a $2000\tau$ time interval, we sampled uniformly 100 frames. For each frame, we took a cylindrical section of the simulation centred on the membrane and axis pointing upwards towards

$z$. We assigned to each particle a conformation or species, respectively. Using the cargo bead as the origin for cylindrical coordinates $(r, \theta, z)$, dropping the angle, and using hexagonal binning on the $r, z$ plane, for each conformation/ species we computed the total number of particles observed. Using these totals, we then thresholded on a minimum number of 40 particles observed per bin and plotted the composition as a fraction of (# studied conformation or specie) / (total in bin) (*Appendix 1—figure 14*).

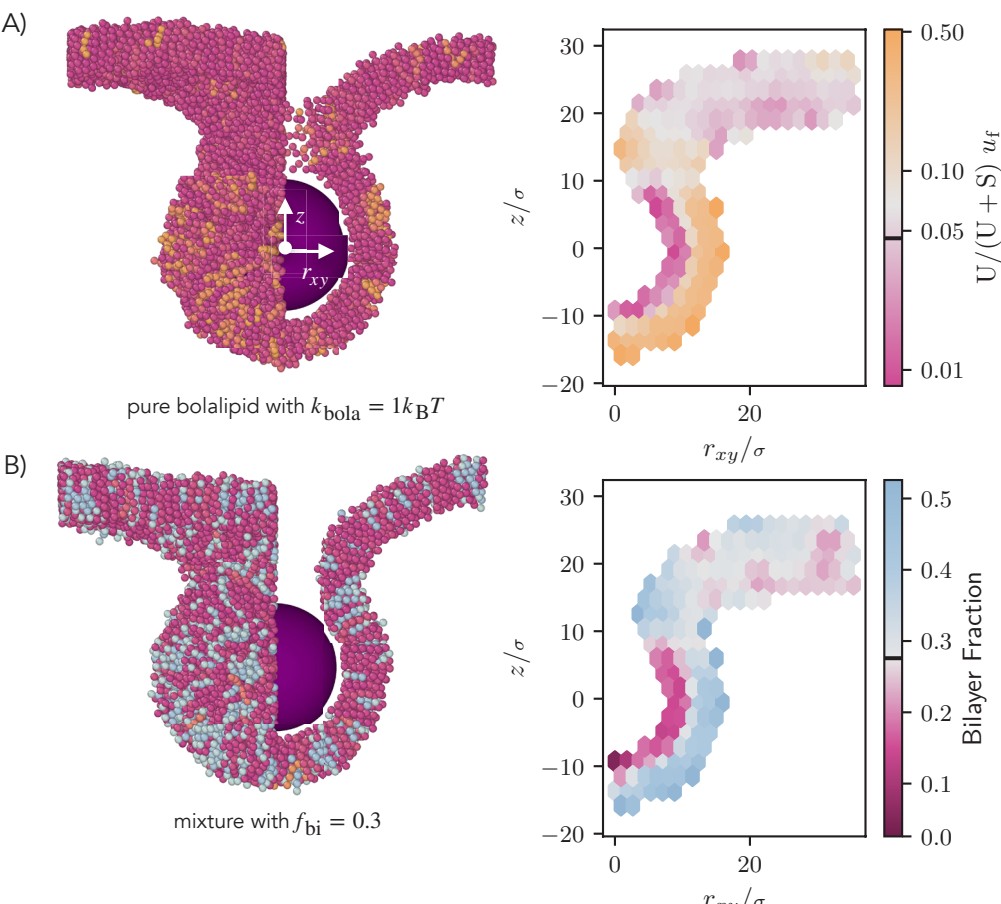

**Appendix 1—figure 14.** Lipid conformation and location in the membrane bud for pure bolalipid membranes, before budding, (**A**) for pure bolalipids at $k_{bola} = 1\,k_B T$ and for (**B**) a mixture of bolalipids with 30% bilayer, averaged over 100 frames spaced over a $2000\tau$ time interval. For both cases, a snapshot (left) is shown with the front right half of the membrane cut away, showing the profile shape, matching the (right) time-averaged spatially varying conformation or specie fraction. Visible at $z > 10\sigma$ and for small $r_{xy}$ is the inner region of the neck. While for mixture membranes the effect on bilayer fraction is visible using a linear scale, for bolalipid membranes at $k_{bola} = 1\,k_B T$ the effect spans multiple orders of magnitude so a logarithmic scale was used. The flat membrane values, taken as average values of the bins at $r_{xy} > 20\sigma$, are indicated by a black mark on the colour bar.

## 14 Scaling temperature

In this section, we show that little changes in our work when scaling temperature $T$ instead of our effective temperature $T_{eff}$. If we compare our phase diagrams when scaling $T_{eff}$ to scaling simulation temperature $T$, we obtain very similar results (*Appendix 1—figure 15*).

Naturally, the fraction of U-shapes becomes now a function of temperature, but by keeping to the previously defined $(T_{eff}, k_{bola})$ line, we maintain the same qualitative disappearance of U-shapes as $k_{bola}$ is increased (*Appendix 1—figure 16*). The effect on bolalipid fraction and bending rigidity becomes, respectively, less abrupt and strong (*Appendix 1—figure 17*), but qualitatively we have the same trends as discussed in the main text (*Figure 2B and C*).

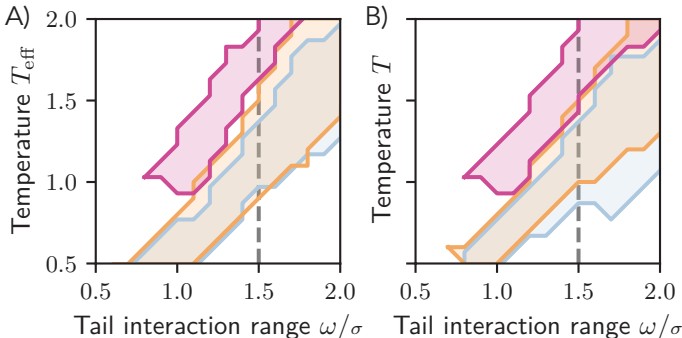

**Appendix 1—figure 15.** Phase diagram for pure bilayer and bolalipid membranes (flexible and rigid), as in *Figure 1F*, for (**A**) $T_{\mathrm{eff}}$ and (**B**) $T$ scaling.

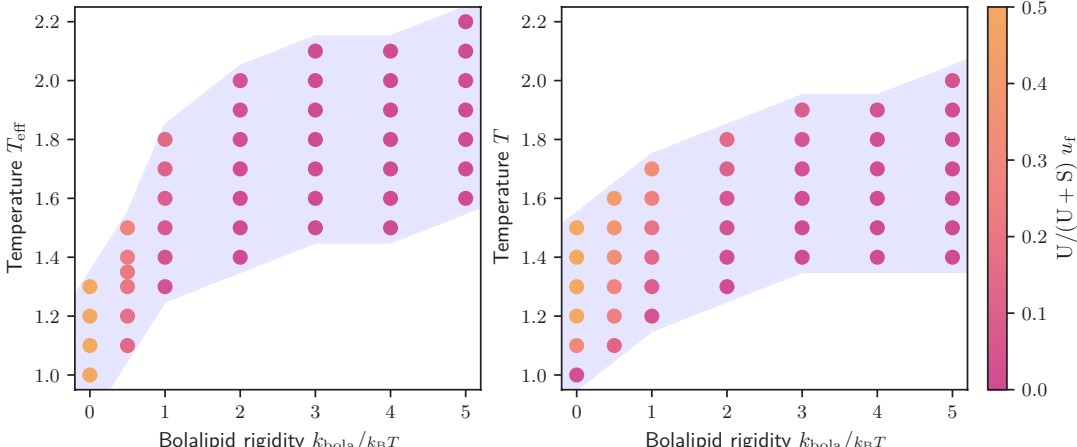

**Appendix 1—figure 16.** Liquid region (blue) for bolalipid membranes with potentials setup as in the main text as a function of $k_{\mathrm{bola}}$ and $T_{\mathrm{eff}}$ (**A**), and as a function of $k_{\mathrm{bola}}$ and $T$ (**B**), with U-shape fraction in colour.

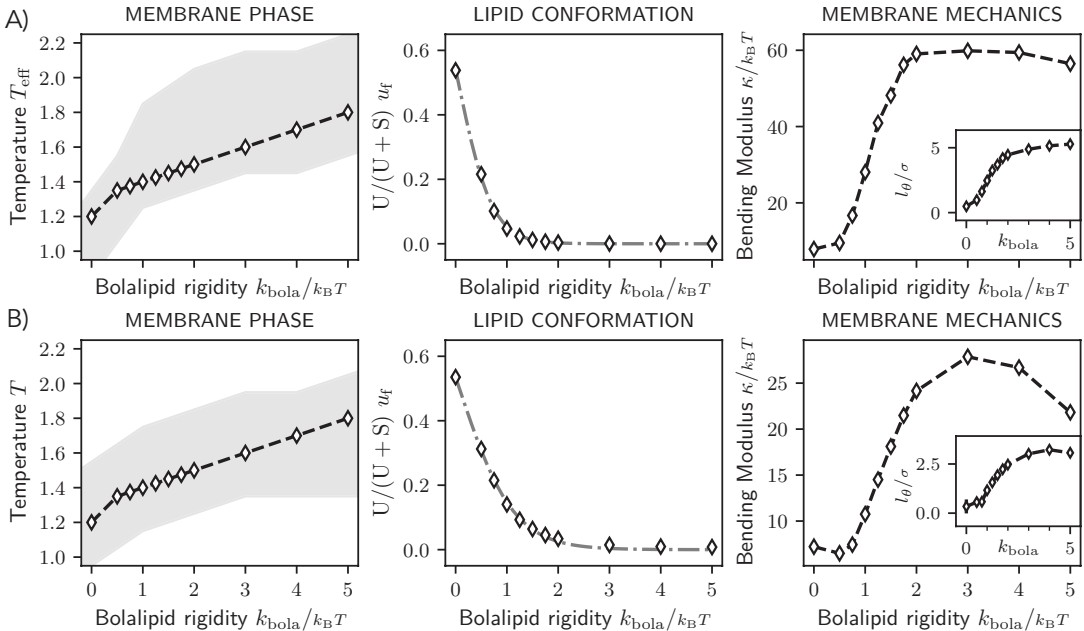

**Appendix 1—figure 17.** Comparison between using $T_{\mathrm{eff}}$ for *Figure 2* (**A**), with scaling temperature $T$ for the same parameters (**B**). The tilt contribution for $k_{\mathrm{bola}} < 1\,k_B T$ is negligible.

## 15 Maximum curvature and the validity of the Monge Gauge

In this section, we show that the maximum curvature imposed on the membrane during fluctuation spectrum measurements is not sufficient to either invalidate the use of the Monge gauge or reach the curvature-dependent regime found for semi-flexible bolalipid membranes (see *Figure 2*).

We first validate the Monge gauge for describing the membrane surface. We can parameterise a periodic surface without overhangs by a height field $h(x, y)$ relative to a reference plane; we can then perform a Fourier transform in space on this field:

$$h = h\left(\vec{r}\right) = \sum_n a_n e^{i(\vec{q}_n \cdot \vec{r} + \phi_n)} = a_0 + \sum_{\substack{n \in n_+: \\ n_x > 0 \text{ or} \\ n_x = 0\,\&\,n_y > 0}} h_n \cos\left(\vec{q}_n \cdot \vec{r} + \phi_n\right),$$

where $n$ are mode number vectors, and to each instantaneous real amplitude $h_n := 2a_n$ corresponds one degree of freedom. $a_0$ is just the average membrane height, kept constant by ensuring the membrane momentum is zero, via constraining the initial thermalisation and the thermostat. The wave vector components are $q_{n,i} = 2\pi n_i / L_i$; we used $a_n = a_{-n}$ and for the phase $\phi_n = -\phi_{-n}$.

To apply the Monge gauge means taking several approximations on derivatives of $h$, and thus is only valid if $|\nabla h| \ll 1$. We expand to obtain:

$$\left(\nabla h\right)_i = \sum_{i, n \in n_+} h_n q_{n,i} \sin\left(\vec{q}_n \cdot \vec{r} + \phi_n\right).$$

Since we observed amplitude decreasing with $|n|$, the first two modes, with $|n| = 1$, contribute the most to the gradient. We discard the other modes and obtain an upper bound:

$$|\nabla h| \leq \frac{2\pi}{L}\sqrt{h_{1,0}^2 + h_{0,1}^2}. \tag{A9}$$

How small should be $|\nabla h|$? From *Cooke and Deserno, 2005* Figure 5 (bilayer membrane with $w = 1.4$, $\epsilon = 1$), we find $L = 50$, $\langle h^2 \rangle L^2 \approx 500$ for $n = 1$ (the point x-axis coordinate matches up with expected $(2\pi/50)^2 = 0.016$). Assuming the average was taken over both modes (with $\vec{n} = (0, 1)$ and $\vec{n} = (1, 0)$) we can write:

$$\langle h_n^2 \rangle_{|n|=1}^{1/2} = \left\langle \sqrt{\frac{h_{1,0}^2 + h_{0,1}^2}{2}} \right\rangle \leq \max \sqrt{h_{1,0}^2 + h_{0,1}^2} \; .$$

While we do not have access to the data to determine the RHS, since the LHS is strictly less or equal to the RHS, we obtain that for Cooke and Deserno the threshold for the norm of the gradient was at 8% or more.

In our sims, we left the modes with $n \leq 2$ unequilibrated. Therefore, we cannot use averages directly for these modes, since they might smooth out an otherwise excessive deviation from the Monge Gauge. Instead, we compute $\max |\nabla h|$ for each frame, take the average of each $1000\tau$ block to exclude transient behaviour unlikely to affect membrane properties, and take the maximum of these averages over the full trajectory. We take the maximum over replicas.

In our simulations, we thus obtain, for the bilayer at $T_{\text{eff}} = 1.1$, 12%; for flexible bolalipids ($k_{\text{bola}} = 0$), 13%; for stiffer bolalipids ($k_{\text{bola}} = 1\,k_BT$), 6%, and for rigid bolalipids ($k_{\text{bola}} = 5\,k_BT$), 4%.

All of these values are close by 5% at most to the threshold used by Cooke and Deserno, so we are confident to apply the Monge Gauge. Arguably, by forcing the most flexible of these membranes to stay flat (e.g. by fixing the simulation box size to a slightly larger value), we could reduce the deviation at the cost of introducing excessive tension, which would mask the effect of the bending modulus $\kappa$.

Improvements notwithstanding, continuing with the Monge Gauge, we have for the maximum mean curvature the expression:

$$H_{\text{max}} = \max \frac{|\nabla^2 h|}{2}$$

$$= \max \left| \frac{1}{2} \sum_{n \in n_+} h_n |\vec{k}_n|^2 \cos\left(\vec{k}_n \cdot \vec{r} + \phi_n\right) \right|$$

$$\leq \frac{|h_{1,0} + h_{0,1}|}{2} \left(\frac{2\pi}{L}\right)^2,$$

where in the last equality we restricted ourselves to the two largest wavelength modes.

Since we are interested in the steady state, we repeat the procedure used for obtaining the slope, obtaining, respectively, for each membrane, 0.025 (bilayer), 0.028 (flexible bolalipids), 0.012 (stiffer bolalipids) and 0.01 (rigid bolalipids), in $\sigma^{-1}$.

The maximum mean curvature for the stiffer bolalipids ($k_{\text{bola}} = 1$) reaches at most $0.012\sigma^{-1}$ in a fluctuation measurement. We can interpolate the data from *Figure 2F* to obtain that at this mean curvature, the bending modulus is reduced by 7%, compared to its value for a flat membrane. Given this is for the maximum value of membrane mean curvature, we expect a much smaller change on the effective bending rigidity. Therefore, we find our analysis of their spectrums, which assumes a constant bending modulus, valid.

## 16 Tilt model versus tension model

### Membrane tension cannot explain the rigid bolalipid spectrum

The tilt model has different asymptotic behaviour compared to the tension model. The tilt model exhibits the functional form $1/\left(\kappa q^4\right) + 1/\left(\kappa_\theta q^2\right)$. In contrast, the tension model exhibits the functional form $1/\left(\kappa q^4 + \Sigma q^2\right)$. For the tilt model, while for small $q$ the amplitude is proportional to $q^{-4}$, for large $q$ the amplitude is proportional to $q^{-2}$. In contrast, for the tension model (with positive tension), while for small $q$ the amplitude is proportional to $q^{-2}$, for large $q$ the amplitude is proportional to $q^{-4}$. If membrane tension were to be negative in the tension model, the slope would cross from negative infinity for small $q$ to $-4$ for large $q$. Consequently, between the tilt and tension models, the only model that can fit the measured slope that goes from $-4$ to $-2$ with increasing $q$ is the tilt model (best fits for each shown in *Appendix 1—figure 18*).

Moreover, we indirectly determined membrane tension by fitting both tension and tilt with the expression:

$$\langle |h_q|^2 \rangle = \frac{k_{\mathrm{B}}T}{L^2} \left( \frac{1}{\kappa q^4 + \Sigma q^2} + \frac{1}{\kappa_\theta q^2} \right) . \tag{A10}$$

The fit to this model with both tension and tilt overlaps with the tilt only model (also shown in *Appendix 1—figure 18*), and the membrane tension fitted is $-0.02 \pm 0.05\, k_{\mathrm{B}}T/\sigma^2$. At the longest wavelength ($q \approx 0.1\sigma^{-1}$), the tension required to match the bending term is reached when $\kappa q^4 + \Sigma q^2 \approx 0 \leftrightarrow \Sigma = -\kappa q^2 \approx -0.6\, k_{\mathrm{B}}T/\sigma^2 \ll -0.02$. Therefore, our barostating was effective at making tension small enough for its effect on the spectrum to be negligible.

We also measured the projected membrane tension $-L_z \langle P_x + P_y \rangle /2$, obtaining $-2(4) \times 10^{-3}\, k_{\mathrm{B}}T/\sigma^2$. This small, negligible negative value of projected tension, that approximately agrees with the fitted tension, confirms our barostat was properly working.

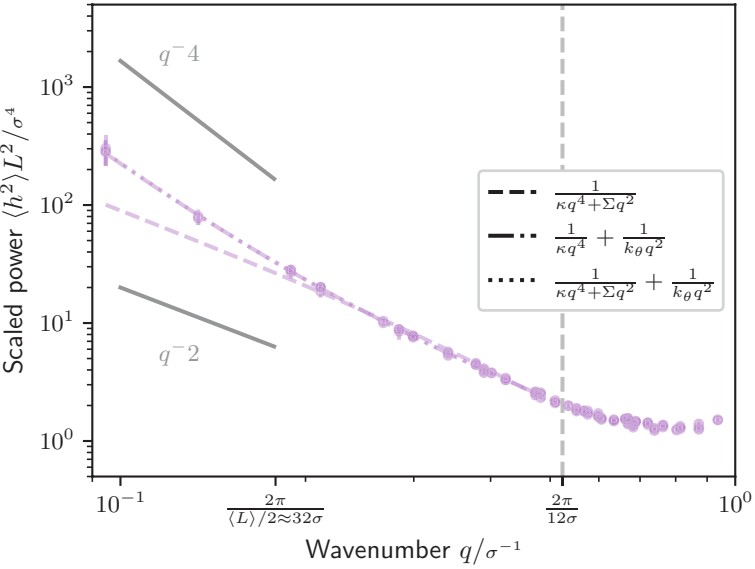

**Appendix 1—figure 18.** Height fluctuation spectrum, for a rigid bolalipid membrane at $T_{\mathrm{eff}} = 1.8$. Solid lines mark the $-2$, $-4$ slopes. The vertical dashed line marks the limit of the data used for fitting. The fits are different dashed lines; the fit that models both tension and tilt overlaps with the fit that only includes tilt.

## 17 Equilibration and ensembles

Here, we cover two details regarding our fluctuation simulations and analysis.

### Equilibrating the longer wavelengths does not change the relevant spectrum shown in the main text

This is expected since our method of measurement involves averaging each mode's amplitudes over 4 replicas, and we checked that our replicas in fact were not stuck in an initial starting position and were instead exploring different regions of the phase space.

To show without doubt that this procedure does not randomly bias our results, we also ran simulations for three representative membranes until all modes were equilibrated. In order to equilibrate the long wavelength modes (mode number $|\vec{n}| < 2$), we simulated for up to $1.44 \times 10^6\, \tau$. On the all modes except the long wavelength modes ($|\vec{n}| \geq 2$), the resulting amplitudes change little (*Appendix 1—figure 19*, 'nph +langevin (short)' and 'nph +langevin'). On the largest wavelength modes, we noticed a small deviation from theory, specifically for the bilayer and flexible bolalipid membranes ($k_{\mathrm{bola}} = 0$). These small deviations can be explained by including a negligible negative tension. Importantly, however, the resulting bending modulus $\kappa$ stays nearly the same. We note that the small negative tension disappears when we halve the timestep (*Appendix 1—figure 19*, 'nph +langevin (timestep halved)').

## Moreover, the relevant spectrum does not depend on how the integration is done

We tested bilayer at $T_{\text{eff}} = 1.1$, flexible bolalipids ($T_{\text{eff}} = 1.2, k_{\text{bola}} = 0$) and rigid bolalipids ($T_{\text{eff}} = 1.8, k_{\text{bola}} = 5$) using different integrators: zero lateral pressure with overdampened Langevin dynamics, implemented in LAMMPS via fix nph and fix langevin, as done in most of our work; a NPT ensemble, implemented in LAMMPS via fix npt; and Brownian dynamics in a fixed volume, implemented in LAMMPS via fix nve and fix langevin, where the box size is set to the average value in nph + langevin simulations. The resulting measurements of $\kappa$ are at most within 10% of the values reported in the main text (*Appendix 1—table 5*). The value of $l_\theta$ does increase by 30% for this bilayer system, but since it is still below our main text threshold of $2\sigma$ our conclusions do not change.

To start with, we get the same results if we equilibrate first with a barostat, fix the box size, and only then sample the fluctuation spectrum (*Appendix 1—figure 19*,'langevin'). It becomes then easy to expect that there is no significant difference for our simulations between using fix nph and fix langevin (*Appendix 1—figure 19*,'nph +langevin') versus fix npt (*Appendix 1—figure 19*,'npt'). Our barostat simply equilibrates our membrane to near zero tension and then does not significantly contribute to the system. While for bilayer and flexible bolalipid membranes, the first modes (i.e. $|\vec{n}| < 2$) amplitudes are noticeably above the fitted line for the tilt-including model, these can be reached by including a negative tension term with the negligible value of $-0.05k_{\text{B}}T/\sigma^2$ (*Appendix 1—figure 19*, 'npt' and 'nph +langevin'). We call this value negligible, since it is only roughly 3% of the tension necessary to rupture a membrane in this model (*Cooke and Deserno, 2005*), and its effect is long vanished at the wavelengths we use for the fits in the main text. We understand this not as an issue with our barostating, but simply as consequence of the $q^{-2}$ dependency on tension. Sufficiently large membranes and small bending modulus $\kappa$ will eventually reveal tension present either due to integration error or the approximation of membrane tension by the projected membrane tension. Nevertheless, to show nothing changes in the spectrum used in the main text, we halved the timestep (setting $\Delta t = 0.005\,\tau$), which reduces the fitted tension by 3 x and allows a good fit over the entire spectrum with the model that only includes tilt and not tension (*Appendix 1—figure 19*, 'timestep halved'). For rigid membranes (e.g. rigid bolalipid membranes), all modes are equilibrated for the shorter duration of $60 \times 10^3\,\tau$ used in the main text results, and we obtain the same results regardless of the integration scheme used (npt, nph +langevin, or fixed box size with Langevin).

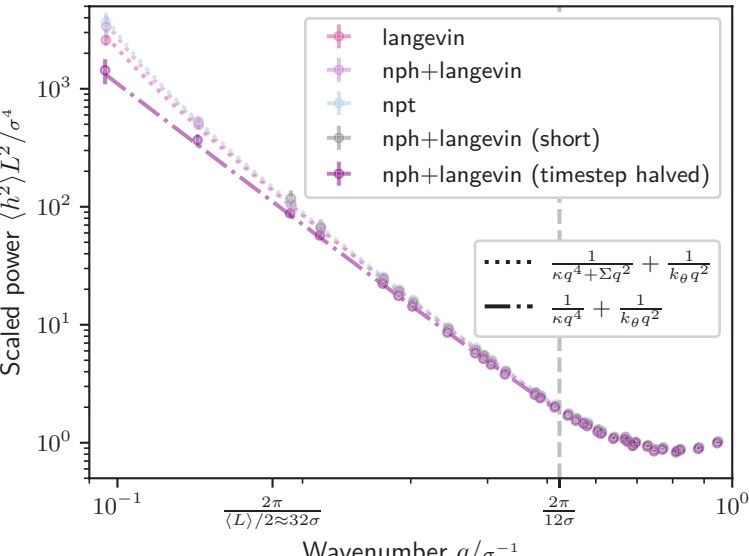

**Appendix 1—figure 19.** Height fluctuation spectrum, for a bilayer membrane at $T_{\text{eff}} = 1.1$ simulated under different integrators. The resulting fit parameters together with the normalized $\chi^2$ are shown in *Appendix 1—table 5*.

**Appendix 1—table 5.** Normalized $\chi^2$ and parameters for fits in *Appendix 1—figure 19*.

| | $\chi^2/N$ | $\kappa/k_B T$ | $\Sigma/\left(k_B T/\sigma^2\right)$ | $l_\theta/\sigma$ |
|---|---|---|---|---|
| langevin | 0.72 | 8.51±0.12 | −0.039±0.0004 | 0.85±0.05 |
| nph +langevin | 0.28 | 8.71±0.11 | −0.050±0.004 | 0.92±0.04 |
| npt | 0.25 | 8.35±0.09 | −0.0492±0.0026 | 0.86±0.04 |
| nph +langevin (short) | 0.17 | 7.88±0.25 | N/A | 0.60±0.16 |
| nph +langevin (timestep halved) | 0.31 | 9.13±0.12 | N/A | 0.97±0.05 |

