## [Editor Report · eLife Assessment]

This **fundamental** study characterizes the mechanics and stability of bolalipids from archaeal membranes using a minimalist, physics-based computational model. The authors present a robust mesoscale model of bolalipids-containing membranes, systematically evaluating it across diverse membrane configurations. The results are **compelling**, demonstrating that the incorporation of bolalipids and regular bilayer lipids in archaeal membranes significantly enhances membrane fluidity and structural stability.

---

## [Referee Report · Reviewer #2 (Public review)]

Summary:

The authors aimed to understand the biophysical properties of archeal membranes made of bolalipids. Bacterial and eukaryotic membranes are made of lipids that self-assemble into bilayers. Archea, instead, use bolalipids, lipids that have two headgroups and can span the entire bilayer. The authors wanted to determine if the unique characteristics of archaea, which are often extremophiles, are in part due to the fact that their membranes contain bolalipids.

The authors develop a minimal computational model to compare the biophysics of bilayers made of lipids, bolalipids, and mixtures of the two. Their model enables them to determine essential parameters such as bilayer phase diagrams, mechanical moduli, and the bilayer behavior upon cargo inclusion and remodeling.

The author demonstrates that bolalipid bilayers behave as binary mixtures, containing bolalipids organized either in a straight conformation, spanning the entire bilayer, or in a u-shaped one, confined to a single leaflet. This dynamic mixture allows bolalipid bilayers to be very sturdy but also provides remodeling. However, remodeling is energetically more expensive than with standard lipids. The authors speculate that this might be why lipids were more abundant in the evolutionary process.

Strengths:

This is a wonderful paper, a very fine piece of scholarship. It is interesting from the point of view of biology, biophysics, and material science. The authors mastered the modeling and analysis of these complex systems. The evidence for their findings is really strong and complete. The paper is written superbly, the language is precise and the reading experience very pleasant. The plots are very well-thought.

Weaknesses:

None. The authors have addressed all the potential weaknesses that were raised by the reviewers.

---

## [Referee Report · Reviewer #3 (Public review)]

Summary:

The authors have studied the mechanics of bolalipid and archaeal mixed-lipid membranes via comprehensive molecular dynamics simulations. The Cooke-Deserno 3-bead-per-lipid model is extended to bolalipids with 6 bead. Phase diagrams, bending rigidity, mechanical stability of curved membranes, and cargo uptake are studied. Effects such as formation of U-shaped bolalipids, pore formation in highly curved regions, and changes in membrane rigidity are studied and discussed. The main aim has been to show how the mixture of bolalipids and regular bilayer lipids in archaeal membrane models enhances the fluidity and stability of these membranes.

The authors have presented a wide range of simulation results for different membrane conditions and conformations. Analyses and findings are presented clearly and concisely. Figures, supplementary information and movies are of very high quality and very well present what has been studied. The manuscript is well written and is easy to follow.

The authors have provided detailed response to the points I raised on the first version and have revised their manuscript accordingly. Hence, I only mention what, in my opinion, still deserves to be noted.

Comments:

I previously raised an issue with respect to the resort to the Hamm-Kozlov model for fitting the power spectrum of membrane undulations. The authors provided very nice arguments against my concerns. For the sake of completeness, I include a simple scenario, which will better highlight the issue:

The tilt contribution to the Helfrich Hamiltonian can be written as a quadratic term 1/2 k_t |T|^2, where T is a tilt vector field. This field is written as the difference between the surface normal and the director field aligned with the lipid orientations. In the small deviation Monge description with z=h(x, y) as the height function, the surface normal has the form N=(-dh/dx, -dh/dy, 1). Now assume the director field, n = (b_x, b_y, 1) with small b_x and b_y components. The tilt contribution to the energy thus reads as 1/2 k_t (N - n)^2 ~ = 1/2 k_t [|grad h|^2 + 2 b . grad h]. The first term, 1/2 k_t |grad h|^2, is indeed similar to a surface tension term, \sigma |grad h|^2 that you get from the (1 + 1/2 |grad h|^2) approximation to the area element. Therefore, if you only look at height fluctuations, while your membrane actually has some surface tension, it will make distinguishing the tilt contributions to the fluctuations in the linear Monge gauge impossible.

However, considering that the authors have made sure that the membrane is indeed tensionless, this argument is settled.

I had also raised an issue about the correct NpT sampling in the simulations, and I'm glad that the authors also set up more rigorously thermostatted/barostatted simulations to check the validity of their findings.

Also, from the SI, I previously noted that the authors had neglected the longest wavelength mode because it was not equilibrated. This was an important problem and the authors looked into it and ran more simulations that were better equilibrated.

The analysis of energy of U-shaped lipids with the linear model E=c_0 + c_1 * k_bola is indeed very interesting. I am glad that the authors have expanded this analysis and included mean energy measurements.

---

## [Author Response]

The following is the authors’ response to the original reviews

**Public Reviews:**

**Reviewer #1 (Public review):**
Summary:Amaral et al. presents a study investigating the mesoscale modelling and dynamics of bolalipids.Strengths:The figures in this paper are exceptional. Both those to outline and introduce the lipid types, but also the quality and resolution of the plots. The data held within also appears to be outstanding and of significant (hopefully) general interest.

We thank the reviewer for their kind words and the appreciation of our work.

Weaknesses:In the introduction, I would like to have read more specifics on the biological role of bolalipids. Archaea are mentioned, but this kingdom is huge - there must be specific species that can be discussed where bolalipids are integral to archaeal life. The authors should go beyond ’extremophiles’. In short, they should unpack why the general audience should be interested in these lipids, within a subset of organisms that are often forgotten about.

Following the reviewer’s advice we have revised the introduction of the manuscript, in which we now discuss specific species (*Sulfolobus acidocaldarius* and *Thermococcus kodakarensis*) and how in these species bolalipids are integral to archaeal life. We explain that the ratio between bilayer and bolalipids, and the number of cyclopentane rings contained within bolalipids can change to adapt to the environment. The revised parts of the introduction read (p.1):

“Like for bacteria and eukaryotes, archaea must keep their lipid membranes in a fluid state (homeoviscous adaptation). This is important even under extreme environmental conditions, such as hot and cold temperatures, or high and low pH values [7]. Because of this, many archaea adapt to changes in their environment by tuning the lipid composition of their membranes: altering the ratio between bola- and bilayer lipids in their membranes [8, 9] and/or by changing the number of cyclopentane rings in their lipid tails, which are believed to make lipid molecules more rigid [5]. For example, *Thermococcus kodakarensis* increases its tetraether bolalipid ratio from around 50% to over 80% when the temperature of the environment increases from 60 to 85 C [10]. Along the same lines, the cell membrane of *Sulfolobus acidocaldarius*, can contain over 90 % of bolalipids with up to 8 cyclopentane rings at 70 C and pH 2.5 [5, 11]. It is worth mentioning that in exceptional cases bacteria also synthesise bolalipids in response to high temperatures [12], highlighting that the study of bolalipid membranes is relevant not only for archaeal biology but also from a general membrane biophysics perspective.”

**Reviewer #2 (Public review):**
Summary:The authors aimed to understand the biophysical properties of archeal membranes made of bolalipids. Bacterial and eukaryotic membranes are made of lipids that self-assemble into bilayers. Archea, instead, use bolalipids, lipids that have two headgroups and can span the entire bilayer. The authors wanted to determine if the unique characteristics of archaea, which are often extremophiles, are in part due to the fact that their membranes contain bolalipids.The authors develop a minimal computational model to compare the biophysics of bilayers made of lipids, bolalipids, and mixtures of the two. Their model enables them to determine essential parameters such as bilayer phase diagrams, mechanical moduli, and the bilayer behaviour upon cargo inclusion and remodelling.The author demonstrates that bolalipid bilayers behave as binary mixtures, containing bolalipids organized either in a straight conformation, spanning the entire bilayer, or in a u-shaped one, confined to a single leaflet. This dynamic mixture allows bolalipid bilayers to be very sturdy but also provides remodelling. However, remodelling is energetically more expensive than with standard lipids. The authors speculate that this might be why lipids were more abundant in the evolutionary process. Strengths:This is a wonderful paper, a very fine piece of scholarship. It is interesting from the point of view of biology, biophysics, and material science. The authors mastered the modelling and analysis of these complex systems. The evidence for their findings is really strong and complete. The paper is written superbly, the language is precise and the reading experience is very pleasant. The plots are very well-thought-out.Weaknesses:I would not talk about weaknesses, because this is really a nice paper. If I really had to find one, I would have liked to see some clear predictions of the model expressed in such a way that experimentalists could design validation experiments.

We thank the reviewer for their very kind assessment. We incorporated their recommendations regarding experimental validation in the discussion section, as follows (p.14):

“Our model makes a number of predictions that could be tested by experiment either in cells or in vitro. First, it predicts that a small increase in the fraction of archaeal bilayer lipids should be sufficient to soften a bolalipid-rich membrane. While this could be tested in the future, so far only very few studies have yet reported experimental analysis of archaeal membrane mixtures [18, 50]. Second, we observed that membranes with moderate bolalipid molecular rigidity *k*_bola_ exhibit curvature-dependent bending rigidity. To experimentally verify this, one could extrude membrane tethers from cells while controlling for membrane tension. Finally, to get to the core mechanism underlying our findings, it will be important to develop experimental methods that will allow the fraction of U-shaped bolalipid conformers per leaflet to be imaged and measured.”

**Reviewer #3 (Public review):**
Summary:The authors have studied the mechanics of bolalipid and archaeal mixed-lipid membranes via comprehensive molecular dynamics simulations. The Cooke-Deserno 3-bead-per-lipid model is extended to bolalipids with 6 beads. Phase diagrams, bending rigidity, mechanical stability of curved membranes, and cargo uptake are studied. Effects such as the formation of U-shaped bolalipids, pore formation in highly curved regions, and changes in membrane rigidity are studied and discussed. The main aim has been to show how the mixture of bolalipids and regular bilayer lipids in archaeal membrane models enhances the fluidity and stability of these membranes.Strengths:The authors have presented a wide range of simulation results for different membrane conditions and conformations. For the most part, the analyses and their results are presented clearly and concisely. Figures, supplementary information, and movies very well present what has been studied. The manuscript is well-written and is easy to follow.

We thank the reviewer for the detailed assessment of our work and their constructive feedback.

Major issuesR3.Q1: The Cooke-Deserno model, while very powerful for biophysical analysis of membranes at the mesoscale, is very much void of chemical information. It is parametrized such that it is good in producing fluid membranes and predicting values for bending rigidity, compressibility, and even thermalexpansioncoefficientfallingintheacceptedrangeofvaluesforbilayermembranes. But it still represents a generic membrane. Now, the authors have suggested a similar model for the archaeal bolalipids, which have chemically different lipids (the presence of cyclopentane rings for one), and there is no good justification for using the same pairwise interactions between their representative beads in the coarse-grained model. This does not necessarily diminish the worth of all the authors’ analyses. What is at risk here is the confusion between ”what we observe this model of bolalipidor mixed-membranes do” and ”how real bolalipid-containing archaeal membranes behave at these mechanical and thermal conditions.”.

As the reviewer correctly notes, Cooke and Deserno used a minimal model, devoid of chemical detail, to represent fluid lipid membranes composed of bilayer lipids. Indeed archaeal lipids are chemically different compared to non-archaeal lipids, but just like non-archaeal lipids, they can be very different from one another. Given the chemical diversity of bolalipids between each other, instead of representing their complexity in a complicated model with many experimentally unconstrained parameters, we here defined a minimal model for bolalipids. The power of this minimal model is to represent the key physical/geometrical characteristics of archaeal membranes, namely the fact that lipid heads on two sides of the membrane are often connected, that bolalipids can exhibit a conformational change, and that bolalipids mix with some percentage of bilayer molecules. We then ask a general question: how do these unique geometrical characteristics of archaeal membranes influence their mechanics and reshaping? The reviewer is however right in pointing out that a model, regardless of its level of details (atomistic, coarse-grained, minimal), is still a model.

Our approach of extending an established coarse-grained model for bilayer lipids to bolalipids is further supported by experimental observations, which report that archaeal bilayer lipids can form membranes of comparable bending rigidity to those of non-archaeal bilayer membranes [53]. Hence, different lipid linkages (archaeal vs. non-archaeal) give rise to fluid, deformable membranes of not too dissimilar rigidities, suggesting that both archaeal and non-archaeal bilayer lipids can be represented by a similar minimal coarse-grained model for the purpose of mesoscopic biophysical investigations. Since archaeal bolalipids have the same core chemical structure as two archaeal bilayer lipids joined by their tail ends, similarly we model a bolalipid by joining two bilayer lipids. Such an approach also efficiently enables us to compare bolalipid with bilayer membranes, and connect to the large body of knowledge on the physics of bilayer membranes.

To conclude, our coarse-grained model is indeed intended to capture the main physical properties of bolalipid membranes, and not their chemical diversity.

R3.Q2: Another more specific, major issue has to do with using the Hamm-Kozlov model for fitting the power spectrum of thermal undulations. The 1/*q*^2^ term can very well be attributed to membrane tension. While a barostat is indeed used, have the authors made absolutely sure that the deviation from 1/*q*^4^ behaviour does not correspond to lateral tension?

To the casual observer, any 1/*q*^2^ trend might point at membrane tension. However, the precise functional form is relevant as it determines whether the 1/*q*^2^ dominates the 1/*q*^4^ trend for small or large values of the wave number *q* in the fitted power spectrum.

The first model (including lipid tilt) exhibits the functional form 1/(*kq*^4^) + 1/(*kq*^2^). In contrast, the second model (including membrane tension) exhibits the functional form 1/(*kq*^4^ + ∑*q*^2^). Importantly, the two models obey a different functional form. Here *k* and *kθ*, are the bending and tilt moduli, which are assumed positive, and ∑ is the membrane tension, which can be either positive or negative. For the first model (with tilt), while for small *q* the amplitude is proportional to *q*^-4^, for large *q* the amplitude is proportional to *q*^-2^. In contrast, for the second model (with positive tension) while for small *q* the amplitude is proportional to *q*^-2^, for large *q* the amplitude is proportional to *q*^-4^. If membrane tension were to be negative in the second model, the slope would cross from negative infinity for small *q* to -4 for large *q*. The functional dependencies are summarized in Author response image 1A.

For rigid bolalipid membranes, it is clearly visible that the slope of the power spectrum plotted against the wave number *q* decreases with increasing *q* (Author response image 1B). While the slope initially assumes a value close to 4, it gradually approaches 2 for larger values of *q*. We conclude that only the model including lipid tilt can fit the power spectrum of membrane fluctuations appropriately (solid-dashed line), whereas the model with tension fails to fit the data (dashed line). We note that the combined model containing both lipid tilt and membrane tension does not give a better fit (dotted line).

To demonstrate that the tension model cannot fit the data, we included the best fits for both models for rigid bolalipid membranes in the new SI section 16 (p. S22) and show that only the tilt model leads to acceptable fits. We also measured the projected membrane tension - \begin{document}$\left\langle P_{x}+P_{y}\right\rangle L_{z} / 2$\end{document}, where *Px,Py* are respectively the pressure in *x* and *y* direction and *Lz* is the dimension of the simulation box in *z* axis. We found the projected membrane tension to give a negligible value similarly to the one that we indirectly measured by fitting a combined model with both tension and tilt, further confirming our conjecture.

**Author response image 1. sa3fig1:** (**A**) Schematic showing the decay of the power spectrum as a function of the wave number *q* in the tilt model (top), in the tension model with positive membrane tension (middle), and in the tension model with negative membrane tension (bottom). (**B**) Fitted power spectrum as a function of *q* for rigid bolalipid membranes (*k*_bola_=5*k*_B_*T*). The fit shows that while the model with tension (dashed line) cannot fit the data, the model with tilt nicely fits the spectrum (solid-dashed line). The combined model including both tension and tilt does not fit the spectrum any better (dotted line).

R3.Q3: I got more worried when I noticed in the SI that the simulations had been done with combined ”fix langevin” and ”fix nph” LAMMPS commands. This combination does not result in a proper isothermal-isobaric ensemble. The importance of tilt terms for bolalipids is indeed very interesting, but I believe more care is needed to establish that.

In what follows, we show that there is no reason to worry. First of all we want to clarify that the physical setup we simulate is that of a membrane contained in a heat bath under negligible tension with correct diffusional dynamics. To achieve this physical setup, for which we use a Langevin thermostat combined with pressure control via an overdamped barostat, which we implement in LAMMPS by combining ”fix langevin” and ”fix nph”.

In more detail: we simulated particles in an implicit solvent, for which we use a Langevin thermostat to get the right diffusional dynamics. To apply the theory of fitting fluctuation spectrums the simulation box length needs to be (near) constant. However, simulating membranes at a fixed box size results in an average non-zero membrane tension, making it hard to measure bending rigidity. The reason is that the effect of membrane tension is most influential on the largest wavelength modes, which are also most decisive when determining mechanical membrane properties like membrane rigidity. To minimize the effect of tension, we perform our simulation with an overdamped barostat (*𝜏*_baro_ = 10 *𝜏*
_langevin_), which keeps the membrane near tensionless, as also done before [32]. In the revised manuscript, we have clarified the statement on the physical ensemble used (p.S2):

“For simulating flat membrane patches of bolalipids, we combined the previously used Langevin thermostat with relaxation time of 1*𝜏* with a Nosé–Hoover barostat with relaxation time of 10*𝜏*. In LAMMPS this amounts to combining the commands ’fix langevin’ with ’fix nph’. We configured the barostat to set lateral pressure *Pxy* to zero by re-scaling the simulation box in the *x-y* plane. We compare this setup to a fixed box length setup, and an NPT ensemble setup, in SI section 17.”

To connect our results with statistical mechanics ensemble theory we tested alternative setups. Similar setups, including the formal isothermal-isobaric ensemble, where *N,P,T* are kept constant using Nose-Hoover style equations for thermostating and barostating with modern corrections [34], which the reviewer refers to, result in very similar fluctuation spectrums. Consequently, our measurements of bending and tilt modulus hold true regardless of the integration scheme. However, such a setup does not correctly capture implicit solvent and diffusional dynamics.

In even more detail: we tested our setup (implemented via ”fix langevin”+”fix nph”) versus a isothermal-isobaric ensemble (implemented via ”fix npt”). We measured volume mean and standard deviation, and found them matching for a reference LJ gas.

To be completely sure, and to please the reviewer, we have performed additional verifications in the new SI section 17, which we summarize in the following. We simulated three representative membranes with different integration schemes: ”fix npt”, ”fix langevin”+”fix nph”, and ”fix langevin” (Langevin dynamics with projected area fixed at the average value obtained from a ”langevin+nph”). We checked that the ”fix nph” barostat is merely equilibrating the membrane to a tensionless configuration, after which the projected membrane area (*Ap* = *LxL<sub<y*) is practically constant. Consequently, the different schemes resulted in minor changes in the longest wavelength modes that we tracked down to small changes in the negligible tension. The resulting measurements of bending modulus change by less than 10%, and our main text conclusions do not change. Author response image 2 compares the fluctuation spectrums for the different integration schemes.

**Author response image 2. sa3fig2:** Height fluctuation spectrum, for a bilayer membrane at *T*_eff_ =1. 1, simulated with Langevin dynamics (pink, ‘langevin‘), our setup (purple, ‘nph+langevin‘), and under an isothermal-isobaric ensemble (blue, ‘npt‘); fits are shown as dotted lines.

R3.Q4: This issue is reinforced when considering Figure 3B. These results suggest that increasing the fraction of regular lipids increases the tilt modulus, with the maximum value achieved for a normal Cooke-Deserno bilayer void of bolalipids. But this is contradictory. For these bilayers, we don’t need the tilt modulus in the first place.

We understand the concern why this might be counter-intuitive, and we thank the reviewer for pointing it out. We first want to stress that the tilt modulus can also be measured for bilayer membranes even if it is not needed to fit the fluctuation spectrum. If we measure the tilt modulus for a bilayer membrane, we obtain a value similar to the previously measured one [36]. Importantly, here we also report measurements for the tilt modulus for bolalipid membranes.

To understand the seemingly contradictory behaviour of the tilt modulus, it is insightful to rewrite the expression for the fluctuation spectrum as done in Eq. (1):\begin{document}$$\displaystyle \left.\left.\langle | h_{n}\right|^{2}\right\rangle & =\frac{k_{\mathrm{B}} T}{L^{2}}\left(\frac{1}{\kappa q^{4}}+\frac{1}{\kappa_{\theta} q^{2}}\right) \\ & =\frac{k_{\mathrm{B}} T}{L^{2}} \frac{1}{\kappa q^{4}}\left(1+q^{2} l_{\theta}^{2}\right),$$\end{document}

where \begin{document}$l_{\theta}=\sqrt{\kappa / \kappa_{\theta}}$\end{document} is a characteristic length scale related to tilt, which we call the tilt persistence length. From the last equation it is easy to see that the tilt modulus *𝜅*_𝜃_ becomes relevant for the fluctuation spectrum if the tilt persistence length *l*_𝜃_ is not negligible. In other words, this means that we have to consider the tilt modulus *𝜅*_𝜃_ as relevant, if it is sufficiently small compared to the bending rigidity *𝜅*.

However, this is not only counter-intuitive, but also difficult to communicate graphically. Per the excellent reviewer’s suggestion, to make the interpretation more accessible, we converted in the main text and its figures the tilt modulus to the more directly interpretable tilt persistence length *l*_𝜃_, as this is small when tilt is irrelevant (for bilayer lipids and flexible bolalipids) and large otherwise (for rigid bolalipids). This includes changes to the main text on p.6 and p.8 , and to the insets in Figs. 2C and 3B. We note that for completeness we also report the tilt modulus 𝜅_𝜃_ in the SI.

R3.Q5: Also, from the SI, I gathered that the authors have neglected the longest wavelength mode because it is not equilibrated. If this is indeed the case, it is a dangerous thing to do, because with a small membrane patch, this mode can very well change the general trend of the power spectrum. As a lot of other analyses in the manuscript rely on these measurements, I believe more elaboration is in order.

We thank the reviewer for the careful examination of our supplementary material. For each fluctuation spectrum measurement, we ran multiple replicas. We observed that the largest wavelength modes were not fully equilibrated. In the simulations the first mode of the fluctuation spectrum is probed at different amplitudes and phases. We thus expected the potential systematic error would show up clearly when comparing spectrums of the different replicas. As we saw no correlation in these systematic offsets between replicas, we concluded that the simulations are sufficiently equilibrated and we could safely exclude the first mode of the fluctuation spectrum from our analysis.

To show without doubt that this procedure does not randomly bias our results, we also ran simulations for three representative membranes until all modes were equilibrated. On the modes previously equilibrated, the resulting spectrums agree with our previous shorter simulations. On the largest wavelength modes that were previously not fully equilibrated, we noticed a small deviation from theory, specifically for flexible membranes (small bending modulus). These small deviations can be explained by including a negligible negative tension. Importantly, however, the resulting bending modulus *σ* stays nearly the same. We note that the small negative tension disappears when we halve the timestep (see Author response image 3). This verification is shown in SI section 17.

R3.Q6: The authors have found that ”there is a strong dependency of the bending rigidity on the membrane mean curvature of stiffer bolalipids.” The effect is negative, with the membrane becoming less stiff at higher mean curvatures. Why is that? I would assume that with more flexible bolalipids, the possibility of reorganization into U-shaped chains should affect the bending rigidity more (as Figure 2E suggests). While for a stiff bolalipid, not much would change if you increase the mean curvature. This should be either a tilt effect, or have to do with asymmetry between the leaflets. But on the other hand, the tilt modulus is shown to decrease with increasing bolalipid rigidity. The authors get back to this issue only on page 10, when they consider U-shaped lipids in the inner and outer leaflets and write, ”this suggested that an additional membrane-curving mechanism must be involved.” But then again, in the Discussion, the authors write, ”It is striking that membranes made from stiffer bolalipids showed a curvature-dependent bending modulus, which is a clear signature that bolalipid membranes exhibit plastic behaviour during membrane reshaping,” adding to the confusion.

**Author response image 3. sa3fig3:** Height fluctuation spectrum, for a bilayer membrane at *T*_eff_ =1. 1, as simulated in the main text (grey, for 60⇥10^3^τ), for longer duration (1_.*44⇥10^6^τ) (pink), and with the longer duration and halved timestep = 0.005*τ_(purple); fits are shown as dotted lines (tension and tilt) or dash-dot lines (tilt only).

We thank the reviewer for asking this important question. Membrane bending rigidity in bolalipid membranes decreases dramatically once a small fraction of U-shapes is allowed to form, but then plateaus once this U-shape fraction reaches 20%. In a curved bolalipid membrane, U-shapes must accumulate in the outer leaflet to accommodate for area difference. Together, the bending rigidity non-linear dependence on U-shape fraction, and the promotion of U-shapes by curvature, explain why in a membrane made of moderately stiff bolalipids (*k*_bola_ = 1*k*_B_*T*), which contain very few U-shapes in the flatstate, the bending rigidity of the membrane decreases as curvature increases. While in a membrane made of flexible bolalipid molecules (*k*_bola_ = 0), where many U-shapes are present in the flat membrane, the bending rigidity does not change with curvature.

Bending rigidity 𝜅 in flat membranes composed of bolalipids decreases dramatically once a small fraction of U-shapes is allowed to form, but plateaus once more than 20% of U-shaped bolalipids are present. In details, our data shows that with an increasing bolalipid molecular rigidity *k*_bola_, both the number of U-shaped bolalipids decreases (Fig. 2B) and the membrane rigidity *𝜅* increases (Fig. 2C). Thus, the correlation suggests that U-shaped bolalipids soften the membrane, in a non-linear way where most of the change in membrane bending rigidity happens for U-shaped bolalipid fraction *<* 20% (Figure S11).

Separately, membrane curvature affects the area difference between curved membrane leaflets and thus drives U-shape accumulation. To be specific, a cylindrical membrane with area *A*, mean curvature *H* and thickness *h* has the outer leaflet with area *A*(1 + *Hh*) and the inner leaflet with smaller area *A*(1 *Hh*). This can be large, in our simulations up to an area change of *Hh* = 25%. For pure bolalipid membranes, straight bolalipids occupy the same space in each leaflet. Area difference can then be achieved only by having a different amount of U-shaped bolalipids in each leaflet, which can result in a different U-shape fraction between leaflets and thus ’asymmetry between leaflets’. Figure S10 confirms U-shape head fraction asymmetry that increases with curvature, for both flexible (*k*_bola_ = 0) and moderately stiff bolalipids (*k*_bola_ = 1*k*_B_*T*).

Together, these two effects result in membrane softening under curvature for the moderately stiff bolalipids, but constant rigidity for flexible bolalipids (Fig. 2F). In details: for membranes composed of moderately stiff bolalipid molecules (*k*_bola_ = 1*k*_B_*T*), the U-shape bolalipid head fraction only increases in the outer leaflet, goingfrom10to20%(Figure S10). This is in the high sensitivity region where the bending rigidity is expected to change the most (Figure S11). We hypothesize that the molecular rigidity of a U-shaped bolalipid creates compression on the outer leaflet that stabilizes the membrane curvature and thus causes membrane softening. We suspect that for membranes composed of rigid bolalipids (*k >* 1*k*_B_*T*), the effect is likely not present due to the absence of U-shape formation even under strong bending.

By contrast, for membranes composed of flexible bolalipids (*k* = 0), the U-shaped bolalipid head fraction changes relatively little from its value for flat membranes (from 50% to respectively 60 and 40% for the outer and inner leaflet, Figure S10). This is in the region where the membrane bending rigidity is expected to respond weakly to U-shape fraction (Figure S11). Additionally, the change is symmetric, so presumably the outer leaflet becomes softer as the inner leaflet becomes stiffer, thus creating opposing effects and only weakly affecting the membrane bending rigidity as a whole. We note that the distinction between the U-shape head fraction that we plot (Figure S10) and U-shape fraction (Figure S11) matters little for this analysis.

We have added this deduction and its plots to SI section 8, and revised the corresponding statement in the main text accordingly (p.7).

“Changing membrane curvature alters the area differently in the two membrane leaflets. To adapt to the area difference, we thus expect the fraction of U-shaped bolalipids to change as the membrane curvature changes. Moreover, the results of Fig. 2B and Fig. 2C showed that the U-shaped bolalipid fraction and the membrane bending rigidity are correlated. As a result, we predict that the fraction of straight versus U-shaped bolalipids in a membrane will change in response to membrane bending, in a way that makes the bending rigidity of a bolalipid membrane curvature dependent.”

R3.Q7: This issue is repeated when the authors study nanoparticle uptake. They write: ”to reconcile these seemingly conflicting observations we reason that the bending rigidity, similar to Figure 2F, is not constant but softens upon increasing membrane curvature, due to dynamic change in the ratio between bolalipids in straight and U-shaped conformation. Hence, bolalipid membranes show stroking plastic behaviour as they soften during reshaping.” But the softening effect that they refer to, as shown in Figure 4B, occurs for very stiff bolalipids, for which not much switching to U-shaped conformation should occur.

We thank the reviewer for locating a particularly dense sentence. We changed the text to explicitly refer to the range *k* 2 [0,2] *k*_B_*T* for which there is significant change in U-shape fraction (p.8):

“To reconcile these seemingly conflicting observations we reason that the bending rigidity κ, similar to Fig. 2F, is not constant but softens in the range *k* 2 [0,2] *k*_B_*T*, upon increasing membrane curvature. This is due to the dynamic change in the ratio between bolalipids in straight and U-shaped conformation.”

As for Fig. 4B, for *k* > 2*k*_B_*T*, pores form thus explaining the plateau in adsorption energy.

R3.Q8: Another major issue is with what the authors refer to as the ”effective temperature”. While plotting phase diagrams for kT/eps value is absolutely valid, I’m not a fan of calling this effective temperature. It is a dimensionless quantity that scales linearly with temperature, but is not a temperature. It is usually called a ”reduced temperature”. Then the authors refer to their findings as studying the stability of archaeal membranes at high temperatures. I have to disagree because eps is not the only potential parameter in the simulations (there are at least space exclusion and angle-bending stiffnesses) so one cannot identify changing eps with changing the global simulation temperature. This only works when you have one potential parameter, like an LJ gas.

We indeed thought about this before and found that it makes little difference in our set-up. To thoroughly show that the distinction matters very little, per reviewer’s question, we computed our phase diagrams by scaling temperature *T* explicitly (and not lipid tail interactions *T*_eff_ = *k*_B_*T* /ϵ_p_). We added these results to the SI section 14 and found no significant difference when comparing scaling tail interactions (Figure S15A) with scaling temperature explicitly (Figure S15B).

We also computed Fig. 2A-C for scaling interactions (Figure S17A) and scaling temperature explicitly (Figure S17B). We found a slightly increased U-shaped bolalipid fraction for low *k* when comparing scaling interactions (Figure S17A) with temperature scaling (Figure S17B). The reason is that the U-shaped fraction depends on temperature, as with higher temperature bolalipids can easier transition into the U-shape. Most importantly, however, we found no qualitative changes on the liquid region or the mechanical membrane properties when we compared the different scaling variants.

The reason why both scaling variants match so well can be understood easily. All pair potentials, including volume exclusion interactions between head beads and other membrane beads, were also scaled in the same manner as tail-to-tail interactions, as described in the SI. In contrast, the energy scales for maintaining the lipid bonds, the bilayer lipid angles and the bolalipid angles are relatively large compared to the energy scales involved in tail-to-tail interactions. This separation of energy scales guarantees that there will be little effect when increasing global temperature. Regarding nomenclature, we take the reviewer’s advice and have added ’reduced temperature’ as an alias for *T*_eff_ in the main text.

In the revised version of the manuscript, we mention these observations in the SI section 14 and point towards these results in the main text (p.4):

“This interaction strength governs the membrane phase behaviour and can be interpreted as the effective temperature or reduced temperature *T*_eff_ = *k*_B_*T* /ϵ_p_. As the distinction between scaling interactions (*T*_eff_) or temperature (*T*) is not important for our analysis (see Supplemental Information (SI) section 14), for simplicity we refer to *T*_eff_ as temperature in the following.”

Minor issuesR3.Q9: As the authors have noted, the fact that the membrane curvature can change the ratio of U-shaped to straight bolalipids would render the curvature elasticity non-linear (though the term ”plastic” should not be used, as this is still structurally reversible when the stress is removed. Technically, it is hypoelastic behaviour, possibly with hysteresis.) With this in mind, when the authors use essentially linear elastic models for fluctuation analysis, they should make a comparison of maximum curvatures occurring in simulations with a range that causes significant changes in bolalipid conformational ratios.

We thank the reviewer for their suggestion on calling the non-linear behaviour of the curvature elasticity hypoelastic. We have edited the main text accordingly (p.8):

“In an elastic material, the strain modulus holds constant and deformation is reversible. For bolalipid membranes at *k* = 1*k*_B_*T*, however, the bending modulus decreases when deformation increases, rendering bolalipid membranes hypoelastic.”

Moreover, regarding the maximum curvatures occurring in the fluctuation simulations: We first note that the ensemble average of the mean curvature *H* from the fluctuation measurements is indicated as a vertical line in Fig. 2F. As the average value is nearly zero, the membrane can be considered as flat in good approximation. To investigate the question in more detail, we extended the SI with a careful analysis of the validity of the maximum membrane curvature and the validity of the Monge gauge approximation (SI section 15).

In short, we found that the involved membrane curvatures are small and therefore are unlikely to trigger any significant changes of the bending modulus. Moreover, since we are dealing with two bolalipid conformations, we also tested the homogeneity of the membrane. In our simulations of flat membrane patches we did not observe clustering or phase separation between the two bolalipid conformations beyond the [2,3]σ range. Furthermore, we get good agreement between our fluctuation measurement and the cylinder simulations in Fig. 2F. We now mention this verification in the revised version of the manuscript (p.8):

“Fortunately, this dependency on curvature does not invalidate our fluctuation results, where the curvature is small enough that its effect on the bending modulus is negligible (SI section 15).”

Last but least, simulating bending/unbending cycles of an arc-shaped membrane (frozen endpoints) shows agreement with cylinder membrane simulations, and no hysteresis at the rates of deformation employed (cf. M. Amaral’s thesis [54], soon to be out of the embargo period).

R3.Q10: The Introduction section of the manuscript is written with a biochemical approach, with very minor attention to the simulation works on this system. Some molecular dynamics works are only cited as existing previous work, without mentioning what has already been studied in archaeal membranes. While some information, like the binding of ESCRT proteins to archaeal membranes, though interesting, helps little to place the study within the discipline. The Introduction should be revised to show what has already been studied with simulations (as the authors mention in the Discussion) and how the presented research complements it.

The present research for the first time covers archaeal membranes with a single coarse-grained model capable of assuming both bolalipid in-membrane conformations and sweeps through temperature, membrane composition, and molecular rigidity. The work shows the first curvature dependent bending modulus for pure bolalipid membranes. It also investigates systematically bending modulus and Gaussian modulus, and tests the model in an all-encompassing budding simulation that incorporates topology changes. Existing atomistic or coarse-grained MD simulations (MARTINI or similar force fields) are limited to small patches of membrane, with no study of large-scale deformations or topology changes; plus, they rely on force fields that were parametrized for bilayer membranes.

To give a comprehensive overview of the field, we revised the introduction section of the manuscript, in which we now discuss previous computational work investigating membrane diffusivity, U-shaped lipid fraction, and bending rigidity (p.3):

“By contrast, only a few studies have investigated bolalipid membranes applying computational or theoretical tools [24, 25]. Specifically, the pore closure time in bolalipid membranes, and the role of cyclopentane rings for membrane properties has been investigated using all-atom simulations, showing decreased lateral mobility, reduced permeability to water, and increased lipid packing [26–28]. Moreover, using coarse-grained simulations, it was suggested that bolalipid membranes are thicker [29], exhibit a gel-to-liquid phase transition at higher temperature [30], and exhibit a reduced diffusivity [31]. However, little research has been devoted to investigating mechanics and reshaping of bolalipid membranes at the mesoscale despite the obvious importance of this question from evolutionary, biophysics, and biotechnological perspectives and although different membrane physics is expected to manifest.”

Following the reviewer’s advice and to keep the introduction concise and focused on bolalipid membranes, we have removed the paragraph on ESCRT-III proteins in the revised manuscript.

R3.Q11: The authors have been a bit loose with using the term ”stability”. I’d like to see the distinction in each case, as in ”chemical/thermal/mechanical/conformational stability”.

We have clarified when applicable the type of stability throughout the manuscript. In all other instances, if not clear from context, we mean simply that the membrane persists being a membrane. At our coarse-grained level, this means the membrane does not disassemble into a gas phase.

R3.Q12: In the original Cooke-Deserno model, a so-called ”poorman’s angle-bending term” is used, which is essentially a bond-stretching term between the first and third particle. However, I notice the authors using the full harmonic angle-bending potential. This should be mentioned.

This is made clear in the SI (Eq. (S3)). Cooke and Deserno mention the harmonic angle potential as a valid alternative in their original publication. We now also added this detail to the main text (p.3):

“The angle formed by the chain of three beads is kept near 180° via an angular potential with strength *k*_0_, instead of the approximation by a bond between end beads of the original model [32].”

R3.Q13: The analysis of energy of U-shaped lipids with the linear model *E* = *c*_0_ + *c*_1_*k* is indeed very interesting. I am curious, can this also be corroborated with mean energy measurements? The minor issue is calling the source of the favorability of U-shaped lipids ”entropic”, while clearly an energetic contribution is found. The two conformations, for example, might differ in the interactions with the neighbouring lipids.

We were also curious and thank the reviewer for the suggestion of mean energy measurements. We concluded that there must be either an entropic contribution to the free energy or an intermolecular interaction energy favouring U-shaped bolalipids. We have now included these measurements in SI section 6 (p.S5):

“By splitting the average potential energy between an internal contribution (bonds, angles and pair interactions between particles in the same molecule) and an external contribution (pair interactions between a molecule and its neighbours), we determined the transition energy from straight to U-shaped bolalipids in detail. We found that this transition lowers the internal potential energy of the bolalipid while increasing its interaction energy. In total, we obtained an energy barrier for the transition of Δ*E*_s→u_ = 0.79±0.01*k*_B_*T*. Since the fit indicates, however, that the U-shaped bolalipid conformation is preferred over the straight conformation, we conclude that there must be either an entropic contribution to the free energy or an intermolecular interaction energy favouring U-shaped bolalipids.”

We refer to these measurements in the main text (p.6):

“For the fit it appears that *c*_0_ < 0, which implies that bolalipids in U-shape conformation are slightly favoured over straight bolalipids at *k* = 0 (explored in SI section 6).”

R3.Q14: The authors write in the Discussion, ”In any case, our results indicate that membrane remodelling, such as membrane fission during membrane traffic, is much more difficult in bolalipid membranes [34].” Firstly, I’m not sure if studying the dependence of budding behaviour on adhesion energy with nanoparticles is enough to make claims about membrane fission. Secondly, why is the 2015 paper by Markus Deserno cited here?

We thank the reviewer for giving us the opportunity to clarify. We make an energetic argument on membrane fission based on the observed difference in the ratio of \begin{document}$|\bar{\kappa}| / \kappa$\end{document}.

Splitting a spherical membrane vesicle into two spherical vesicles (fission) increases the bending energy by 8𝜋𝜅 and decreases the energy related to the Gaussian bending modulus by \begin{document}$4 \pi \mid \bar{\kappa}$\end{document}. The second part of the argument is given for example in the review by Markus Deserno (p.23, right column), that’s why we cite the paper here. Together, this gives an energy barrier, required for membrane fission in the considered geometry of ∆*E*_fission_ = \begin{document}$4 \pi \kappa(2-|\bar{\kappa}| / \kappa)$\end{document}. We found that *𝜅* was typically larger in bolalipid membranes we thus expect the energy barrier for fission ∆\begin{document}$|\bar{\kappa}| / \kappa$\end{document} is around 0.5 for bolalipid membranes and around 1 for bilayer membranes. Since *E*_fission_ to be larger for bolalipid membranes. We therefore predict that membrane remodelling, such as membrane fission during membrane trafficking, is harder in bolalipid membranes. We explain our reasoning in the discussion of the revised manuscript (p.13):

“Membrane remodelling, such as the fission of one spherical vesicle into two, increases the bending energy by 8πκ but decreases the energy related to the Gaussian modulus by –\begin{document}$4 \pi \mid \bar{\kappa}$\end{document} [39], giving rise to a fission energy barrier of ∆*E*_fission_ = \begin{document}$4 \pi \kappa(2-|\bar{\kappa}| / \kappa)$\end{document}. Our results indicated that while in bolalipid membranes *𝜅* is larger, \begin{document}$E|\bar{\kappa}| / \kappa$\end{document} is smaller compared to bilayer membranes. Our results thus predict a larger energy barrier for membrane fission ∆_fission_ in bolalipid membranes compared to bilayer membranes.”

R3.Q15: In the SI, where the measurement of the diffusion coefficient is discussed, the expression for D is missing the power 2 of displacement.

We thank the reviewer for spotting this oversight. We corrected it in the revised version of the SI (p.S5).

R3.Q16: Where cargo uptake is discussed, the term ”adsorption energy” is used. I think the more appropriate term would be ”adhesion energy”.

For the sake of simplicity, we changed the term to adhesion energy (caption of Fig. 4, and p.10). We do not have a strong opinion on this, but we believe that adsorption energy would be equally correct as we describe the adsorption of many lipid head beads to a nanoparticle.

R3.Q17: Typos:Page 1, paragraph 2: Adaption → Adaptation. Page 10, paragraph 1: Stroking → Striking.

We thank the reviewer for spotting these typos which we have corrected in the revised version of the manuscript.

**Recommendations for the authors**

**Reviewer #1 (Recommendations for the authors):**
A few thoughts (likely out of the scope of this paper but possibly to consider upon revision):R1.Q1: Do bolalipids always have the same headgroup? I don’t recall reading this in the introduction/discussion. R1 and R2 are in Figure 1, but I don’t know whether there are standard types. Could this be expanded upon? Is the model able to take these differences into account?

We thank the reviewer for raising this important question. Similar to bacteria and eukaryotes, in archaea there is a huge variety in terms of the different head groups that lipids can contain and thus also lipid variety. Most archaeal lipids have head groups that contain either phosphate groups or sugar residues. Typically, archaeal bolalipids are asymmetric and contain a phosphatidyl and a sugar moiety at the two ends of the lipid molecule. Within the membrane the lipid is oriented such that the phosphatidyl moiety points towards the interior of the cell whereas the sugar moiety points towards the outside of the cell as it occupies more space [5].

In our computational model, however, we consider symmetric bolalipids for the sake of simplicity and to decouple the role of ”connected geometry” from other effects. In principle, we could investigate the effect of lipid asymmetry by increasing the size of one of the lipid head beads. However, this investigation exceeds the scope of the present study and therefore requires future work.

In the revised version of the manuscript, we now clarify that bolalipids can have different headgroups (p.1 and the caption of Fig. 1):

“The hydrophilic heads can be composed of different functional groups with phosphatidyl and sugar being the most relevant moieties. For bolalipids the two head groups at either end of the molecule are typically distinct (Fig. 1A right) [5].”

“The hydrophilic head of a bolalipid can be composed of different functional groups represented by R1 and R2 (right).”

We also explicitly state that we neglect lipid head group asymmetry for the sake of simplicity (p.4):

“To decouple the effect of the connected geometry of the bolalipids from that of lipid asymmetry, we assume both head beads of a bolalipid to share the same properties.”

R1.Q2: Is it possible to compare the mesoscale models to either Coarse-grained or even all-atom lipid models? Have simulations previously been performed for bolalipids at those levels of description?

A few studies have investigated bolalipids membranes in simulations previously. These studies either used all-atom or coarse-grained simulations. However, none of these studies investigated how bolalipids respond to membrane deformations. Therefore, it is currently not possible to directly compare our results to studies in the literature. However, to recapitulate our predictions experimentally is certainly something that could and should be done in the future. As a reply to this reviewer and reviewer 3, we discuss the current state of modelling bolalipid membranes in simulations in the revised version of the manuscript (p.3):

“By contrast, only a few studies have investigated bolalipid membranes applying computational or theoretical tools [24, 25]. Specifically, the pore closure time in bolalipid membranes, and the role of cyclopentane rings for membrane properties has been investigated using all-atom simulations, showing decreased lateral mobility, reduced permeability to water, and increased lipid packing [26–28]. Moreover, using coarse-grained simulations, it was suggested that bolalipid membranes are thicker [29], exhibit a gel-to-liquid phase transition at higher temperature [30], and exhibit a reduced diffusivity [31]. However, little research has been devoted to investigating mechanics and reshaping of bolalipid membranes at the mesoscale despite the obvious importance of this question from evolutionary, biophysics, and biotechnological perspectives and although different membrane physics is expected to manifest.”

We want to mention, however, that we do compare membrane diffusivity, U-shaped lipid fraction, and bending rigidity to the behaviour and values that have been previously measured in simulations in the discussion section. In general, we find good agreement between our results and previously reported behaviour/values (p.13):

“While flexible bolalipid membranes are liquid under the same conditions as bilayer membranes, we found that stiff bolalipids form membranes that operate in the liquid regime at higher temperatures. These results agree well with previous molecular dynamics simulations that suggested that bolalipid membranes are more ordered and have a reduced diffusivity compared to bilayer membranes [24, 29]. In our simulations, this is due to the fact that completely flexible bolalipids molecules adopt both straight (transmembrane) as well as the U-shaped (loop) conformation with approximately the same frequency. In contrast, stiff bolalipids typically only take on the straight conformation when assembled in a membrane. These results agree with the previous coarse-grained molecular dynamics simulations using the MARTINI force field which showed that the ratio of straight to U-shaped bolalipids increased upon stiffening the linker between the lipid tails [29].

[...]

When we determined the bending rigidity of bolalipid membranes by measuring their response to thermal fluctuations, we found that membranes made from flexible bolalipids are only slightly more rigid than bilayer membranes. This result is consistent with previous atomistic simulations, which showed that the membrane rigidity was similar for membranes composed of bilayer lipids and flexible synthetic bolalipids [45].”

R1.Q3: How would membrane proteins alter the behaviour of bolalipids? Either those integral to the membrane or those binding peripherally?

The reviewer asks an important question. However, the question is difficult to answer due to its scope and the gaps in the current literature. Important examples of integral or peripheral membrane proteins that alter the behaviour of bolalipids and archaeal bolalipid membranes are involved in cell homeostasis, cell division, membrane trafficking, and lipid synthesis.

The cells of many archaeal species are enclosed in a paracrystalline protein layer called the Slayer, which is attached to the lipid membrane [4, 55]. The main function of the S-layer is to keep the cell’s shape and to protect it against osmotic stress. Due to the embedding of the S-layer in the membrane at specific locations, it is to be expected that the membrane properties are influenced by the S-layer. Furthermore, archaea execute cell division by locally reshaping the membrane using FtsZ and ESCRT-III proteins [56]. While Asgard archaeal genomes encode proteins with homology to those regulating aspects of eukaryotic membrane remodelling and trafficking [57], they have yet to be observed undergoing a process like endocytosis [58]. In addition, it has been speculated that the proteins that drive the synthesis of two diether lipids into a tetraether lipid are either membrane associated or integral membrane proteins [59].

However, to the best of our knowledge it is not known how membrane proteins specifically alter the behaviour of bolalipids. Future work will need to be executed to answer this question. Following the advice of reviewer 3 and to keep the introduction concise and focused on bolalipid membranes, we do not mention these observations in the revised manuscript.

R1.Q4: Is there a mechanism in cells to convert or switch bolalipids from a straight to a u-shaped description? Does this happen spontaneously or are there enzymes responsible for this?

We thank the reviewer for bringing up this important point. Despite the relevance of the question, little is currently known about the mechanism that make bolalipids transition between a straight and a U-shaped configuration mainly because there is to date no established experimental method.

Besides our own results, most of what we know comes from coarse-grained molecular dynamics simulations, which showed that bolalipids can spontaneously transition between the straight and U-shaped configuration [29]. In addition, by using comparative genomic analysis, it has been predicted that many archaeal species contain flippases, i.e., membrane proteins that are able, upon the consumption of energy, to transfer (flipflop) bilayer lipids between the two membrane leaflets [43]. Moreover, it has been shown that *Halobacterium salinarum* (an archaeon with a bilayer lipid membrane) [44] contains scramblases, which are membrane proteins that passively transfer bilayer lipids from one membrane leaflet to the other. It is therefore tempting to speculate that similar proteins might exist for bolalipids which could facilitate the straight to U-shaped transition.

In addition, it has been reported that vesicles composed of bolalipid membranes can undergo fusion with enveloped influenza viruses [17]. In this context, it has been suggested that the influenza fusion protein hemagglutinin may locally induce U-shaped bolalipids to facilitate membrane fusion. However, all these hints are by far no proof of a mechanism that can drive the straight to U-shaped bolalipid transition, and further work needs to be done to investigate this question in detail.

In the revised version of the manuscript, we now discuss what is known about potential mechanisms to facilitate the straight to U-shaped transition in the discussion section (p.13):

“While previous coarse-grained simulations predicted that bolalipids spontaneously transition between the straight and U-shaped conformations [29], how this happens in archaeal membranes and whether membrane proteins are involved in this conformational transition needs to be clarified in the future. Experimental studies suggest that archaeal membranes contain flippases and scramblases for the transitioning of bilayer lipids between membrane leaflets [43, 44], raising the possibility that similar proteins could also facilitate conformational transitions in bolalipids. In addition, it has been suggested that the viral fusion protein hemagglutinin could cause a transition from straight to U-shaped bolalipid conformation during the fusion of bolalipid vesicles with influenza viruses [17]. However, future investigation is required.”

R1.Q5: Ideally, coordinates and any parameter files required to run the molecular simulations should be included for reproducibility.

We absolutely share the reviewer’s concern with reproducibility and as such have included in the original submission as part of our data availability section a link to a code repository (available at: https://doi.org/10.5281/zenodo.13934991 [51]) that allows initializing and simulating flat membrane patches, with user control of the parameters explored in this paper (𝜔,*T*_eff_,*k*_bola_,*f*^bi^).

**Reviewer #2 (Recommendations for the authors):**
This is a great paper and I congratulate the authors for writing such a fine piece of scholarship. The only nitty-gritty feedback that I have is summarized in the following three points:R2.Q1: In the introduction the authors talk about archaea adapting their membrane to retain membrane fluidity. However, homeoviscous adaptation is also fundamental in bacteria and eukaryotes.

The reviewer is correct, like archaea the membranes of bacteria and eukaryotes must balance between flexibility and stability. Moreover, the cell membranes in all 3 domains of life need to maintain membrane fluidity and provide mobility to the embedded lipids and membrane proteins (homeoviscous adaptation). The general idea is that these organisms change the ratio of different lipids to change membrane properties and thereby optimally adapt to their environments [10]. Importantly, however, there are differences of how homeoviscous adaptation is maintained across the different domains of life. As a reply to this reviewer and reviewer 3, we now discuss the underlying mechanisms in the revised parts of the introduction (p.1):

“Like for bacteria and eukaryotes, archaea must keep their lipid membranes in a fluid state (homeoviscous adaptation). This is important even under extreme environmental conditions, such as hot and cold temperatures, or high and low pH values [7]. Because of this, many archaea adapt to changes in their environment by tuning the lipid composition of their membranes: altering the ratio between bola- and bilayer lipids in their membranes [8, 9] and/or by changing the number of cyclopentane rings in their lipid tails, which are believed to make lipid molecules more rigid [5]. For example, *Thermococcus kodakarensis* increases its tetraether bolalipid ratio from around 50% to over 80% when the temperature of the environment increases from 60 to 85 C [10]. Along the same lines, the cell membrane of *Sulfolobus acidocaldarius*, can contain over 90 % of bolalipids with up to 8 cyclopentane rings at 70 C and pH 2.5 [5, 11]. It is worth mentioning that in exceptional cases bacteria also synthesise bolalipids in response to high temperatures [12], highlighting that the study of bolalipid membranes is relevant not only for archaeal biology but also from a general membrane biophysics perspective.”

R2.Q2: Uncertainties in Gaussian rigidity modulus estimates are not properly reported.

The large uncertainties in the Gaussian rigidity modulus \begin{document}$\bar{K}$\end{document} were due to the fact how they were calculated. In short, \begin{document}${K}$\end{document} is determined in cap folding simulations [41] (SI section 9), by using the measured values of the dimensionless parameter 𝜉, related to the folding probability, the bending modulus 𝜅, the membrane line tension , and the cap radius R. In our case, the main source of uncertainty for determining \begin{document}$\bar{K}$\end{document} comes from the uncertainty in the measurement of the bending rigidity 𝜅. To obtain 𝜅, previously, we fitted fluctuation spectra for different seeds and only then averaged the obtained values. In the revised version of the manuscript, we now first pool the fluctuation spectra of the different simulation seeds before we fit all spectra at the same time. This new approach results in smaller uncertainties for the bending rigidity 𝜅 and also the Gaussian rigidity modulus \begin{document}$\bar{K}$\end{document}.

As a consistency check, in addition to the simulations that we previously performed at *T*_eff_ = 1.3, we have repeated the cap folding and line tension simulations at *T*_eff_ = 1.2, resulting in similar values for \begin{document}$\bar{K}$\end{document}. In the revised version of the manuscript, we report the newly calculated values and uncertainties for \begin{document}$\bar{K}$\end{document} at *T*_eff_ = 1.2 in the main text (p.8):

“At *T*_eff_ = 1.2, we obtained \begin{document}$\bar{K}$\end{document} = 4.30±0.22kBT and thus a ratio of \begin{document}$\bar{\kappa} / \kappa$\end{document} = 0.89±0.04 for bilayer membranes, similar to what has been reported previously [41]. For flexible bolalipid membranes, we got a slightly smaller value for \begin{document}$\bar{K}$\end{document} = 5.04 ± 0.37kBT. Due to the larger bending modulus, however, flexible bolalipid membranes show a significantly smaller ratio *k* = 0). At larger temperature (Teff = 1.3), the ratio can be even smaller \begin{document}$\bar{\kappa} / \kappa$\end{document} = 0.64± 0.04 (\begin{document}$\bar{\kappa} / \kappa$\end{document} = 0.45 ± 0.07 (see SI section 9).”

In addition, we report the values at *T*_eff_ = 1.3 and *T*_eff_ = 1.2 in the SI (p.S15 , Tabl. S4):

We have also adapted the discussion of the Gaussian bending modulus accordingly (p.13):

“Another marked difference between bilayer and flexible bolalipid membranes is the ratio of the Gaussian rigidity to the bending modulus. Instead of being around 1 as for bilayer membranes [41], it is around 1/2 and therefore only half of that of bilayer lipids.”

**Reviewer #3 (Recommendations for the authors):**
While I think the bulk of the work presented is useful, some of the issues that I raised in my review are indeed major. Without properly addressing them, it is hard to accept the conclusions of the manuscript. I hope the authors can address them by revising their analysis.

We thank the reviewer for their constructive feedback, which helped us to improve the manuscript. We have addressed all points raised by the reviewer in our detailed point-by-point response to the reviewer (see above). We hope the reviewer will now find it easier to accept our conclusions.

(1) R. Phillips, J. Kondev, J. Theriot, and H. Garcia, Physical biology of the cell (Garland Science, New York, 2012).

(2) H. T. McMahon and J. L. Gallop, Membrane curvature and mechanisms of dynamic cell membrane remodelling, Nature 438, 590 (2005).

(3) S. B. Gould, Membranes and evolution, Curr. Biol. 28, R381 (2018).

(4) S.-V. Albers and B. H. Meyer, The archaeal cell envelope, Nat. Rev. Microbiol. 9, 414 (2011).

(5) P. M. Oger and A. Cario, Adaptation of the membrane in Archaea, Biophys. Chem. 183, 42 (2013).

(6) K. Rastädter, D. J. Wurm, O. Spadiut, and J. Quehenberger, The Cell Membrane of Sulfolobus spp.—Homeoviscous Adaption and Biotechnological Applications, International Journal of Molecular Sciences 21, 3935 (2020).

(7) P. L.-G. Chong, Archaebacterial bipolar tetraether lipids: Physico-chemical and membrane properties, Chem. Phys. Lipids 163, 253 (2010).

(8) M. Tourte, P. Schaeffer, V. Grossi, and P. M. Oger, Functionalized Membrane Domains: An Ancestral Feature of Archaea?, Front. Microbiol. 11, 526 (2020).

(9) Y. H. Kim, G. Leriche, K. Diraviyam, T. Koyanagi, K. Gao, D. Onofrei, J. Patterson, A. Guha, N. Gianneschi, G. P. Holland, M. K. Gilson, M. Mayer, D. Sept, and J. Yang, Entropic effects enable life at extreme temperatures, Sci. Adv. 5, eaaw4783 (2019).

(10) M. F. Siliakus, J. van der Oost, and S. W. M. Kengen, Adaptations of archaeal and bacterial membranes to variations in temperature, pH and pressure, Extremophiles 21, 651 (2017).

(11) D. W. Grogan, Phenotypic characterization of the archaebacterial genus sulfolobus: comparison of five wild-type strains, J. Bacteriol. 171, 6710 (1989).

(12) D. X. Sahonero-Canavesi, M. F. Siliakus, A. Abdala Asbun, M. Koenen, F. von Meijenfeldt, S. Boeren, N. J. Bale, J. C. Engelman, K. Fiege, L. Strack van Schijndel, J. S. Sinninghe Damsté, and L. Villanueva, Disentangling the lipid divide: Identification of key enzymes for the biosynthesis of membrane-spanning and ether lipids in Bacteria, Sci. Adv. 8, eabq8652 (2022).

(13) M. van Wolferen, A. A. Pulschen, B. Baum, S. Gribaldo, and S.-V. Albers, The cell biology of archaea, Nat. Microbiol. 10.1038/s41564-022-01215-8 (2022).

(14) U. Bakowsky, U. Rothe, E. Antonopoulos, T. Martini, L. Henkel, and H.-J. Freisleben, Monomolecular organization of the main tetraether lipid from Thermoplasma acidophilum at the water–air interface, Chem. Phys. Lipids 105, 31 (2000).

(15) C. Jeworrek, F. Evers, M. Erlkamp, S. Grobelny, M. Tolan, P. L.-G. Chong, and R. Winter, Structure and Phase Behavior of Archaeal Lipid Monolayers, Langmuir 27, 13113 (2011).

(16) D. P. Brownholland, G. S. Longo, A. V. Struts, M. J. Justice, I. Szleifer, H. I. Petrache, M. F. Brown, and D. H. Thompson, Phase Separation in Binary Mixtures of Bipolar and Monopolar Lipid Dispersions Revealed by 2H NMR Spectroscopy, Small Angle X-Ray Scattering, and Molecular Theory, Biophysical Journal 97, 2700 (2009).

(17) A. Bhattacharya, I. D. Falk, F. R. Moss, T. M. Weiss, K. N. Tran, N. Z. Burns, and S. G. Boxer, Structure–function relationships in pure archaeal bipolar tetraether lipids, Chem. Sci. 15, 14273 (2024).

(18) V. Vitkova, D. Mitkova, V. Yordanova, P. Pohl, U. Bakowsky, G. Staneva, and O. Batishchev, Elasticity and phase behaviour of biomimetic membrane systems containing tetraether archaeal lipids, Colloids Surf. A Physicochem. Eng. Asp. 601, 124974 (2020).

(19) E. Chang, Unusual thermal stability of liposomes made from bipolar tetraether lipids, Biochem. Biophys. Res. Commun. 202, 673 (1994).

(20) O. V. Batishchev, A. S. Alekseeva, D. S. Tretiakova, T. R. Galimzyanov, A. Y. Chernyadyev, N. R. Onishchenko, P. E. Volynsky, and I. A. Boldyrev, Cyclopentane rings in hydrophobic chains of a phospholipid enhance the bilayer stability to electric breakdown, Soft Matter 16, 3216 (2020).

(21) U. Seifert, Configurations of fluid membranes and vesicles, Adv. Phys. 46, 13 (1997).

(22) H. Noguchi, Membrane Simulation Models from Nanometer to Micrometer Scale, J. Phys. Soc. Jpn. 78, 041007 (2009).

(23) F. Frey and T. Idema, More than just a barrier: using physical models to couple membrane shape to cell function, Soft Matter 17, 3533 (2021).

(24) C. Huguet, S. Fietz, A. Rosell-Melé, X. Daura, and L. Costenaro, Molecular dynamics simulation study of the effect of glycerol dialkyl glycerol tetraether hydroxylation on membrane thermostability, Biochimica et Biophysica Acta (BBA) - Biomembranes 1859, 966 (2017).

(25) T. R. Galimzyanov, P. I. Kuzmin, P. Pohl, and S. A. Akimov, Elastic deformations of bolalipid membranes, Soft Matter 12, 2357 (2016).

(26) T. R. Galimzyanov, P. E. Volynsky, and O. V. Batishchev, Continuum elasticity and molecular dynamics of a pore in archaeal bolalipid membranes, Soft Matter 21, 687 (2025).

(27) A. O. Chugunov, P. E. Volynsky, N. A. Krylov, I. A. Boldyrev, and R. G. Efremov, Liquid but Durable: Molecular Dynamics Simulations Explain the Unique Properties of Archaeal-Like Membranes, Sci. Rep. 4, 7462 (2015).

(28) L. F. Pineda De Castro, M. Dopson, and R. Friedman, Biological Membranes in Extreme Conditions: Simulations of Anionic Archaeal, PLoS One 11, e0155287 (2016).

(29) M. Bulacu, X. Périole, and S. J. Marrink, In Silico Design of Robust Bolalipid Membranes, Biomacromolecules 13, 196 (2012).

(30) C. H. Davis, H. Nie, and N. V. Dokholyan, Insights into thermophilic archaebacterial membrane stability from simplified models of lipid membranes, Phys. Rev. E 75, 051921 (2007).

(31) S. Dey and J. Saha, Minimal Coarse-Grained Modeling toward Implicit Solvent Simulation of Generic Bolaamphiphiles, J. Phys. Chem. B 124, 2938 (2020).

(32) I. R. Cooke and M. Deserno, Solvent-free model for self-assembling fluid bilayer membranes: Stabilization of the fluid phase based on broad attractive tail potentials, J. Chem. Phys. 123, 224710 (2005).

(33) P. L.-G. Chong, U. Ayesa, V. Prakash Daswani, and E. C. Hur, On Physical Properties of Tetraether Lipid Membranes: Effects of Cyclopentane Rings, Archaea 2012, 1 (2012).

(34) A. P. Thompson, H. M. Aktulga, R. Berger, D. S. Bolintineanu, W. M. Brown, P. S. Crozier, P. J. in ’t Veld, A. Kohlmeyer, S. G. Moore, T. D. Nguyen, R. Shan, M. J. Stevens, J. Tranchida, C. Trott, and S. J. Plimpton, LAMMPS - a flexible simulation tool for particle-based materials modeling at the atomic, meso, and continuum scales, Comput. Phys. Commun. 271, 108171 (2022).

(35) A. Stukowski, Visualization and analysis of atomistic simulation data with ovito–the open visualization tool, Modelling and Simulation in Materials Science and Engineering 18, 015012 (2009).

(36) E. R. May, A. Narang, and D. I. Kopelevich, Role of molecular tilt in thermal fluctuations of lipid membranes, Physical Review E 76, 021913 (2007).

(37) W. Helfrich, Elastic Properties of Lipid Bilayers: Theory and Possible Experiments, Z. Naturforsch. C 28, 693 (1973).

(38) M. Hamm and M. Kozlov, Elastic energy of tilt and bending of fluid membranes, Eur. Phys. J. E 3, 323 (2000).

(39) M. Deserno, Fluid lipid membranes: From differential geometry to curvature stresses, Chemistry and Physics of Lipids 185, 11 (2015).

(40) V. A. Harmandaris and M. Deserno, A novel method for measuring the bending rigidity of model lipid membranes by simulating tethers, The Journal of Chemical Physics 125, 204905 (2006).

(41) M. Hu, J. J. Briguglio, and M. Deserno, Determining the Gaussian Curvature Modulus of Lipid Membranes in Simulations, Biophys. J. 102, 1403 (2012).

(42) M. Deserno, Elastic deformation of a fluid membrane upon colloid binding, Phys. Rev. E 69, 031903 (2004), arXiv: cond-mat/0303656.

(43) K. S. Makarova, M. Y. Galperin, and E. V. Koonin, Comparative genomic analysis of evolutionarily conserved but functionally uncharacterized membrane proteins in archaea: Prediction of novel components of secretion, membrane remodeling and glycosylation systems, Biochimie 118, 302 (2015).

(44) A. Verchère, W.-L. Ou, B. Ploier, T. Morizumi, M. A. Goren, P. Bütikofer, O. P. Ernst, G. Khelashvili, and A. K. Menon, Light-independent phospholipid scramblase activity of bacteriorhodopsin from *Halobacterium salinarum*, Sci. Rep. 7, 9522 (2017).

(45) T. B. H. Schroeder, G. Leriche, T. Koyanagi, M. A. Johnson, K. N. Haengel, O. M. Eggenberger, C. L. Wang, Y. H. Kim, K. Diraviyam, D. Sept, J. Yang, and M. Mayer, Effects of lipid tethering in extremophile-inspired membranes on H(+)/OH(-) flux at room temperature, Biophys. J. 110, 2430 (2016).

(46) R. Xu, A. Dehghan, A.-C. Shi, and J. Zhou, Elastic property of membranes self-assembled from diblock and triblock copolymers, Chem. Phys. Lipids 221, 83 (2019).

(47) Z. Dogic and S. Fraden, Ordered phases of filamentous viruses, Curr. Opin. Colloid Interface Sci. 11, 47 (2006).

(48) E. Barry and Z. Dogic, Entropy driven self-assembly of nonamphiphilic colloidal membranes, Proc. Natl. Acad. Sci. U.S.A. 107, 10348 (2010).

(49) A. J. Balchunas, R. A. Cabanas, M. J. Zakhary, T. Gibaud, S. Fraden, P. Sharma, M. F. Hagan, and Z. Dogic, Equation of state of colloidal membranes, Soft Matter 15, 6791 (2019).

(50) M. Saracco, P. Schaeffer, M. Tourte, S.-V. Albers, Y. Louis, J. Peters, B. Demé, S. Fontanay, and P. M. Oger, Bilayer-Forming Lipids Enhance Archaeal Monolayer Membrane Stability, Int. J. Mol. Sci. 26, 3045 (2025).

(51) M. Amaral, archaeal_membranes : code and examples (2024), available at https://doi.org/10.5281/zenodo. 13934991.

(52) M. F. Ergüder and M. Deserno, Identifying systematic errors in a power spectral analysis of simulated lipid membranes, The Journal of Chemical Physics 154, 214103 (2021).

(53) J. Genova, N. Ulrih, V. Kralj-Iglič, A. Iglič, and I. Bivas, Bending Elasticity Modulus of Giant Vesicles Composed of Aeropyrum Pernix K1 Archaeal Lipid, Life 5, 1101 (2015).

(54) M. Amaral, Archaeal Membranes: In Silico Modelling and Design, Ph.D. thesis, Institute of Science and Technology Austria (2024).

(55) M. Pohlschroder, F. Pfeiffer, S. Schulze, and M. F. A. Halim, Archaeal cell surface biogenesis, FEMS Microbiol. Rev. 42, 694 (2018).

(56) K. S. Makarova, N. Yutin, S. D. Bell, and E. V. Koonin, Evolution of diverse cell division and vesicle formation systems in Archaea, Nat. Rev. Microbiol. 8, 731 (2010).

(57) C. W. Stairs and T. J. Ettema, The Archaeal Roots of the Eukaryotic Dynamic Actin Cytoskeleton, Curr. Biol. 30, R521 (2020).

(58) B. Baum and D. A. Baum, The merger that made us, BMC Biol. 18, 72 (2020).

(59) Z. Zeng, H. Chen, H. Yang, Y. Chen, W. Yang, X. Feng, H. Pei, and P. V. Welander, Identification of a protein responsible for the synthesis of archaeal membrane-spanning GDGT lipids, Nat. Commun. 13, 1545 (2022).